# Slowly evolving dopaminergic activity modulates the moment-to-moment probability of reward-related self-timed movements

**Allison E Hamilos[1]\*, Giulia Spedicato[1], Ye Hong[1], Fangmiao Sun[2], Yulong Li[2], John A Assad[1,3]\***

[1]Department of Neurobiology, Harvard Medical School, Boston, United States; [2]State Key Laboratory of Membrane Biology, Peking University School of Life Science, Beijing, China; [3]Istituto Italiano di Tecnologia, Genova, Italy

**Abstract** Clues from human movement disorders have long suggested that the neurotransmitter dopamine plays a role in motor control, but how the endogenous dopaminergic system influences movement is unknown. Here, we examined the relationship between dopaminergic signaling and the timing of reward-related movements in mice. Animals were trained to initiate licking after a self-timed interval following a start-timing cue; reward was delivered in response to movements initiated after a criterion time. The movement time was variable from trial-to-trial, as expected from previous studies. Surprisingly, dopaminergic signals ramped-up over seconds between the start-timing cue and the self-timed movement, with variable dynamics that predicted the movement/reward time on single trials. Steeply rising signals preceded early lick-initiation, whereas slowly rising signals preceded later initiation. Higher baseline signals also predicted earlier self-timed movements. Opto-genetic activation of dopamine neurons during self-timing did not trigger immediate movements, but rather caused systematic early-shifting of movement initiation, whereas inhibition caused late-shifting, as if modulating the probability of movement. Consistent with this view, the dynamics of the endogenous dopaminergic signals quantitatively predicted the moment-by-moment probability of movement initiation on single trials. We propose that ramping dopaminergic signals, likely encoding dynamic reward expectation, can modulate the decision of when to move.

**\*For correspondence:**
allison_hamilos@hms.harvard.
edu (AEH);
jassad@hms.harvard.edu (JAA)

## Editor's evaluation

Dopamine loss in Parkinson's disease results in impaired movement initiation and execution, but the precise relationship between dopamine activity and the decision to move is poorly understood. Here, the authors imaged mesostriatal dopamine signals as head-fixed mice decided when, after a cue, to retrieve water from a spout. Surprisingly, ramps in dopamine activity predicted, even on single trials, the timing of licks. Fast ramps preceded early retrievals; slow ones preceded late ones. Optogenetic activation or suppression of dopamine activity accelerated or delayed lick initiation, respectively. Together, these findings reveal strong links between ramps in dopamine activity and the timing of self-initiated movement.

## Introduction

What makes us move? Empirically, a few hundred milliseconds before movement, thousands of neurons in the motor system suddenly become active in concert, and this neural activity is relayed via

spinal and brainstem neurons to recruit muscle fibers that power movement (*Shenoy et al., 2013*). Yet just before this period of intense neuronal activity, the motor system is largely quiescent. How does the brain suddenly and profoundly rouse motor neurons into the coordinated action needed to trigger movement?

In the case of movements made in reaction to external stimuli, activity evoked first in sensory brain areas is presumably passed along to appropriate motor centers to trigger this coordinated neural activity, thereby leading to movement. But humans and animals can also self-initiate movement without overt, external input (*Deecke, 1996*; *Hallett, 2007*; *Lee and Assad, 2003*; *Romo et al., 1992*). For example, while reading this page, you may decide without prompting to reach for your coffee. In that case, the movement cannot be clearly related to an abrupt, conspicuous sensory cue. What 'went off' in your brain that made you reach for your coffee at this *particular* moment, as opposed to a moment earlier or later?

Human movement disorders may provide clues to this mystery. Patients and animal models of Parkinson's Disease experience difficulty self-initiating movements, exemplified by perseveration (*Hughes et al., 2013*), trouble initiating steps when walking (*Bloxham et al., 1984*), and problems timing movements (*Malapani et al., 1998*; *Meck, 1986*; *Meck, 2006*; *Mikhael and Gershman, 2019*). In contrast to these self-generated actions, externally cued reactions are often less severely affected in Parkinson's, a phenomenon sometimes referred to as 'paradoxical kinesia' (*Barthel et al., 2018*; *Bloxham et al., 1984*). For example, patients' gait can be normalized by walking aids that prompt steps in reaction to visual cues displayed on the ground (*Barthel et al., 2018*).

Because the underlying neuropathophysiology of Parkinson's includes the loss of midbrain dopaminergic neurons (DANs), the symptomatology of Parkinson's suggests DAN activity plays an important role in deciding when to self-initiate movement. Indeed, pharmacological manipulations of the neurotransmitter dopamine causally and bidirectionally influence movement timing (*Dews and Morse, 1958*; *Lustig and Meck, 2005*; *Meck, 1986*; *Mikhael and Gershman, 2019*; *Schuster and Zimmerman, 1961*). This can be demonstrated in the context of *self-timed* movement tasks, in which subjects reproduce a target timing interval by making a movement following a self-timed delay that is referenced to a start-timing cue (*Malapani et al., 1998*). Species across the animal kingdom, from rodents and birds to primates, can learn these tasks and produce self-timed movements that occur, on average, at about the target time, although the exact timing exhibits considerable variability from trial-to-trial (*Gallistel and Gibbon, 2000*; *Meck, 2006*; *Mello et al., 2015*; *Merchant et al., 2013*; *Rakitin et al., 1998*; *Remington et al., 2018*; *Schuster and Zimmerman, 1961*; *Sohn et al., 2019*; *Wang et al., 2018*). In such self-timed movement tasks, decreased dopamine availability/efficacy (e.g., Parkinson's, neuroleptic drugs) generally produces late-shifted movements (*Malapani et al., 1998*; *Meck, 1986*; *Meck, 2006*; *Merchant et al., 2013*), whereas high dopamine conditions (e.g., amphetamines) produce early-shifting (*Dews and Morse, 1958*; *Schuster and Zimmerman, 1961*).

Although exogenous dopamine manipulations can influence timing behavior, it remains unclear whether endogenous DAN activity is involved in determining when to move. DANs densely innervate the striatum, where they modulate the activity of spiny projection neurons of the direct and indirect pathways, which are thought to exert a push-pull influence on movement centers (*Albin et al., 1989*; *DeLong, 1990*; *Freeze et al., 2013*; *Grillner and Robertson, 2016*). Most studies on endogenous DAN activity have focused on reward-related signals, but there are also reports of movement-related DAN signals. For example, phasic bursts of dopaminergic activity have been observed just prior to movement onset (within ~500ms; *Coddington and Dudman, 2018*; *Coddington and Dudman, 2019*; *da Silva et al., 2018*; *Dodson et al., 2016*; *Howe and Dombeck, 2016*; *Wang and Tsien, 2011*), and dopaminergic signals have been reported to reflect more general encoding of movement kinematics (*Barter et al., 2015*; *Engelhard et al., 2019*; *Parker et al., 2016*). However, optogenetic activation of dopamine neurons—within physiological range—does not elicit immediate movements (*Coddington and Dudman, 2018*; *Coddington and Dudman, 2019*). We hypothesized that rather than overtly triggering movements, the ongoing activity of nigrostriatal DANs could influence movement initiation over longer timescales by controlling or modulating the moment-by-moment decision of *when* to execute a planned movement.

To test this hypothesis, we trained mice to make a movement (lick) after a self-timed interval following a start-timing cue. The mice learned the timed interval, but, as observed in other species, the exact timing of movement was highly variable from trial-to-trial, spanning seconds. We exploited this inherent

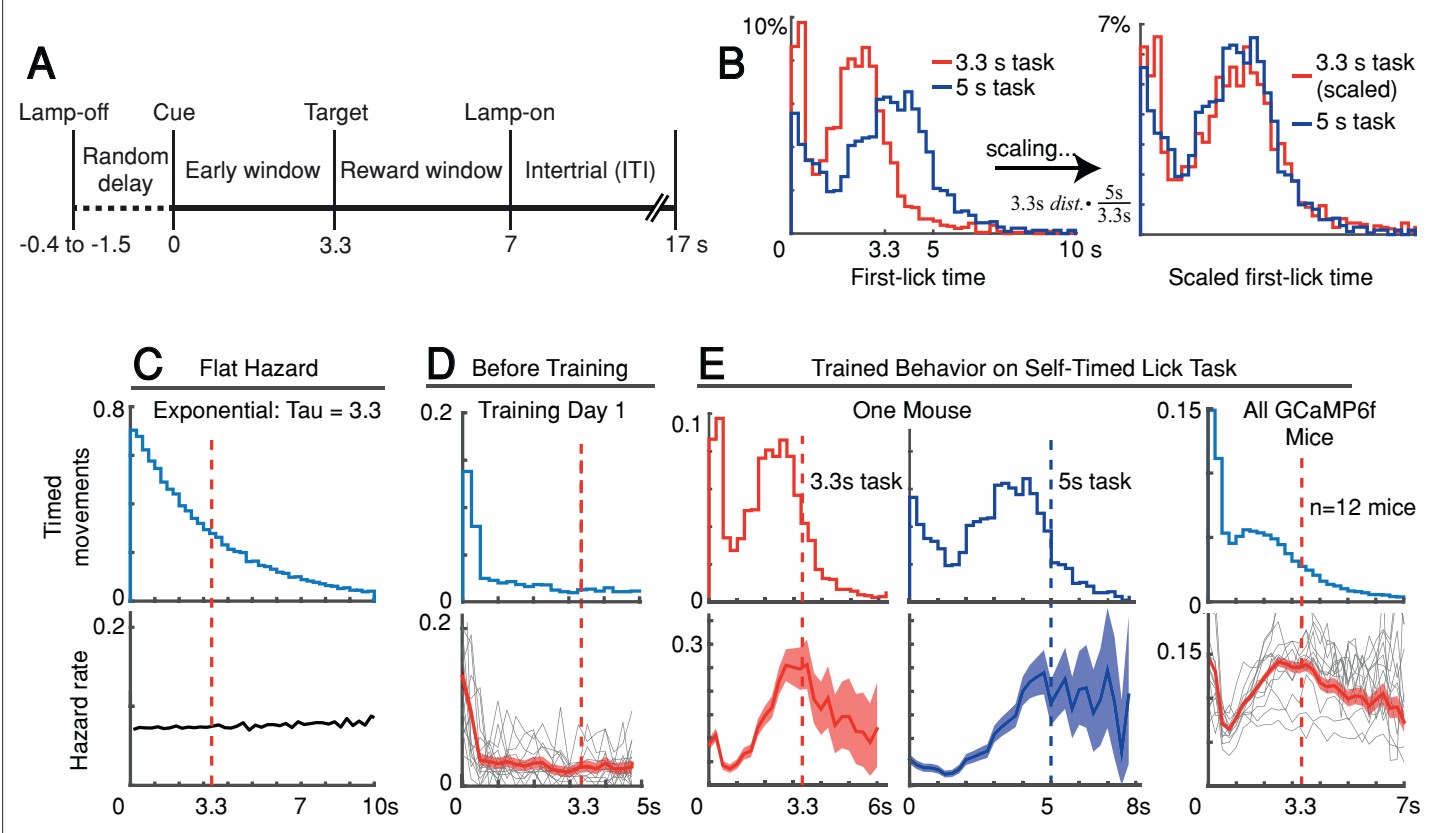

**Figure 1.** Self-timed movement task. (**A**) Task schematic (3.3 s version shown). (**B**) First-lick timing distributions generated by the same mouse exhibit the scalar property of timing (Weber's Law). Red: 3.3 s target time (four sessions); Blue: 5 s target time (four sessions). For all mice, see *Figure 1—figure supplement 1B*. (**C–E**) Hazard-function analysis. Time = 0 is the start-timing cue; dashed vertical lines are target times. (**C**) Uniform instantaneous probability of movement over time is equivalent to a flat hazard rate (bottom) and produces an exponential first-lick timing distribution (top). (**D**) Before Training: First day of exposure to the self-timed movement task. Top: average first-lick timing distribution across mice; bottom: corresponding hazard functions. Gray traces: single session data. Red traces: average among all sessions, with shading indicating 95% confidence interval produced by 10,000x bootstrap procedure. (**E**) Trained Behavior: Hazard functions (bottom) computed from the first-lick timing distributions for the 3.3 s- and 5 s tasks (top) reveal peaks at the target times. Right: average first-lick timing distribution and hazard functions for all 12 GCaMP6f photometry animals. Source data: *Figure 1—source data 1*.

The online version of this article includes the following source data and figure supplement(s) for figure 1:

**Source data 1.** Self-timed movement task behavioral data.

**Figure supplement 1.** Self-timed movement task learning and variations.

**Figure supplement 2.** Fiber optic placement and histology.

variability by examining how moment-to-moment nigrostriatal DAN signals differed when animals decided to move relatively early versus late. We found that dopaminergic signals 'ramped up' during the timing interval, with variable dynamics that were highly predictive of trial-by-trial movement timing, even seconds before the movement occurred. Because reward was delivered at the time of movement, the ramping dopaminergic signals likely related to the animal's expectation of when reward would be available in response to movement. Furthermore, optogenetic DAN manipulation during the timing interval produced bidirectional changes in the probability of movement timing, with activation causing a bias toward earlier self-timed movements and suppression causing a bias toward later self-timed movements. These combined observations suggest a novel role for the dopaminergic system in the timing of movement initiation, wherein slowly evolving dopaminergic signals, likely driven by reward expectation, can modulate the moment-to-moment probability of whether a reward-related movement will occur.

# Results

We trained head-fixed mice to make self-timed movements to receive juice rewards (*Figure 1A*). Animals received an audio/visual start-timing cue and then had to decide when to first-lick in the absence of further cues. Animals only received juice if they waited a proscribed interval following the cue before making their first-lick (>3.3 s in most experiments). As expected from previous studies, the distribution of first-lick timing was broadly distributed over several seconds, and exhibited the canonical scalar property of timing, as described by Weber's Law (*Figure 1B* and *Figure 1—figure supplement 1A-B*; *Gallistel and Gibbon, 2000*). We note this variability in timing was not imposed on the animal by training it to reproduce a variety of target intervals (e.g., 2 *vs.* 5 s), but is rather a natural consequence of timing behavior, even for a single target interval.

Our main objective was to exploit the inherent variability in self-timed behavior to examine how differences in neural activity might relate to variability in movement timing. Nonetheless, the trained animals well-understood the timing contingencies of the task. In self-timed movement tasks in which a *single* movement is used to assess timing, the distributions of movement times (in both rodents and monkeys) tend to anticipate the target interval, even at the expense of reward on many trials (*Eckard and Kyonka, 2018*; *Kirshenbaum et al., 2008*; *Lee and Assad, 2003*). In these paradigms, however, once a movement occurs, it removes future opportunities to move, which creates premature 'bias' in the raw timing distributions (*Anger, 1956*). To correct this bias, movement times must be normalized by the (ever-diminishing) number of opportunities to move at each timepoint (*Jaldow et al., 1990*). This yields the hazard function (the conditional probability of movement given that movement has not already occurred, as a function of time), which is equivalent to the instantaneous probability of movement. For example, on the first day of training, our animals displayed fairly flat hazard functions, indicating a uniform instantaneous probability of movement over time—that is, the animals did not yet understand the timing contingency (*Figure 1C–D*). However, after training, the hazard function for our animals peaked near the target time (either 3.3 or 5 s), suggesting an accurate latent timing process reflected in the instantaneous movement probability (*Figure 1E*). Mice trained on a variant of the self-timed movement task without lamp-off/on events showed no systematic differences in their timing distributions (*Figure 1—figure supplement 1C*), suggesting that the mice referenced their timing to the start-timing cue rather than the lamp-off event.

When mice were fully trained, we employed fiber photometry to record the activity of genetically-defined DANs expressing the calcium-sensitive fluorophore GCaMP6f (12 mice, substantia nigra pars compacta [SNc]; *Figure 1—figure supplement 2*). We controlled for mechanical/optical artifacts by simultaneously recording fluorescence modulation of a co-expressed, calcium-insensitive fluorophore, tdTomato. We also recorded bodily movements with neck-muscle EMG, high-speed video, and a back-mounted accelerometer.

## DAN signals ramp up slowly between the start-timing cue and self-timed movement

DAN GCaMP6f fluorescence typically exhibited brief transients following cue onset and immediately before first-lick onset (*Figure 2A*), as observed in previous studies (*Coddington and Dudman, 2018*; *da Silva et al., 2018*; *Dodson et al., 2016*; *Howe and Dombeck, 2016*; *Schultz et al., 1997*). However, during the timed interval, we observed slow 'ramping up' of fluorescence over seconds, with a minimum after the cue-aligned transient and maximum just before the lick-related transient. The relatively fast intrinsic decay kinetics of GCaMP6f ($t_{1/2}$ <100 ms at 37°; *Helassa et al., 2016*) should not produce appreciable signal integration over the seconds-long timescales of the ramps we observed.

We asked whether this ramping differed between trials in which the animal moved relatively early or late. Strikingly, when we averaged signals pooled by movement time, we observed systematic differences in the steepness of ramping that were highly predictive of movement timing (*Figure 2B–C*). Trials with early first-licks exhibited steep ramping, whereas trials with later first-licks started from lower fluorescence levels and rose more slowly toward the time of movement. The fluorescence ramps terminated at nearly the same amplitude, regardless of the movement time. Ramping dynamics were not evident in control tdTomato signals (*Figure 2C*), indicating that the ramping in the GCaMP6f signals was not an optical artifact. The quantitative relationship between GCaMP6f dynamics and movement time will be addressed in a subsequent section of this paper.

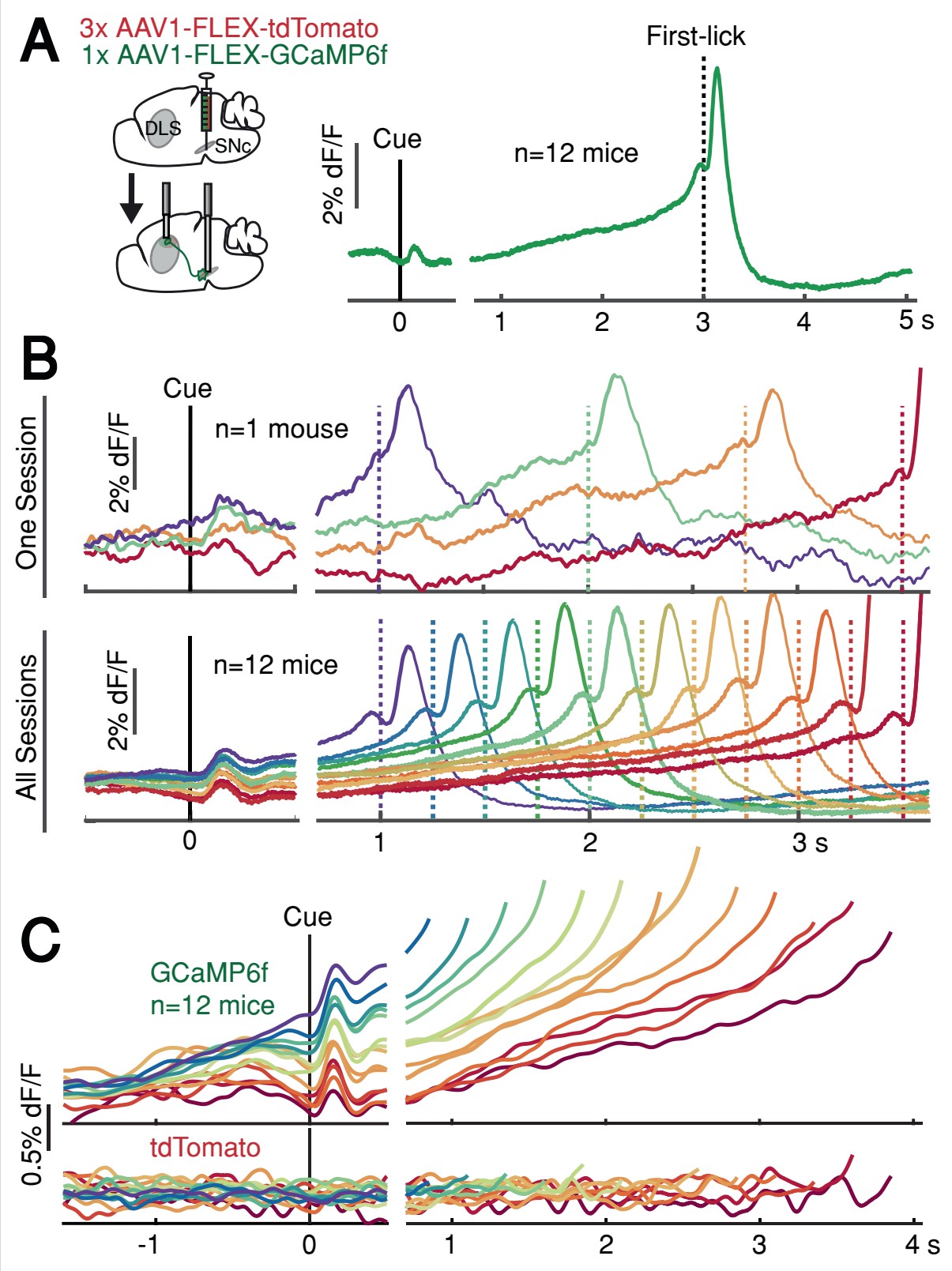

**Figure 2.** SNc DAN signals preceding self-timed movement. (**A**) Left: surgical strategy for GCaMP6f/tdTomato fiber photometry. Right: average SNc DAN GCaMP6f response for first-licks between 3 and 3.25 s (12 mice). Data aligned separately to both cue-onset (left) and first-lick (right), with the break in the time axis indicating the change in plot alignment. (**B**) Average SNc DAN GCaMP6f responses for different first-lick times (indicated by dashed

*Figure 2 continued on next page*

Figure 2 continued

vertical lines). (**C**) Comparison of average DAN GCaMP6f and tdTomato responses on expanded vertical scale. Traces plotted up to 150 ms before first-lick. See also *Figure 2—figure supplements 1–3*. *Figure 2—source data 1*.

The online version of this article includes the following source data and figure supplement(s) for figure 2:

**Source data 1.** SNc DAN signals during self-timing.

**Source data 2.** Baseline SNc DAN signals.

**Source data 3.** df/F methods.

**Source data 4.** DAN signals recorded at SNc, DLS and VTA.

**Figure supplement 1.** Baseline correlation of dopaminergic signal with first-lick time is not dependent on the duration of the lamp-off interval.

**Figure supplement 2.** dF/F method validation.

**Figure supplement 3.** Average photometry signals, pooled every 250 ms by first-lick time, spanning 0.5 s (purple) to 7 s (red).

## Higher pre-cue DAN signals are correlated with earlier self-timed movements

In addition to ramping dynamics, average DAN GCaMP6f signals were correlated with first-lick timing even before cue-onset, with higher baseline fluorescence predicting earlier first-licks (*Figure 2B–C*). This correlation began before the lamp-off event (the 2 s 'Baseline' period before lamp-off; Pearson's $r = -0.63$ (95% CI=[-0.92,–0.14]), n = 12 mice) and grew stronger during the 'Lamp-Off Interval' between lamp-off and the cue (Pearson's $r = -0.89$ (95% CI=[-0.98,–0.68]), n = 12 mice; *Figure 2—figure supplement 1A-B*). This correlation was independent of the duration of the lamp-off interval (*Figure 2—figure supplement 1C*). Because dF/F correction methods can potentially distort baseline measurements, we rigorously tested and validated three different dF/F methods, and we also repeated analyses with raw fluorescence values compared between pairs of sequential trials with different movement times (*Figure 2—figure supplement 2*; see Materials and methods). All reported results, including the systematic baseline differences, were robust to dF/F correction.

In principle, the amplitude of the baseline signal on a given trial *n* could be related to the animal's behavior during the baseline interval or the outcome of the previous trial. To test this, we performed four-way ANOVA to compare the main effects of the following factors on the pre-cue signal (averaged for each trial between lamp-off and the start-timing cue, the 'lamp-off interval' (LOI), n = 12 mice): (1) presence or absence of spontaneous licking during the LOI; (2) outcome of the previous trial (rewarded or unrewarded); (3) upcoming movement time on trial *n* (categorized as <3.3 s or >3.3 s to provide a simple binary proxy for movement time); and (4) session number (to account for signal variability across animals and daily sessions). Although the effects of LOI-licking and previous trial outcome were statistically significant (F(1,18282) = 10.7, p = 0.008, $\eta_p^2=5.9 \cdot 10^{-4}$ and F(1,18282) = 281.2, p = 7.5·10$^{-47}$, $\eta_p^2$=0.015, respectively), the upcoming movement time had an independent, statistically significant effect (F(1,18282) = 63.4, p = 5.9·10$^{-6}$, $\eta_p^2$=0.0035). This raises the possibility of an additional source of variance in baseline dopaminergic activity that is independent from previous trial events, but potentially influences the upcoming movement time on that trial.

## Ramping dynamics in other dopaminergic areas and striatal dopamine release

We found similar ramping dynamics in SNc DAN axon terminals in the dorsolateral striatum (DLS; *Figure 2—figure supplement 3A-B*) at a location involved in goal-directed licking behavior (*Sippy et al., 2015*). Ramping was also present in GCaMP6f-expressing DAN cell bodies in the ventral tegmental area (VTA, *Figure 2—figure supplement 3C*), reminiscent of mesolimbic ramping signals described in goal-oriented navigation tasks (*Howe et al., 2013*; *Kim et al., 2019*).

To determine if these movement timing-related signals are available to downstream targets that may be involved in movement initiation, we monitored dopamine release in the DLS with two complementary florescent dopamine sensors (dLight1.1 and DA$_{2m}$) expressed broadly in striatal cells (*Figure 3* and *Figure 2—figure supplement 3D-E*). The decay kinetics of the two extracellular dopamine sensors differ somewhat (*Patriarchi et al., 2018*; *Sun et al., 2020*), which we confirmed (dLight1.1 $t_{1/2}$~75 ms, DA$_{2m}$ $t_{1/2}$~125 ms; *Figure 3—figure supplement 1*), yet both revealed similar timing-related ramping dynamics on average (*Figure 3 inset*). These combined data argue that the

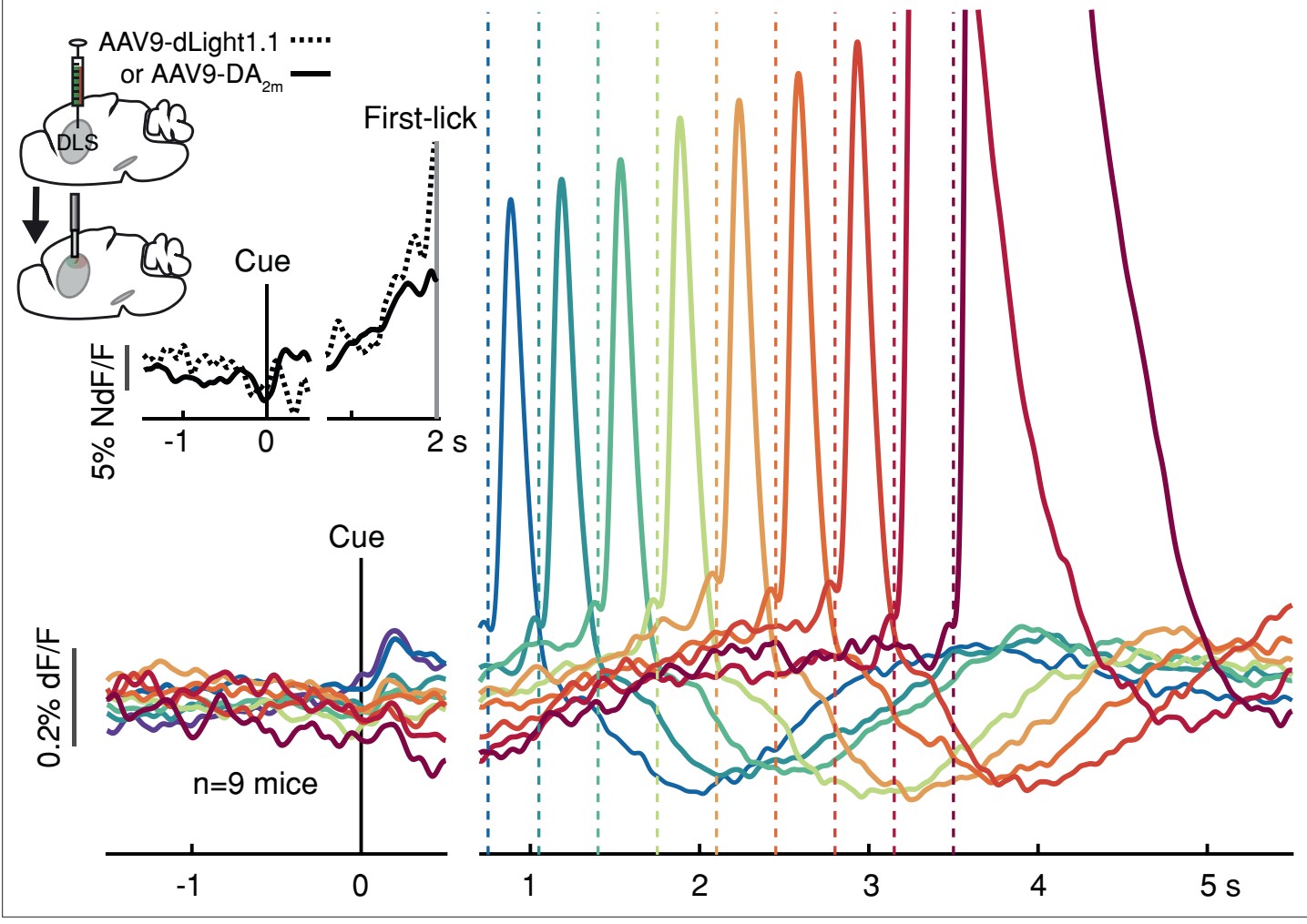

**Figure 3.** Striatal dopamine release during the self-timed movement task. Photometry signals averaged together from DA$_{2m}$ signals (n = 4 mice) and dLight1.1 signals (n = 5 mice) recorded in DLS. Axis break and plot alignment as in **Figure 2**. Dashed lines: first-lick times. **Inset, left**: surgical strategy. **Inset, right**: Comparison of dLight1.1 and DA$_{2m}$ dynamics. Expanded vertical scale to show ramping in the average signals for DA$_{2m}$ (solid trace) and dLight1.1 (dashed trace) up until the time of the first-lick (first-lick occurred between 2 and 3 s after the cue for this subset of the data). See also: *Figure 3—figure supplement 1*. *Figure 3—source data 1*.

The online version of this article includes the following source data and figure supplement(s) for figure 3:

**Source data 1.** Striatal dopamine indicator signals.

**Figure supplement 1.** Comparison of dLight1.1 (dashed) and DA$_{2m}$ (solid) kinetics surrounding peak of unrewarded transient (first-lick time: 0.5–3.3 s).

seconds-long dopaminergic ramping signals were not artifacts of sluggish temporal responses of the various fluorescent sensors and were ultimately expressed as ramp-like increases in dopamine release in the striatum.

## First-lick timing-predictive DAN signals are not explained by ongoing body movements

The systematic ramping dynamics and baseline differences were not observed in the tdTomato optical control channel nor in any of the other movement-control channels, at least on average (*Figure 4*), making it unlikely that ramping dynamics resulted from optical artifacts. Nevertheless, because DANs show transient responses to salient cues and movements (*Coddington and Dudman, 2018*; *da Silva et al., 2018*; *Dodson et al., 2016*; *Howe and Dombeck, 2016*; *Schultz et al., 1997*), it is possible that fluorescence signals could reflect the superposition of dopaminergic responses to multiple task events, including the cue, lick, ongoing spurious body movements, and hidden cognitive processes

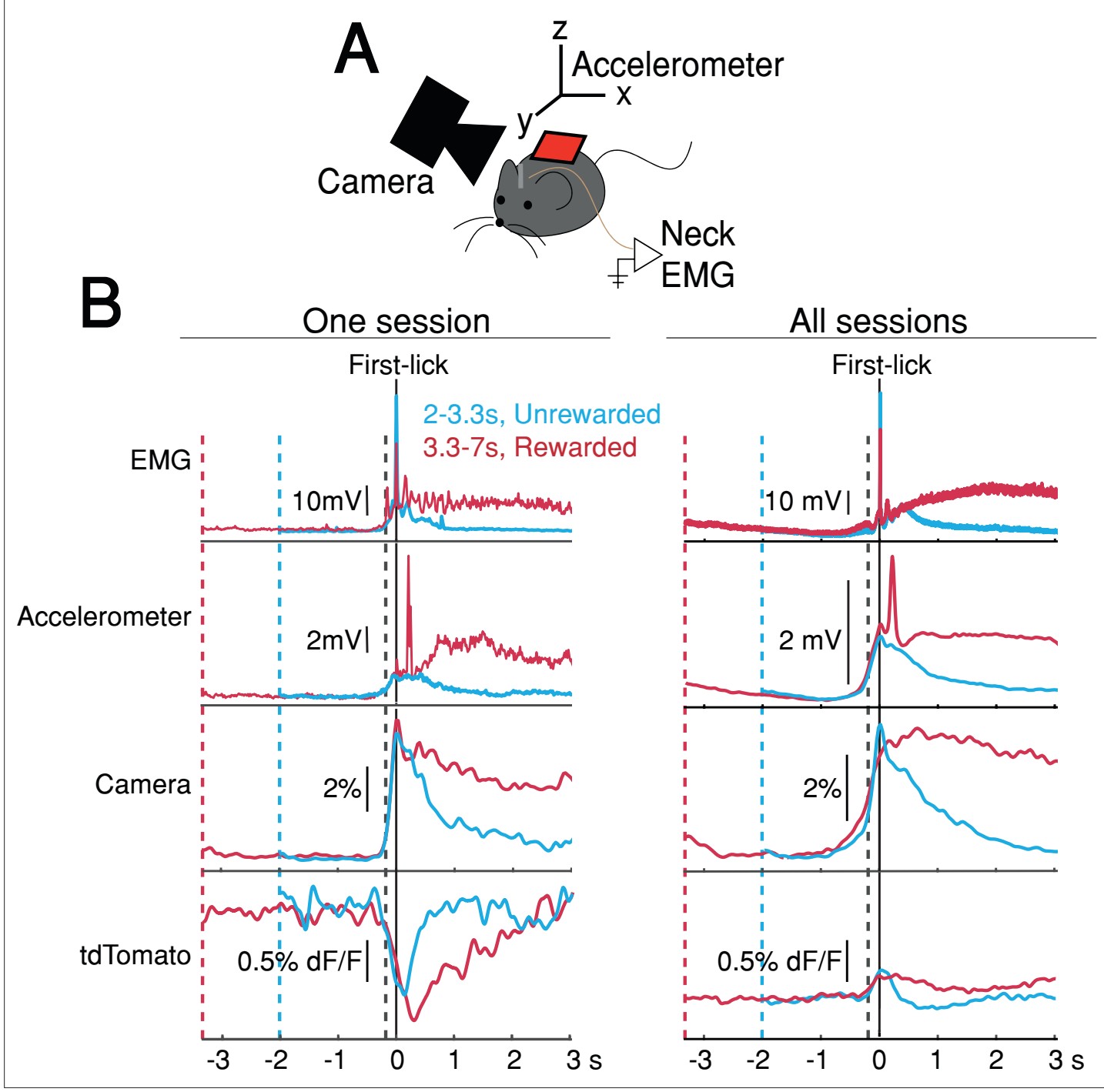

**Figure 4.** Movement controls reliably detected movements, but there were no systematic differences in movement during the timing interval. (**A**) Schematic of movement-control measurements. (**B**) First-lick-aligned average movement signals on rewarded (red) and unrewarded (blue) trials. Pre-lick traces begin at the nearest cue-time (dashed red, dashed blue). Left: one session; Right: all sessions. Dashed grey line: time of earliest-detected movement on most sessions (150ms before first-lick). Average first-lick-aligned tdTomato optical artifacts showed inconsistent excursion directions (up/down), even within the same session; signals for each artifact direction shown in *Figure 4—figure supplement 1*. Source data: *Figure 4—source data 1*.

The online version of this article includes the following source data and figure supplement(s) for figure 4:

**Source data 1.** Movement control signals.

**Figure supplement 1.** Average tdTomato optical artifacts (aligned to first-lick time) showed inconsistent excursion directions even within the same session.

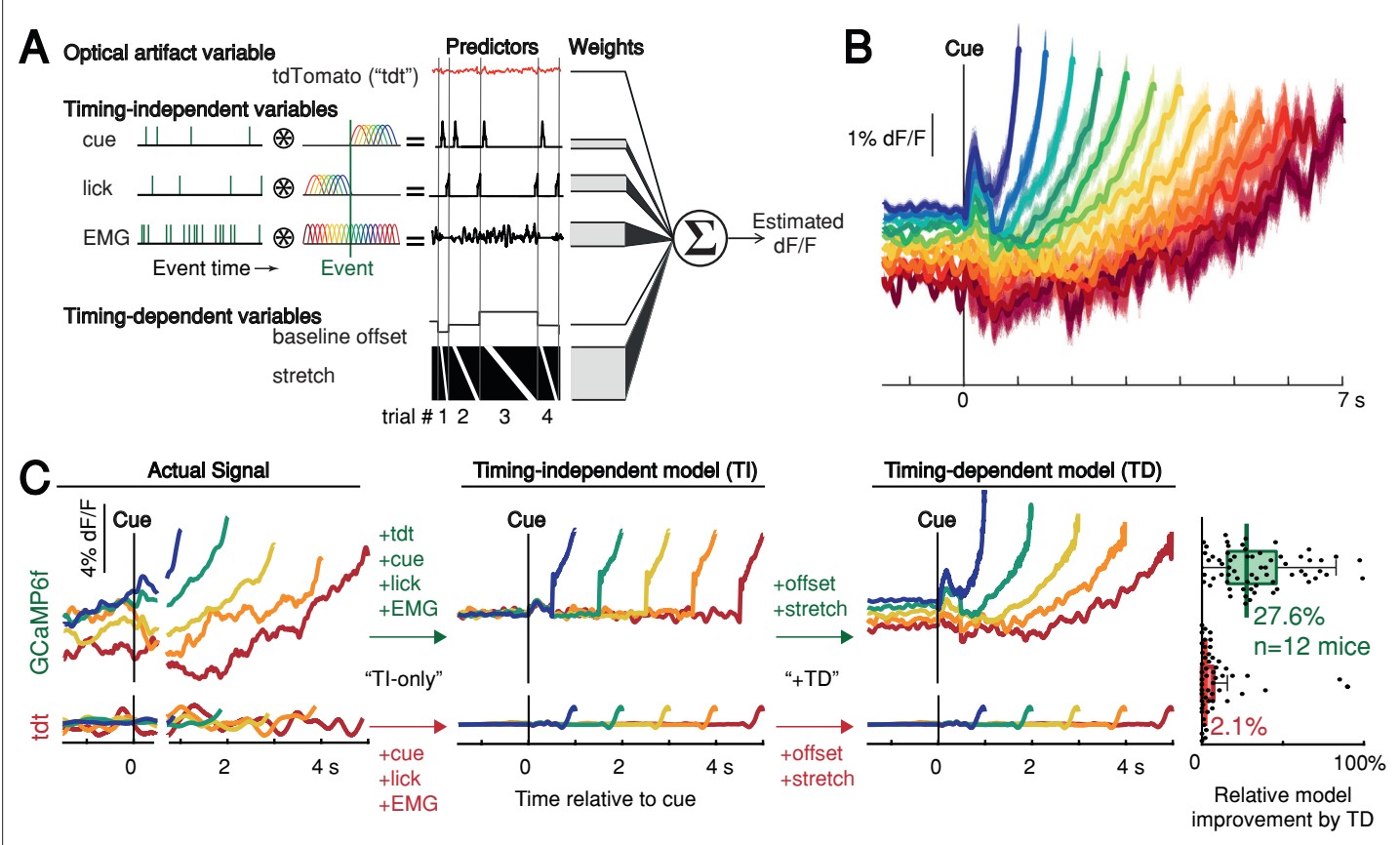

**Figure 5.** Contribution of optical artifacts, task variables and nuisance bodily movements to SNc GCaMP6f signals. (**A**) Nested encoding model comparing the contribution of timing-independent predictors (TI) to the contribution of timing-dependent predictors (TD). (**B**) Predicted dF/F signal for one session plotted up to time of first-lick. Model error simulated 300x (shading). (**C**) Nested encoding model for one session showing the actual recorded signal (1st panel), the timing-independent model (2nd panel), and the full, timing-dependent model with all predictors (3rd panel). Top: GCaMP6f; Bottom: tdTomato (tdt). Right: relative loss improvement by timing-dependent predictors (grey dots: single sessions, line: median, box: lower/upper quartiles, whiskers: 1.5x IQR). See also *Figure 5—figure supplement 1*. Source data: *Figure 5—source data 1*.

The online version of this article includes the following source data and figure supplement(s) for figure 5:

**Source data 1.** DAN signal encoding model.

**Figure supplement 1.** DAN signal encoding model parameterization and model selection.

**Figure supplement 2.** Principal component analysis (PCA) of the ramping interval (0.7 s up to first-lick; relative to cue).

like timing. For example, accelerating spurious movements could, in principle, produce motor-related neural activity that ramps up during the timed interval, perhaps even at different rates on different trials.

We thus derived a nested generalized linear encoding model of single-trial GCaMP6f signals (*Engelhard et al., 2019*; *Park et al., 2014*; *Runyan et al., 2017*), a data-driven, statistical approach designed to isolate and quantify the contributions of task events (timing-independent predictors) from processes predictive of movement timing (timing-dependent predictors; *Figure 5A–B* and *Figure 5— figure supplement 1A-D*). The model robustly detected task-event GCaMP6f kernels locked to cue, lick and EMG/accelerometer events, but these timing-independent predictors alone were insufficient to capture the rich variability of GCaMP6f signals for trials with different first-lick times, especially the timing-dependent ramp-slope and baseline offset (n = 12 mice, *Figure 5C* and *Figure 5—figure supplement 1E-G*). In contrast, two timing-dependent predictors robustly improved the model: (1) a baseline offset with amplitude linearly proportional to first-lick time; and (2) a 'stretch' feature representing percentages of the timed interval (*Figure 5B–C* and *Figure 5—figure supplement 1E*). The baseline offset term fit a baseline level inversely proportional to movement time, and the temporal stretch feature predicted a ramping dynamic from the time of the cue up to the first-lick, whose slope

was inversely proportional to first-lick time. Similar results were obtained for SNc DAN axon terminals in the DLS, VTA DAN cell bodies, and extracellular striatal dopamine release (*Figure 5—figure supplement 1H*).

We note that the stretch feature of this GLM makes no assumptions about the underlying shape of the dopaminergic signal; it only encodes percentages of timing intervals to allow for temporal 'expansion' or 'contraction' to fit whatever shape(s) were present in the data. In particular, the stretch feature cannot produce ramping unless ramping is present in the signal *and* temporally scales with the length of the interval. Because this feature empirically found a ramp (although not constrained to do so), the stretch aspect indicated that the underlying ramping process took place at different rates for trials with different movement times, at least on average.

In contrast to the GCaMP6f model, when the same GLM was applied to the tdTomato control signal, the timing-independent predictors (which could potentially cause optical/mechanical artifacts—the cue onset, first-lick, and EMG/accelerometer signal) improved the model, but timing-dependent predictors did not (*Figure 5C* and *Figure 5—figure supplement 1F-H*). In addition, separate principal component (PC) analysis revealed ramp-like and baseline-offset-like components that explained as

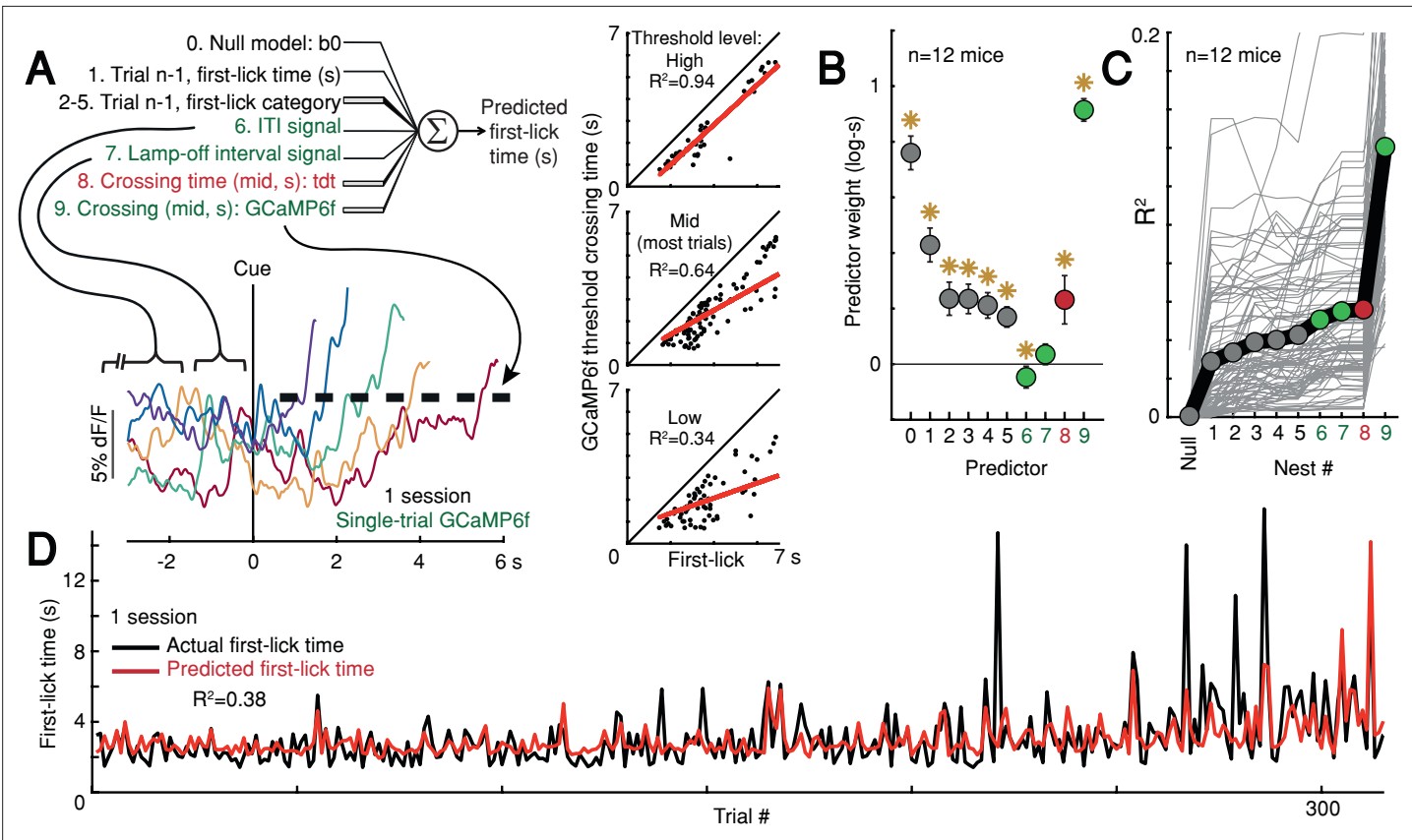

**Figure 6.** Single-trial DAN signals predict first-lick timing. (**A**) Schematic of nested decoding model. Categories for n-1[th] trial predictors: (2) reaction, (3) early, (4) reward, (5) ITI first-lick (see Materials and methods). Bottom: single-trial cue-aligned SNc DAN GCaMP6f signals from one session (six trials shown for clarity). Traces plotted up to first-lick. Right: threshold-crossing model. Low/Mid/High label indicates threshold amplitude. Dots: single trials. (**B**) Model weights. Error bars: 95% CI, *: p<0.05, two-sided t-test. Numbers indicate nesting-order. (**C**) Variance explained by each model nest. Gray lines: single sessions; thick black line: average. For model selection, see *Figure 6—figure supplement 1*. (**D**) Predicted *vs.* actual first-lick time, same session as *6A*. See also *Figure 6—figure supplements 1–4*. Source data: *Figure 6—source data 1*.

The online version of this article includes the following source data and figure supplement(s) for figure 6:

**Source data 1.** Movement time decoding model.

**Figure supplement 1.** Variations of the first-lick time decoding model.

**Figure supplement 2.** Analysis of single-trial dynamics: Hierarchical Bayesian Ramp *vs* Step Modeling.

**Figure supplement 3.** Geometric analysis of single-trial dynamics with Multiple Threshold Modeling.

**Figure supplement 4.** Assessing single-trial dynamics.

much as 93% of the variance in DAN signals during the timing interval (mean: 66%, range: 16–93%), but similar PCs were not present when tdTomato control signals were analyzed with PCA (mean variance explained: 4%, range: 1.6–15%, *Figure 5—figure supplement 2*).

## Single-trial DAN ramping and baseline signals predict movement timing

Given that ramping and baseline-offset signals were not explained by nuisance movements or optical artifacts, we asked whether DAN GCaMP6f fluorescence could predict first-lick timing on single trials. Using a simple threshold-crossing decoding model (*Maimon and Assad, 2006*), we found that single-trial GCaMP6f signals were predictive of first-lick time even for low thresholds intersecting the 'base' of the ramp, with the predictive value of the model progressively improving for higher thresholds (n = 12 mice: mean $R^2$ low = 0.54, mid = 0.71, high = 0.82 (95% CI: low=[0.44,0.64], mid=[0.68,0.75], high=[0.76,0.87]); analysis for one mouse shown in *Figure 6A*). We will return to this observation in more detail in the upcoming section on single-trial dynamics.

To more thoroughly determine the independent, additional predictive power of DAN baseline and ramping signals over other task variables (e.g., previous trial first-lick time and reward outcome, etc.), we derived a nested decoding model for first-lick time (*Figure 6A*). In this model, the pre-cue 'baseline' was divided into two components: the pre-lamp-off intertrial interval signal ('ITI') and the lamp-off to cue interval signal ('LOI'). All predictors contributed to the predictive power of the model. However, even when we accounted for the contributions of prior trial history, tdTomato artifacts and baseline GCaMP6f signals, GCaMP6f threshold-crossing time robustly dominated the model and absorbed much of the variance explained by baseline dopaminergic signals, alone explaining 10% of the variance in first-lick time on average (range: 1–27%, *Figure 6B–D*). Alternate formulations of the decoding model produced similar results (*Figure 6—figure supplement 1*).

## Characterizing single-trial dopaminergic dynamics

Although the threshold-crossing analysis made no assumptions about the underlying dynamics of the GCaMP6f signals on single-trials, in principle, ramping dynamics in *averaged* neural signals could be produced from individual trials with a single, discrete 'step' occurring at different times on different trials. Ramping has long been observed in averaged neural signals recorded during perceptual decision tasks in monkeys, and there has been considerable debate over whether single-trial responses in these experiments are better classified as 'ramps' or a single 'step' (*Latimer et al., 2015*; *Latimer et al., 2016*; *Shadlen et al., 2016*; *Zoltowski et al., 2019*; *Zylberberg and Shadlen, 2016*). It has even been suggested that different sampling distributions can produce opposite model classifications in ground-truth synthetic datasets (*Chandrasekaran et al., 2018*).

We attempted to classify single-trial dynamics as a discrete stepping or ramping process with hierarchical Bayesian models implemented in probabilistic programs (*Figure 6—figure supplement 2A-B*). However, like the perceptual decision-making studies, we also found ambiguous results, with about half of single trials best classified by a linear ramp and half best classified by a discrete step dynamic (*Figure 6—figure supplement 2C*). Nonetheless, three separate lines of evidence suggest that single trials are better characterized by slowly evolving ramps:

First, the relationship of threshold-crossing time to first-lick time is different for the step *vs*. ramp models when different threshold levels are sampled (*Maimon and Assad, 2006*), as schematized in *Figure 6—figure supplement 3A*: Increasing slope of this relationship is consistent with ramps on single trials, but not with a discrete step, which would be expected to have the same threshold-crossing time regardless of threshold level (*Figure 6—figure supplement 3B*). We found that the slope of this relationship increased markedly as the threshold level was increased, consistent with the ramp model (n = 12 mice: mean slope low = 0.46, mid = 0.7, high = 0.82 (95% CI: low=[0.37,0.54], mid=[0.66,0.73], high=[0.74,0.88]), *Figure 6—figure supplement 3C*).

Second, if single trials involve a step change occurring at different times from trial-to-trial then aligning trials on that step should produce a clear step on average, rather than a ramp (*Latimer et al., 2015*). We thus aligned single-trial GCaMP6f signals according to that optimal step position determined from a Bayesian step model fit for each trial and then averaged the step-aligned signals across trials. The averaged signals did not resemble a step function, but rather yielded a sharp transient superimposed on a 'background' ramping signal (*Figure 6—figure supplement 4A*). Step-aligned tdTomato and EMG averages showed a small inflection at the time of the step, but neither signal

showed background ramping. This suggests that the detected 'steps' in the GCaMP6f signals were likely transient movement artifacts superimposed on the slower ramping dynamic rather than bona fide steps.

Third, the ideal step model holds that the step occurs at different times from trial-to-trial, producing a ramping signal when trials are averaged together. In this view, the trial-by-trial variance of the signal should be maximal at the time at which 50% of the steps have occurred among all trials, and the variance should be minimal at the beginning and end of the interval (when no steps or all steps have occurred, respectively). We thus derived the optimal step time for each trial using the Bayesian step model, and we then calculated variance as a function of time within pools of trials with similar movement times. The signal variance showed a monotonic downward trend during the timed interval, with a minimum variance at the time of movement rather than at the point at which 50% of steps had occurred among trials, inconsistent with the discrete step model (*Figure 6—figure supplement 4B*).

Taken together, we did not find evidence for a discrete step dynamic on single trials; on the contrary, our observations concord with slow ramping dynamics on single trials. Regardless, our GLM movement-time decoding approaches in *Figure 6* did not make any assumptions about underlying single-trial dynamics.

## Moment-to-moment DAN activity causally controls movement timing on single trials

Because dopaminergic ramping signals robustly predicted first-lick timing and were apparently transmitted via dopamine release to downstream striatal neurons, ramping DAN activity may causally determine movement timing. However, because the animals could expect reward within a few hundred milliseconds of the first-lick, it is also possible that the dopaminergic ramps could instead serve as a 'passive' monitor of reward expectation without influencing movement initiation. To distinguish these possibilities, we optogenetically activated or inhibited DANs (in separate experiments) on 30% of randomly interleaved trials (*Figure 7A* and *Figure 7—figure supplement 1*). For activation experiments, we chose light levels that elevated DAN activity within the physiological range observed in our self-timed movement task, as assayed by simultaneous photometry in the DLS with a fluorescent sensor of released dopamine (dLight1.1, *Figure 7—figure supplement 2*). DAN activation significantly early-shifted the distribution of self-timed movements on stimulated trials compared to unstimulated trials (12 mice; 2-sample Kolmogorov-Smirnov (KS) Test, D = 0.078 (95% CI: [0.067,0.093]), p = 2.8·10$^{-26}$), whereas inhibition produced significant late-shifting compared to unstimulated trials (4 mice; two-sample KS Test, D = 0.051 (95% CI: [0.034,0.077]), p = 3.1·10$^{-4}$; *Figure 7B* and *Figure 7—figure supplement 3A*). Stimulation of mice expressing no opsin produced no consistent effect on timing (5 mice; two-sample KS Test, D = 0.017 (95% CI: [0.015,0.040]), p = 0.62). The direction of these effects was consistent across all animals tested in each category (*Figure 7C*). Complementary analysis methods revealed consistent effects (bootstrapped difference in median first-lick times between categories: Δ(activation - no-opsin) = –0.22 s (95% CI=[–0.32 s,–0.12 s]), Δ(inhibition – no-opsin) = +0.19 s (95% CI=[+0.09 s,+0.30 s]), *Figure 7C–D*; bootstrapped comparison of difference in area under the cdf curves: Δ(activation – no-opsin) = –0.31 dAUC (95% CI=[–0.47 dAUC,–0.15 dAUC]), Δ(inhibition – no-opsin) = +0.23 dAUC (95% CI=[+0.08 dAUC,+0.37 dAUC]), *Figure 7—figure supplement 3B*; bootstrapped difference in mean first-lick times between categories: Δ(activation – no-opsin) = –0.34 s (95% CI=[–0.49 s,–0.19 s]), Δ(inhibition – no-opsin) = +0.24 s (95% CI=[+0.09 s,+0.39 s]), *Figure 7—figure supplement 3C*). Similar effects were obtained with activation of SNc DAN axon terminals in the DLS (2 mice, *Figure 7—figure supplement 3A-B*). Because these exogenous manipulations of DAN activity modulated movement timing on the same trial as the stimulation/inhibition, this suggests that the *endogenous* dopaminergic ramping we observed during the self-timed movement task likewise affected movement initiation in real time, rather than serving solely as a passive monitor of reward expectation.

Recent studies have shown that physiological ranges of optogenetic DAN activation (as assayed by simultaneous recordings from DANs) fail to elicit overt movements (*Coddington and Dudman, 2018*). We likewise found that optogenetic DAN activation did not elicit immediate licking outside the context of the task (*Figure 7—figure supplement 4A*). Additionally, optogenetic DAN inhibition did not reduce the rate of spontaneous licking outside the context of the task (*Figure 7—figure supplement 4B*). In both cases, we used the same light levels that had elicited the robust shifts in timing

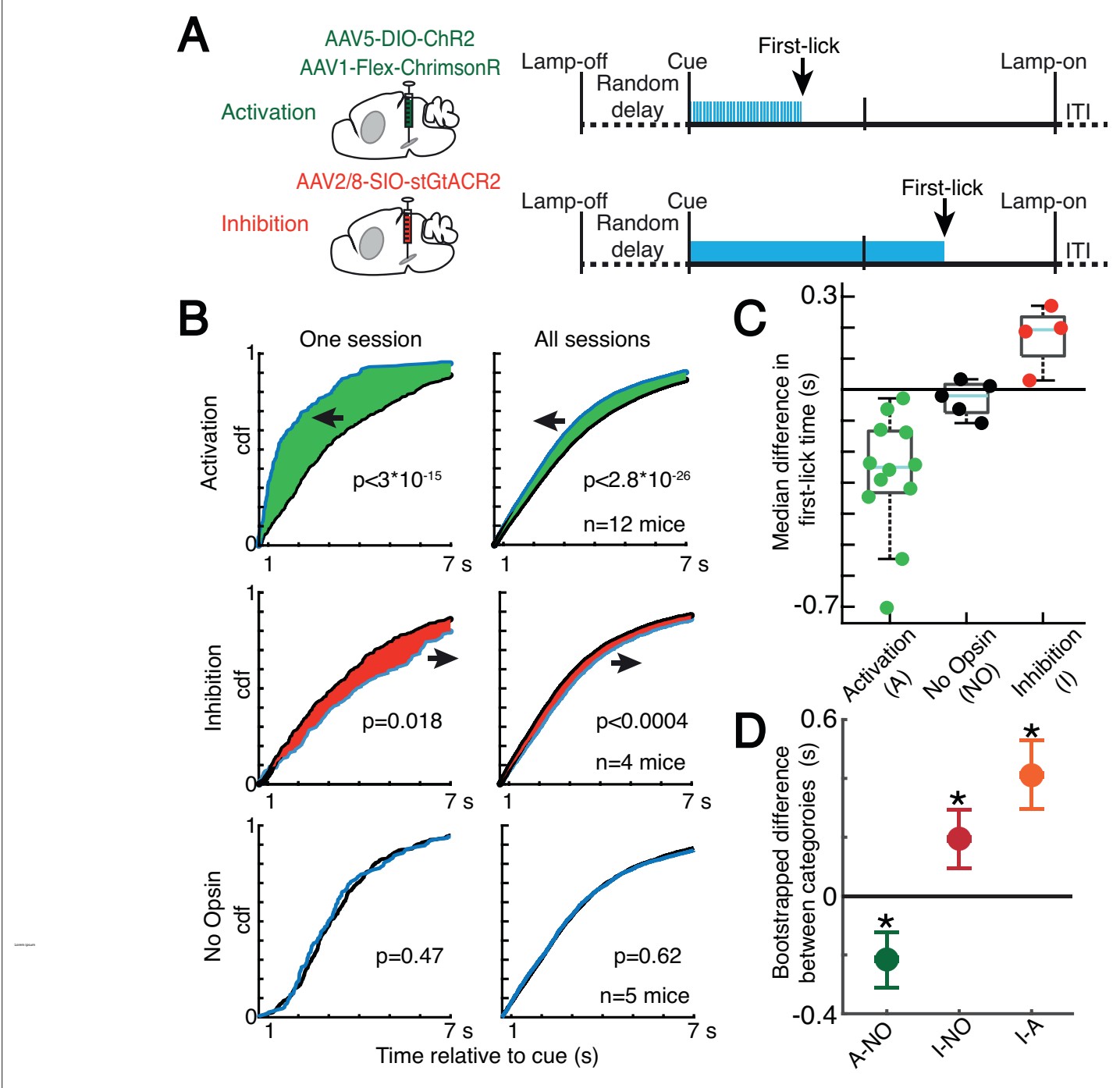

**Figure 7.** Optogenetic DAN manipulation systematically and bidirectionally shifts the timing of self-timed movements. (**A**) Strategy for optogenetic DAN activation or inhibition. Mice were stimulated from cue-onset until first-lick or 7 s. (**B**) Empirical continuous probability distribution functions (cdf) of first-lick times for stimulated (blue line) versus unstimulated (grey line) trials. Arrow and shading show direction of effect. p-Values calculated by Kolmogorov-Smirnov test (for other metrics, see *Figure 7—figure supplements 1 and 3*). (**C**) Median 1,000,000x bootstrapped difference in first-lick time, stimulated-minus-unstimulated trials. Box: upper/lower quartile; line: median; whiskers: 1.5x IQR; dots: single mouse. (**D**) Comparison of median first-lick time difference across all sessions. Error bars: 95% confidence interval (*: p<0.05, 1,000,000x bootstrapped median difference in first-lick time between sessions of different stimulation categories). See also *Figure 7—figure supplements 1–4*. Source data: *Figure 7—source data 1*.

The online version of this article includes the following source data and figure supplement(s) for figure 7:

**Source data 1.** Optogenetic manipulation of SNc DANs.

**Figure supplement 1.** Variations on measurements of optogenetic effects.

*Figure 7 continued on next page*

*Figure 7 continued*

**Figure supplement 2.** Light-power calibration for optogenetic activation of DANs.

**Figure supplement 3.** Quantification of optogenetic effects with additional metrics.

**Figure supplement 4.** Optogenetic DAN stimulation does not cause or prevent licking.

behavior during the self-timed movement task. In other control experiments, we purposefully drove neurons into non-physiological activity regimes during the task by applying higher activation light levels. Over-stimulation caused large, immediate, sustained increases in DLS dopamine (*Figure 7—figure supplement 2*), comparable in amplitude to the typical reward-related dopamine transients on interleaved, unstimulated trials. These non-physiological manipulations resulted in rapid, nonpurposive body movements and disrupted performance of the task. Together, these results suggest that the optogenetic effects on timing in *Figure 7* did not result from direct, immediate triggering or suppression of movement, nor from non-physiological dopamine release due to over-stimulation.

## Linking endogenous DAN signals to the moment-to-moment probability of movement initiation

Optogenetic manipulations of DAN activity in the physiological range appeared to modulate the *probability* of initiating the pre-potent, self-timed movement. Given that endogenous DAN signals increased during the timing interval of the self-timed movement task, we reasoned that the probability of movement should likewise increase over the course of the timed interval. We thus derived a nested probabilistic movement-state decoding model to explore the link between DAN signals and movement propensity (*Figure 8A*). We applied a GLM based on logistic regression, in which we classified each moment of time as either a non-movement (0) or movement (1) state (*Figure 8A–B*), and we examined how well various parameters could predict the probability of transitioning from the non-movement state to the movement state. Unlike the decoding model in *Figure 6*, which considers a single threshold-crossing time, the probabilistic approach takes into account continuous DAN signals. Initial model selection included previous trial history (movement time and reward outcome history) in addition to the DAN GCaMP6f signal, but Bayesian Information Criterion (BIC) analysis indicated that the instantaneous GCaMP6f signal alone was a robustly significant predictor of movement state, whereas previous trial outcomes were insignificant contributors and did not further improve the model (*Figure 8—figure supplement 1*). We thus only considered the DAN GCaMP6f signal as a predictor in subsequent analyses.

The continuous DAN GCaMP6f signal was indeed predictive of current movement state at any time *t*, and it served as a significant predictor of movement state up to at least 2 s in the past (*Figure 8C*). However, the signals became progressively more predictive of the current movement state as time approached *t*. That is, the dopaminergic signal level closer to time *t* tended to absorb the behavioral variance explained by more distant, previous signal levels (*Figure 8C*), reminiscent of how threshold crossing time absorbed the variance explained by the baseline dopaminergic signal in the movement-timing decoding model (*Figure 6B–C*). This observation is consistent with a diffusion-like ramping process on single trials, in which the most recent measurement gives the best estimate of whether there will be a transition to the movement state (but is difficult to reconcile with a discrete step process on single trials, consistent with the results in *Figure 6—figure supplements 3–4*).

We applied the fitted instantaneous probabilities of transitioning to the movement state to derive a fitted hazard function for each behavioral session (*Figure 8D*). The DAN GCaMP6f signals were remarkably predictive of the hazard function, both for individual sessions and on average, explaining 65% of the variance on average (n = 12 mice). Conversely, when the model was fit on the same data in which the timepoint identifiers were shuffled, this predictive power was essentially abolished, explaining only 5% of the variance on average (*Figure 8E*).

Together, these results demonstrate that slowly evolving dopaminergic signals are predictive of the moment-to-moment probability of movement initiation. When combined with the optogenetics results, they argue that dopaminergic signals causally modulate the moment-to-moment probability of the pre-potent movement. In this view, trial-by-trial variability in the DAN signal gives rise to trial-by-trial differences in movement timing in the self-timed movement task.

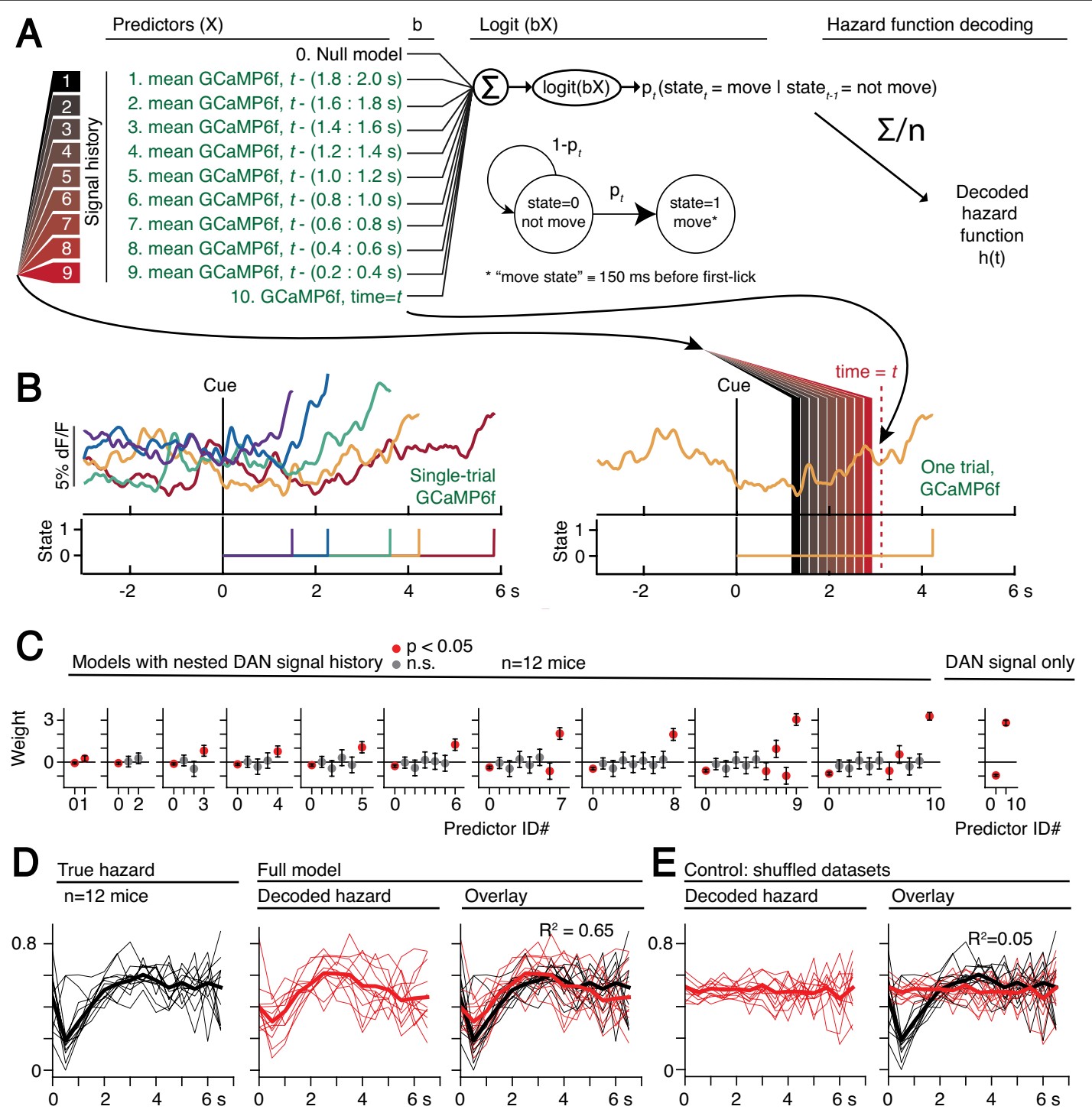

**Figure 8.** Single-trial dynamic dopaminergic signals predict the moment-to-moment probability of movement initiation. (**A**) Probabilistic movement-state model schematic. (**B**) Single-trial DAN GCaMP6f signals at SNc from one session. First-lick time truncated 150 ms before movement detection to exclude peri-movement signals. Bottom: Movement states for the trials shown as a function of time. Diagram on the right schematizes the model predictors relative to an example time = t on a single trial. (**C**) Nested model fitted coefficients. (**D**) Decoded hazard functions from full model (with all 10 predictors). Thick line: mean. n = 12 mice. (**E**) Hazard function fitting with shuffled datasets abolished the predictive power of the model (same 12 mice). See also *Figure 8—figure supplements 1–2*. *Figure 8—source data 1*.

The online version of this article includes the following source data and figure supplement(s) for figure 8:

**Source data 1.** Movement state decoding model.

*Figure 8 continued on next page*

*Figure 8 continued*

**Figure supplement 1.** Probabilistic movement time decoding model: model selection.

**Figure supplement 2.** Average Intertial Interval (ITI) GCaMP6f signals aligned to most recent previous lick-time.

## Discussion

We made two main findings. First, both baseline and slowly ramping DAN signals were predictive of the timing of the first-lick. Second, optogenetic modulation of DANs affected the timing of movements on the *concurrent trial*, suggesting that DANs can play a 'real time' role in behavior. These observations raise two (presumably separable) questions of interpretation: (1) what is the mechanistic origin of ramping DAN signals in the self-timed movement task, and (2) how do DAN signals affect self-timed movements in real time?

### The origin of ramping DAN signals

A number of studies have reported short-latency (<500 ms) modulations in DAN activity following reward-predicting sensory cues and immediately preceding movements (*Coddington and Dudman, 2018*; *da Silva et al., 2018*; *Dodson et al., 2016*; *Howe and Dombeck, 2016*; *Schultz et al., 1997*), similar to the sensory- and motor-related transients we observed within ~500 ms of the cue and first-lick. However, the ramping DAN signals we observed during self-timing were markedly slower, unfolding over *seconds* and preceding the first-lick by as long as 10 s. Previous studies have reported similarly slow ramping dopaminergic signals in other behavioral contexts, including goal-directed navigation toward rewarded targets (*Howe et al., 2013*), multi-step tasks leading to reward (*Hamid et al., 2016*; *Howard et al., 2017*; *Mohebi et al., 2019*), and passive observation of dynamic visual cues indicating proximity to reward (*Kim et al., 2019*). A common feature in these experiments and our self-timed movement task is that trials culminated in the animal's receiving reward. Thus, parsimony suggests that dopaminergic ramping could reflect reward expectation. However, dopaminergic ramping is generally *absent* in Pavlovian paradigms, in which animals learn to expect passive reward delivery at a fixed time following a conditioned stimulus (*Menegas et al., 2015*; *Tian et al., 2016*; *Schultz et al., 1997*; *Starkweather et al., 2017*). One exception is a report of ramping activity in monkey DANs during a Pavlovian paradigm with reward uncertainty (*Fiorillo et al., 2003*); however, ramping was not subsequently reproduced under similar conditions, either in monkeys (*Fiorillo, 2011*; *Matsumoto and Hikosaka, 2009*; *Tobler et al., 2005*) or rodents (*Hart et al., 2015*; *Tian and Uchida, 2015*). Thus, while dopaminergic ramping is likely related to reward expectation, the preponderance of evidence suggests that reward expectation *alone* is insufficient to cause DAN ramping.

To reconcile these disparate findings, Gershman and colleagues proposed a formal model in which dopaminergic ramping encodes reward expectation in the form of an 'ongoing' reward-prediction error (RPE) that arises from resolving uncertainty of one's position in the value landscape (i.e., one's spatial-temporal distance to reward delivery/omission). For example, uncertainty is resolved if animals are provided visuospatial cues indicating proximity to reward (*Howe et al., 2013*; *Kim et al., 2019*). In contrast, because animals can only imprecisely estimate the passage of time, the animal is uncertain of when reward will be delivered/omitted in Pavlovian tasks. The RPE model holds that this temporal uncertainty flattens the Pavlovian value landscape, thereby flattening dopaminergic ramping to the degree that it is obscured (*Gershman, 2014*; *Kim et al., 2019*; *Mikhael and Gershman, 2019*; *Mikhael et al., 2019*). Although both our task and Pavlovian tasks involve timing, the key difference may be that the animal actively determines when reward will be delivered/omitted in the self-timed movement task—just after it moves. Certainty in the timing of reward relative to its own movement would resolve the animal's uncertainty of its position in the value landscape, and may thus explain why dopaminergic ramping occurs prominently in the self-timed movement task, but not in Pavlovian tasks (*Hamilos and Assad, 2020*). Although the RPE model provides a plausible explanation for our findings, dopaminergic ramping signals are also consistent with broader views of 'reward expectation', such as value tracking as animals approach reward (*Hamid et al., 2016*; *Mohebi et al., 2019*). In a companion theoretical paper (*Hamilos and Assad, 2020*), we explore the reward expectation-based computational framework in more detail, including a reconciliation of apparently contradictory DAN signals reported in the context of a perceptual timing task (*Soares et al., 2016*).

## How do DAN signals affect movement in real time?

We found that trial-by-trial variability in ramping dynamics explained the precise timing of self-timed licks. However, because the animals could expect reward shortly after the first-lick, the ramping dopaminergic signal might serve as a passive monitor of reward expectation rather than causally influencing the timing of movement initiation. To distinguish these possibilities, we optogenetically manipulated SNc DAN activity. We found that exciting or inhibiting DANs altered the timing of the first-lick on the *concurrent* trial, in a manner suggesting an increase/decrease in the probability of movement, respectively. This suggests that endogenous DAN signaling could play a causal role in the initiation of reward-related movements in real time—but by what mechanism?

One possibility is that endogenous or exogenous DAN signals could increase the animal's motivation or heighten its expectation of reward, which then secondarily influences reward-related movement. There is some evidence that might support this view. Phillips et al. found that electrical stimulation of the VTA in rats elicited approach behavior for self-delivery of intravenous cocaine; however, the electrical stimulation could have activated non-DAN fibers/pathways *via* the VTA (*Phillips et al., 2003*). Hamid et al. found that selective optogenetic stimulation of DANs could shorten the latency for rats to engage in a port-choice task—but only if the rat was disengaged from the task; if the rat was already engaged in task performance, the latency became slightly *longer* (*Hamid et al., 2016*).

In contrast to these equivocal findings, a large body of evidence suggests that selective optogenetic stimulation or inhibition of DANs generally does *not* affect reward-related movements *on the same trial*. First, we ourselves could not evoke licking (nor inhibit spontaneous licking) outside the context of our self-timed movement task (*Figure 7—figure supplement 4*). Our mice were thirsty and perched near their usual juice tube, but offline DAN stimulation/inhibition did not alter licking behavior, even though we applied the same optical power that altered movement probability during the self-timed movement task. Numerous studies have also examined the effects of optogenetic modulation of DANs in Pavlovian conditioning paradigms, with the general finding that DAN modulation affects conditioned movements on *subsequent* trials or sessions—a learning effect—but not on the *same* trial (*Coddington and Dudman, 2018*; *Coddington and Dudman, 2021*; *Lee et al., 2020*; *Maes et al., 2020*; *Morrens et al., 2020*; *Pan et al., 2021*; *Saunders et al., 2018*). For example, Lee et al. found that optogenetic inhibition of mouse DANs at the same time as an olfactory conditioned stimulus had no effect on anticipatory licking on the concurrent trial, even though inhibition at the time of reward delivery reduced the probability and rate of anticipatory licking on subsequent trials (*Lee et al., 2020*). Thus, the preponderance of evidence argues against a simple scheme whereby modulating DAN activity leads to a change in motivation that automatically evokes or suppresses reward-related movements in real time. The fact that we observed robust, concurrent optogenetic modulation of movement timing in our experiment suggests that additional factors were at play for self-timed movements.

One possibility is that during self-timing, exogenous (optogenetic) stimulation of DANs summed with the endogenous ramping DAN signal, leading to supra-heightened motivation to obtain reward. However, when we deliberately over-stimulated DANs—eliciting even higher dopamine signals in the DLS (*Figure 7—figure supplement 2*)—we observed 'dyskinetic' body movements rather than purposive licking. An alternative possibility is that the explicit timing requirement of the self-timed movement task made it particularly responsive to dopaminergic modulation. A long history of pharmacological and lesion experiments suggests that the dopaminergic system modulates timing behavior (*Meck, 2006*; *Merchant et al., 2013*). Broadly speaking, conditions that increase/decrease dopamine availability or efficacy speed/slow the 'internal clock', respectively (*Dews and Morse, 1958*; *Mikhael and Gershman, 2019*; *Schuster and Zimmerman, 1961*; *Malapani et al., 1998*; *Meck, 1986*; *Meck, 2006*; *Merchant et al., 2013*). The dopaminergic ramping signals we observed also bear resemblance to Pacemaker-Accumulator models of neural timing, in which a hypothetical accumulator signals that an interval has elapsed when it reaches a threshold level (*Gallistel and Gibbon, 2000*; *Lustig and Meck, 2005*; *Meck, 2006*). To 'self-time' a movement also implies that the movement is prepared and pre-potent during the timing period, potentially making the relevant neural motor circuits more sensitive to dopaminergic modulation.

Regardless of the detailed mechanism, our results provide a link between dopaminergic signaling and the initiation of self-timed movements. Although endogenous dopaminergic ramping likely reflects reward expectation, we propose that these reward-related ramping signals can influence the

timing of movement initiation, at least in certain behavioral contexts. This framework also provides a link between two seemingly disparate roles that have been proposed for the dopaminergic system— reward/reinforcement-learning on one hand, and movement modulation on the other.

Importantly, we are not suggesting that DANs *directly* drive movement (like corticospinal or corti-cobulbar neurons). To the contrary, outside of the context of the self-timed movement task, we could not evoke reward-related movements by activating DANs. Even during the self-timed movement task, DAN stimulation did not elicit *immediate* movements: first-lick times still spanned a broad distribution from trial-to-trial. Moreover, dopaminergic ramping does not invariably lead to movement. For example, Kim et al. found dopaminergic ramping in the presence of visual cues that signaled proximity to reward, independent of reward-related movements (*Kim et al., 2019*). Consequently, we propose that when a movement is pre-potent (as in our self-timed movement task), dopaminergic signaling can modulate the *probability* of initiating that movement. Consistent with this view, we found that the endogenous ramping dynamics were highly predictive of the moment-by-moment probability of movement (as captured by the hazard function), with DAN signals becoming progressively better predictors as the time of movement onset approached.

This view of dopaminergic modulation of movement probability could be related to classic findings from extrapyramidal movement disorders, in which dysfunction of the nigrostriatal pathway produces aberrations in movement initiation rather than paralysis or paresis (*Bloxham et al., 1984*; *Fahn, 2011*; *Hallett and Khoshbin, 1980*). That is, movements *do* occur in extrapyramidal disorders, but at inappropriate times, either too little/late (e.g., Parkinson's), or too often (e.g., dyskinesias). Based on the deficits observed in Parkinsonian states (e.g., perseveration), this role may extend to behavioral transitions more generally, for example, starting new movements or stopping ongoing movements (*Guru et al., 2020*).

## Is DAN ramping also present before 'spontaneous' movements?

We have suggested that the ramping DAN signals in the self-timed movement task could be related to reward expectation coupled with the explicit timing requirement of the task. However, when we averaged DAN signals aligned to 'spontaneous' licks during the ITI, we also observed noisy, slow ramping signals building over seconds up to the time of the next lick, with a time course related to the duration of the inter-*lick* interval (*Figure 8—figure supplement 2*). This observation raises the possibility that slowly evolving DAN signals may be related to the generation of self-initiated movements more generally—although our highly trained animals may have also been 'rehearsing' timed movements between trials and/or expecting reward even for spontaneous licks.

## Relationship to setpoint and stretching dynamics in other neural circuits

We found that DAN signals predict movement timing via two low-dimensional signals: a baseline offset and a ramping dynamic that 'stretches' depending on trial-by-trial movement timing. Intriguingly, similar stretching of neural responses has been observed before self-timed movement in other brain areas in rats and primates, including the dorsal striatum (*Emmons et al., 2017*; *Mello et al., 2015*; *Wang et al., 2018*), lateral interparietal cortex (*Maimon and Assad, 2006*), presupplementary and supplementary motor areas (*Mita et al., 2009*), and dorsomedial frontal cortex (DMFC; *Remington et al., 2018*; *Sohn et al., 2019*; *Wang et al., 2018*; *Xu et al., 2014*). In the case of DMFC, applying dimensionality reduction to the population responses revealed two lower-dimensional characteristics that resembled our findings in DANs: (1) the speed at which the population dynamics unfolded was scaled ('stretched') to the length of the produced timing interval (*Wang et al., 2018*), and (2) the population state at the beginning of the self-timed movement interval ('setpoint') was correlated with the timed interval (*Remington et al., 2018*; *Sohn et al., 2019*). Recurrent neural network models suggested variation in stretching and setpoint states could be controlled by (unknown) tonic or monotonically-ramping inputs to the cortico-striatal system (*Remington et al., 2018*; *Sohn et al., 2019*; *Wang et al., 2018*). We found that DANs exhibit both baseline (e.g., 'setpoint') signals related to timing, as well as monotonically ramping input during the timing interval. Thus, through their role as diffusely-projecting modulators, DANs could potentially orchestrate variations in cortico-striatal dynamics observed during timing behavior. Ramping DAN signals could also be related to the slow

ramping signals that have been observed in the human motor system in anticipation of self-initiated movements, for example, readiness potentials in EEG recordings (*Deecke, 1996*; *Libet et al., 1983*).

### Possible relationship to motivational/movement vigor

In operant tasks in which difficulty is systematically varied over blocks of trials, increased intertrial dopamine in the nucleus accumbens has been associated with higher average reward rate and decreased latency to engage in a new trial, suggesting a link between dopamine and 'motivational vigor', the propensity to invest effort in work (*Hamid et al., 2016*; *Mohebi et al., 2019*). Intriguingly, we observed the *opposite* relationship in the self-timed movement task: periods with higher average reward rates had *lower* average baseline dopaminergic signals and later first-lick times. Moreover, for a given first-lick time (e.g., 3.5–3.75 s), we did not detect differences in baseline (or ramping) signals during periods with different average reward rates, such as near the beginning or end of a session. This difference between the two tasks may be due to their opposing strategic constraints: in the afore-mentioned experiments, faster trial initiation increased the number of opportunities to obtain reward, whereas earlier first-licks tended to decrease reward acquisition in our self-timed movement task.

The basal ganglia have also been implicated in controlling 'movement vigor,' generally referring to the speed, force or frequency of movements (*Bartholomew et al., 2016*; *Dudman and Krakauer, 2016*; *Panigrahi et al., 2015*; *Turner and Desmurget, 2010*; *Yttri and Dudman, 2016*). The activity of nigrostriatal DANs has been shown to correlate with these parameters during movement bouts and could promote more vigorous movement via push-pull interactions with the direct and indirect pathways (*Barter et al., 2015*; *da Silva et al., 2018*; *Mazzoni et al., 2007*; *Panigrahi et al., 2015*). Movement vigor might also entail earlier self-timed movements, mediated by moment-to-moment increases in dopaminergic activity.

If moving earlier is a signature of greater movement vigor, then earlier self-timed movements might also be executed with greater force/speed. We looked for movement-related vigor signals, examining both the amplitude of lick-related EMG signals and the latency between lick initiation and lick-tube contact. We detected no consistent differences in these force- or speed-related parameters as a function of movement time; on the contrary, the EMG signals were highly stereotyped irrespective of the first-lick time (data not shown). It is possible that vigor might affect movement timing without affecting movement kinematics/dynamics—but, if so, the distinction between 'timing' and 'vigor' would seem largely semantical.

### Overall view

We have posited that dopaminergic ramping reflects reward expectation, a common element of behavioral paradigms that reveal slow dopaminergic ramping. Furthermore, our optogenetic manip-ulations indicate that dopaminergic signals do not directly trigger movements, but rather act as if modulating the probability of the pre-potent self-timed movement. Taken together, these observa-tions suggest that as DAN activity ramps up, the probability of movement likewise increases. In this view, different rates of increase in DAN activity lead to shorter or longer elapsed intervals before movement, on average. This framework leaves open the question of what makes movement timing 'probabilistic.' One possibility is that recurrent cortical-basal ganglia–thalamic circuits could act to generate movements 'on their own,' without direct external triggers (e.g., a 'go!' cue). By providing crucial modulation of these circuits, DANs could tune the propensity to make self-timed movements—and pathological loss of DANs could reduce the production of such movements. Future experiments should address how dynamic dopaminergic input influences downstream motor circuits involved in self-timed movements.

# Materials and methods

**Key resources table**

| Reagent type (species) or resource | Designation | Source or reference | Identifiers | Additional information |
|---|---|---|---|---|
| Strain, strain background (*M. musculus*) | DAT-Cre | The Jackson Laboratory, Bar Harbor, ME | B6.SJL-*Slc6a3*[tm1.1(cre)Bkmm]/J RRID:IMSR_JAX:020080 | Cre expression in dopaminergic neurons |

*Continued on next page*

*Continued*

| Reagent type (species) or resource | Designation | Source or reference | Identifiers | Additional information |
|---|---|---|---|---|
| Strain, strain background (*M. musculus*) | Wild-type | The Jackson Laboratory, Bar Harbor, ME | C57BL/6 RRID:IMSR_JAX:000664 | |
| Other | tdTomato ("tdt") | UNC Vector Core, Chapel Hill, NC | AAV1-CAG-FLEX-tdT | Virus, for control photometry expression |
| Other | gCaMP6f | Penn Vector Core, Philadelphia, PA | AAV1.Syn.Flex.GCaMP6f.WPRE.SV40 | Virus, for photometry expression |
| Other | DA2m | Vigene, Rockville, MD | AAV9-hSyn-DA4.4(DA2m) | Virus, for photometry expression |
| Other | dLight1.1 | Lin Tian Lab; Children's Hospital Boston Viral Core, Boston, MA | AAV9.hSyn.dLight1.1.wPRE | Virus, for photometry expression |
| Other | turboRFP | Penn Vector Core | AAV1.CB7.CI.TurboRFP.WPRE.rBG | Virus, for control photometry expression |
| Other | ChR2 | UNC Vector Core, Chapel Hill, NC | AAV5-EF1a-DIO-hChR2(H134R)-EYFP-WPRE-pA | Virus, for opsin expression |
| Other | ChrimsonR | UNC Vector Core, Chapel Hill, NC | AAV1-hSyn-FLEX-ChrimsonR-tdT | Virus, for opsin expression |
| Other | stGtACR2 | Addgene/Janelia Viral Core, Ashburn, VA | AAV2/8-hSyn1-SIO-stGtACR2-FusionRed | Virus, for opsin expression |
| Software, algorithm | Matlab | Mathworks | Matlab2018B | For most analyses |
| Software, algorithm | Julia Programming Language | The Julia Project | Julia 1.5.3 | For probabilistic models |
| Software, algorithm | Gen.jl | The Gen Team | Gen.jl | For probabilistic models |

## Animals

Adult male and female hemizygous DAT-cre mice (*Bäckman et al., 2006*; B6.SJL-*Slc6a3*$^{tm1.1(cre)Bkmm}$/J, RRID:IMSR_JAX:020080; The Jackson Laboratory, Bar Harbor, ME) or wild-type C57BL/6 mice were used in all experiments (>2 months old at the time of surgery; median body weight 23.8 g, range 17.3–31.9 g). Mice were housed in standard cages in a temperature and humidity-controlled colony facility on a reversed night/day cycle (12 hr dark/12 hr light), and behavioral sessions occurred during the dark cycle. Animals were housed with enrichment objects provided by the Harvard Center for Comparative Medicine (IACUC-approved plastic toys/shelters, e.g., Bio-Huts, Mouse Tunnels, Nest Sheets, etc.) and were housed socially whenever possible (1–5 mice per cage). All experiments and protocols were approved by the Harvard Institutional Animal Care and Use Committee (IACUC protocol #05098, Animal Welfare Assurance Number #A3431-01) and were conducted in accordance with the National Institutes of Health Guide for the Care and Use of Laboratory Animals.

## Surgery

Surgeries were conducted under aseptic conditions and every effort was taken to minimize suffering. Mice were anesthetized with isoflurane (0.5–2% at 0.8 L/min). Analgesia was provided by *s.c.* 5 mg/kg ketoprofen injection during surgery and once daily for 3 d postoperatively (Ketofen, Parsippany, NJ). Virus was injected (50 nL/min), and the pipet remained in place for 10 min before removal. 200 µm, 0.53 NA blunt fiber optic cannulae (Doric Lenses, Quebec, Canada) or tapered fiber optic cannulae (200 µm, 0.60 NA, 2 mm tapered shank, OptogeniX, Lecce, Italy) were positioned at SNc, VTA, or DLS and secured to the skull with dental cement (C&B Metabond, Parkell, Edgewood, NY). Neck EMG electrodes were constructed from two Teflon-insulated 32 G stainless steel pacemaker wires attached

to a custom socket mounted in the dental cement. Sub-occipital neck muscles were exposed by blunt dissection and electrode tips embedded bilaterally.

## Stereotaxic coordinates (from bregma and brain surface)

Viral Injection:

SNc: 3.16 mm posterior, ±1.4 mm lateral, 4.2 mm ventral
VTA: 3.1 mm posterior, ±0.6 mm lateral, 4.2 mm ventral
DLS: 0 mm anterior, ±2.6 mm lateral, 2.5 mm ventral.

Fiber Optic Tips:

SNc/VTA: 4.0 mm ventral (photometry) or 3.9 mm ventral (optogenetics).
DLS: 2.311 mm ventral (blunt fiber) or 4.0 mm ventral (tapered fiber)

## Virus

Photometry:

**tdTomato ("tdt")**: AAV1-CAG-FLEX-tdT (UNC Vector Core, Chapel Hill, NC), 100 nL used alone or in mixture with other fluorophores (below), working concentration $5.3 \cdot 10^{12}$ gc/mL

**gCaMP6f (at SNc or VTA)**: 100 nL AAV1.Syn.Flex.GCaMP6f.WPRE.SV40 ($2.5 \cdot 10^{13}$ gc/mL, Penn Vector Core, Philadelphia, PA). Virus was mixed in a 1:3 ratio with tdt (200 nL total)

**DA$_{2m}$ (at DLS)**: 200–300 nL AAV9-hSyn-DA4.4(DA2m) (working concentration: *ca.* $3 \cdot 10^{12}$ gc/mL, Vigene, Rockville, MD) + 100 nL tdt

**dLight1.1 (at DLS)**: 300 nL AAV9.hSyn.dLight1.1.wPRE bilaterally at DLS (*ca.* $9.6 \cdot 10^{12}$ gc/mL, Children's Hospital Boston Viral Core, Boston, MA) + 100 nL AAV1.CB7.CI.TurboRFP.WPRE.rBG (*ca.* $1.01 \cdot 10^{12}$ gc/mL, Penn Vector Core)

Optogenetic stimulation/inhibition (all bilateral at SNc):

**ChR2**: 1,000 nL AAV5-EF1a-DIO-hChR2(H134R)-EYFP-WPRE-pA ($3.2 \cdot 10^{13}$ gc/mL, UNC Vector Core, Chapel Hill, NC)

**ChrimsonR±dLight1.1**: 700 nL AAV1-hSyn-FLEX-ChrimsonR-tdT ($4.1 \cdot 10^{12}$ gc/mL, UNC Vector Core, Chapel Hill, NC)±400–550 nL AAV9-hSyn-dLight1.1 bilaterally at DLS (*ca.* $10^{13}$ gc/mL, Lin Tian Lab, Los Angeles, CA)

**stGtACR2**: 300 nL 1:10 AAV2/8-hSyn1-SIO-stGtACR2-FusionRed (working concentration $4.7 \cdot 10^{11}$ gc/mL, Addgene/Janelia Viral Core, Ashburn, VA)

## Water-deprivation and acclimation

Animals recovered for 1 week postoperatively before water deprivation. Mice received daily water supplementation to maintain ≥80% initial body weight and fed *ad libitum*. Mice were habituated to the experimenter and their health was monitored carefully following guidelines reported previously (*Guo et al., 2014*). Training commenced when mice reached the target weight (~8–9 d post-surgery).

## Histology

Mice were anesthetized with >400 mg/kg pentobarbital (Somnasol, Henry Schein Inc, Melville, NY) and perfused with 10 mL 0.9% sodium chloride followed by 50 mL ice-cold 4% paraformaldehyde in 0.1 M phosphate buffer. Brains were fixed in 4% paraformaldehyde at 4 °C for >24 hr before being transferred to 30% sucrose in 0.1 M phosphate buffer for >48 hr. Brains were sliced in 50 µm coronal sections by freezing microtome, and fluorophore expression was assessed by light microscopy. The sites of viral injections and fiber optic placement were mapped with an Allen Mouse Brain Atlas.

## Behavioral rig, data acquisition, and analysis

A custom rig provided sensory cues, recorded events and delivered juice rewards under the control of a Teensy 3.2 microprocessor running a custom Arduino state-system behavioral program with MATLAB serial interface. Digital and analog signals were acquired with a CED Power 1400 data acquisition system/Spike2 software (Cambridge Electronic Design Ltd, Cambridge, England). Photometry and behavioral events were acquired at 1000 Hz; movement channels were acquired at 2000 Hz. Video was acquired with FlyCap2 or Spinnaker at 30 fps (FLIR Systems, Wilsonville, OR). Data were analyzed with custom MATLAB statistics packages.

## Self-timed movement task

Mice were head-fixed with a juice tube positioned in front of the tongue. The spout was placed as far away from the mouth as possible so that the tongue could still reach it to discourage compulsive licking (*Guo et al., 2014*), ~1.5 mm ventral and ~1.5 mm anterior to the mouth. During periods when rewards were not available, a houselamp was illuminated. At trial start, the houselamp turned off, and a random delay ensued (0.4–1.5 s) before a cue (simultaneous LED flash and 3300 Hz tone, 100 ms) indicated start of the timing interval. The timing interval was divided into two windows, early (0–3.33 s in most experiments; 0–4.95 s in others) and reward (3.33–7 s; 4.95–10 s), followed by the intertrial interval (ITI, 7–17 s; 10–20 s). The window in which the mouse first licked determined the trial outcome (early, reward, or no-lick). An early first-lick caused an error tone (440 Hz, 200 ms) and houselamp illumination, and the mouse had to wait until the full timing interval had elapsed before beginning the ITI. Thus there was no advantage to the mouse of licking early. A first-lick during the reward window caused a reward tone (5050 Hz, 200 ms) and juice delivery, and the houselamp remained off until the end of the trial interval. If the timing interval elapsed with no lick, a time-out error tone played (131 Hz, 2 s), the houselamp turned on, and ITI commenced. During the ITI and pre-cue delay ('lamp-off interval'), there was no penalty for licking.

Mice learned the task in three stages (*Figure 1—figure supplement 1A*). On the first 1–4 days of training, mice learned a beginner-level task, which was modified in two ways: (1) to encourage participation, if mice did not lick before 5 s post-cue, they received a juice reward at 5 s; and (2) mice were not penalized for licking in reaction to the cue (within 500 ms). When the mouse began self-triggering ≥50% of rewards (days 2–6 of training), the mouse advanced to the intermediate-level task, in which the training reward at 5 s was omitted, and the mouse had to self-trigger all rewards. After completing >250 trials/day on the intermediate task (usually days 4–7 of training), mice advanced to the mature task, with no reaction licks permitted. All animals learned the mature task and worked for ~400–1,500 trials/session.

## Hazard function correction of survival bias in the timing distribution

The raw frequency of a particular response time in the self-timed movement task is 'distorted' by how often the animal has the chance to respond at that time (*Anger, 1956*). This bias was corrected by calculating the hazard function, which takes into account the number of response opportunities the animal had at each timepoint. The hazard function is defined as the conditional probability of moving at a time, *t*, given that the movement has not yet occurred (referred to as 'IRT/Op' analysis in the old Differential Reinforcement of Low Rates (DRL) literature). The hazard function was computed by dividing the number of first-movements in each 250 ms bin of the first-lick timing histogram by the total number of first-movements occurring at that bin-time or later—the total remaining 'opportunities'.

## Online movement monitoring

Movements were recorded simultaneously during behavior with four movement-control measurements: neck EMG (band-pass filtered 50–2000 Hz, 60 Hz notch, amplified 100–1000x), back-mounted accelerometer (SparkFun Electronics, Boulder, CO), high-speed camera (30 Hz, FLIR Systems, Wilsonville, OR), and tdTomato photometry. All control signals contained similar information, and thus only a subset of controls was used in some sessions.

## Photometry

Fiber optics were illuminated with 475 nm blue LED light (Plexon, Dallas, TX) (SNc/VTA: 50 μW, DLS: 35 μW) measured at patch cable tip with a light-power meter (Thorlabs, Newton, NJ). Green

fluorescence was collected via a custom dichroic mirror (Doric Lenses, Quebec, Canada) and detected with a Newport 1401 Photodiode (Newport Corporation, Irvine, CA). Fluorescence was allowed to recover ≥1 d between recording sessions. To avoid crosstalk in animals with red control fluorophore expression, the red channel was recorded at one of the three sites (SNc, VTA, or DLS, 550 nm lime LED, Plexon, Dallas, TX) while GCaMP6f, dLight1.1 or $DA_{2m}$ was recorded simultaneously only at the other implanted sites.

## dF/F

Raw fluorescence for each session was pre-processed by removing rare singularities (single points > 15 STD from the mean) by interpolation to obtain F(t). To correct photometry signals for bleaching, dF/F was calculated as:

$$\frac{dF}{F}(t) = \frac{F(t) - F_0(t)}{F_0(t)}$$

where $F_0(t)$ is the 200 s moving average of F(t) (*Figure 2—figure supplement 2A*). We tested several other complementary methods for calculating dF/F and all reported results were robust to dF/F method (see Materials and methods: dF/F method characterization and validation). To ensure dF/F signal processing did not introduce artifactual scaling or baseline shifts, we also tested several complementary techniques to isolate undistorted F(t) signals where possible and quantified the amount of signal distortion when perfect isolation was not possible (see Materials and methods: 'dF/F method characterization and validation', below, and *Figure 2—figure supplement 2C*).

## dF/F method characterization and validation

dF/F calculations are intended to reduce the contribution of slow fluorescence bleaching to fiber photometry signals, and many such methods have been described (*Kim et al., 2019*; *Mohebi et al., 2019*; *Soares et al., 2016*). However, dF/F methods have the potential to introduce artifactual distortion when the wrong method is applied in the wrong setting. Thus, to derive an appropriate dF/F method for use in the context of the self-timed movement task, we characterized and quantified artifacts produced by four candidate dF/F techniques.

### Detailed description of complementary dF/F methods

1. Normalized baseline: a commonly used dF/F technique in which each trial's fluorescence is normalized to the mean fluorescence during the 5 s preceding the trial.
2. Low-pass digital filter: $F_0$ is the low-pass, digital infinite impulse response (IIR)-filtered raw fluorescence for the whole session (implemented in MATLAB with the built-in function *lowpass* with $f_c = 5 \cdot 10^{-5}$ Hz, steepness = 0.95).
3. Multiple baseline: a variation of Method 1, in which each trial's fluorescence is normalized by the mean fluorescence during the 5 s preceding the current trial, as well as five trials before the current trial and five trials after the current trial.
4. Moving average: $F_0$ is the 200 s moving average of the raw fluorescence at each point (100 s on either side of the measured timepoint).

Although *normalized baseline* (Method 1) is commonly used to correct raw fluorescence signals (F) for bleaching, this technique assumes that baseline activity has no bearing on the trial outcome; however, because the mouse decides when to move in the self-timed movement task, it is possible that baseline activity may differ systematically with the mouse's choice on a given trial. Thus, normalizing F to the baseline period would obscure potentially physiologically relevant signals. More insidiously, if baseline activity *does* vary systematically with the mouse's timing, normalization can also introduce substantial amplitude scaling and y-axis shifting artifacts when correcting F with this method (*Figure 2—figure supplement 2C*, middle panels). Thus, Methods 2–4 were designed and optimized to isolate photometry signals minimally distorted by bleaching signals and systematic baseline differences during the self-timed movement task. Methods 2–4 produced the same results in all statistical analyses, and the moving average method is shown in all figures.

## Isolating minimally-distorted photometry signals with paired trial analyses of raw fluorescence

Although slow bleaching prevents comparison of raw photometry signals (F) at one time in a behavioral session with those at another time, the time course of appreciable bleaching was slow enough in the reported behavioral sessions that minimal bleaching occurred over the course of three trials (~1 min, *Figure 2—figure supplement 2A*). Thus, to observe the least distorted photometry signals possible, we compared F between *pairs of consecutive* trials (*Figure 2—figure supplement 2B-C*). We compared F baseline signals between all paired trials in which an early trial (unrewarded first-lick between 0.7 and 2.9 s; abbreviated as 'E') was followed by a rewarded trial (first-lick between 3.4 and 7 s; abbreviated as 'R'); this two-trial sequence is thus referred to as an 'ER' comparison. To ensure systematic differences did not result from subtle bleaching in the paired-trial interval, we reversed the ordering contingency and also compared all Rewarded trials preceding Early trials ('RE' comparison). The same systematic relationship between baseline signals and first-lick time was found for paired trials analyzed by raw F (*Figure 2—figure supplement 2C*, left panels).

## Quantification of artifactual amplitude scaling/baseline shifts introduced by dF/F processing

Each Candidate dF/F Method was applied to the same Paired Trial datasets described above. The resulting paired-fluorescence datasets were normalized after processing (minimum dF/$F$ = 0, maximum = 1). The amount of distortion introduced by dF/F was quantified with a Distortion Index (DI), which was calculated as:

$$\text{Distortion Index, DI(t)} = |F(t) - dF/F(t)|$$

where F(t) and dF/F(t) are the normalized, paired-trial raw fluorescence signal or dF/F signal at time $t$, respectively. $t$ spanned from the beginning of the n-1th trial (–20 s) to the end of the nth trial (20 s), aligned to the cue of the nth trial (*Figure 2—figure supplement 2C, bottom panels*). The DI shown in plots has been smoothed with a 200 ms moving average kernel for clarity.

As expected, normalizing fluorescence to the baseline period (*normalized baseline*) erased the correlation of baseline dF/F signals with first-lick time (*Figure 2—figure supplement 2C*, middle panels). More insidiously, this also resulted in distortion of GCaMP6f dynamics *during* the timing interval, evident in the diminished difference between E-signals compared to R-signals relative to the shapes observed in the raw fluorescence paired-trial comparison (*Figure 2—figure supplement 2C*, middle-bottom panel). However, dF/F Methods 2–4 visually and quantitatively recapitulated the dynamics observed in the raw fluorescence comparison (*Figure 2—figure supplement 2C*, right panels).

These results were corroborated by time-in-session permutation tests in which datasets for single sessions were divided into thirds (beginning of session, middle of session, and end of session). The differences between baseline and ramping dynamics observed in whole-session averages were present even within these shorter blocks of time within the session (i.e., faster ramping and elevated baseline signals on trials with earlier self-timed licks). Furthermore, permutation tests in which the block identity (beginning, middle, end) was shuffled showed that this pattern held when trials with earlier first-licks from the end of the session were compared with trials with later first-licks from the beginning of the session (and vice versa).

### Normalized dF/F for comparing dopamine sensor signals

DA$_{2m}$ was about twice as bright as dLight1.1, and thus generally yielded larger and less noisy dF/F signals. To compare the two extracellular dopamine sensors in the same plot, dF/F was normalized for each signal by the amplitude of its lick-related transient. dF/F was calculated as usual, and then the mean baseline-to-transient peak amplitude was measured for trials with first-licks occurring between 2 and 3 s. Percentage NdF/F is reported as the percentage of this amplitude.

### Dopamine sensor kinetics

dLight1.1 is an extracellular dopamine sensor derived from the dopamine-1-receptor, and has fast reported kinetics: rise t$_{1/2}$ = 9.5 ± 1.1 ms, decay t$_{1/2}$ = 90 ± 11 ms (*Patriarchi et al., 2018*). DA$_{2m}$ is a

new extracellular dopamine indicator derived from the dopamine-2-receptor, which provides brighter signals. $DA_{2m}$ signals have been reported to decay slowly in slice preparations but are much faster *in vivo*, presumably because endogenous dopamine-clearance mechanisms are preserved: reported rise $t_{1/2}$ ~50 ms, decay $t_{1/2}$ ~360 ms in freely behaving mice; decay $t_{1/2}$ ~190 ms in head-fixed *Drosophila* (**Sun et al., 2020**). To estimate the dopamine-sensor kinetics in our head-fixed mice, we examined the phasic fluorescence transient occurring on unrewarded first-licks (0.5–3.3 s), which showed a stereotyped fast rise and decay with both sensors (**Figure 2—figure supplement 3D-E**). While the transient was somewhat complex, reminiscent of phasic burst-pause responses sometimes observed for movement-related DAN activity (**Coddington and Dudman, 2018**; **Coddington and Dudman, 2019**), we measured the time for average fluorescence to decay from the peak of the transient to half the baseline-to-peak amplitude. We found decay $t_{1/2}$~75 ms for dLight1.1 and $t_{1/2}$~125 ms for $DA_{2m}$ (**Figure 3—figure supplement 1**). Given that the dopaminergic ramping signals in our study evolved over several seconds, the kinetics of both dopamine sensors are thus fast enough that they should not have caused appreciable distortion of the slow ramping dynamics.

## Pearson's correlation of baseline/lamp-off-to-cue interval signals to first-lick time

The mean SNc GCaMP6f signal during the 'baseline' (2 s interval before the lamp-off event) or minimum lamp-off interval ('LOI'; –0.4 s to 0 s, the cue-time) was compared to the first-lick time for pooled trials in **Figure 2C** by calculating the Pearson correlation coefficient. There were at least 700 trials in each pooled set of trials (0.75–4 s included).

## DAN signal encoding model

To test the independent contribution of each task-related input to the photometry signal and to select the best model, we employed a nested fitting approach, in which each dataset was fit multiple times (in 'nests'), with models becoming progressively more complex in subsequent nests. The nests fit to the GCaMP6f photometry data employed the inputs $X^{(j)}$ at each $j^{th}$ nest:

Null Model: $X^{(0)} = x_0$
Nest 1: $X^{(1)} = X^{(0)}+$ tdTomato (tdt)
Nest 2: $X^{(2)} = X^{(1)}+$ cue + first-lick
Nest 3: $X^{(3)} = X^{(2)}+$ EMG/accelerometer
Nest 4: $X^{(4)} = X^{(3)}+$ time-dependent baseline offset
Nest 5: $X^{(5)} = X^{(4)}+$ stretch representing percentages of interval

Overfitting was penalized by ridge regression, and the optimal regularization parameter for each nest was obtained by five-fold cross-validation to derive the final model fit for each session. Model improvement by each input was assessed by the percentage loss improvement at the nest where the input first appeared compared to the prior nest. The loss improvement of Nest 1 was compared to the Null Model (the average of the photometry timeseries). The nested model of tdt control photometry signals was the same, except Nest 1 was omitted.

The GLM for each nest takes the form:

$$Y = \Theta X^{(j)}$$

where Y is the *1xn* vector of the photometry signal across an entire behavioral session (*n* is the total number of sampled timepoints); $X^{(j)}$ is the *dxn* design matrix for nest *j*, where the rows correspond to the $d_j$ predictors for nest *j* and the columns correspond to each of the *n* sampled timepoints of Y; and $\Theta$ is the *dx1* vector of fit weights.

Y is the concatenated photometry timeseries taken from trial start (lamp-off) to the time of first-lick. Because of day-to-day/mouse-to-mouse variation (ascribable to many possible sources, for example, different neural subpopulations, expression levels, behavioral states, etc.), each session was fit separately.

The $d_j$ design matrix predictors were each scaled (maximum amplitude 1) and grouped by input to the model. The timing-independent inputs were: 1. Null offset ($x_0$, 1 predictor), 2. tdt (1 predictor), 3. cue (24 predictors), 4. first-lick (28 predictors), and 5. EMG/accelerometer (44 predictors). The timing-dependent inputs were: 6. timing-dependent baseline offset (1 predictor), 7. stretch (500 predictors).

To reduce the number of predictors, the cue, first-lick and EMG/accelerometer predictors (*Figure 5—figure supplement 1C*) were composed from sets of basis kernels as described previously (*Park et al., 2014*; *Runyan et al., 2017*). The cue basis kernels were spaced 0 to 500 ms post-cue, and first-lick basis kernels were spaced –500 to 0 ms relative to first-lick, the typically-observed windows of stereotypical sensory and motor-related neural responses. For nuisance movements (EMG/accelerometer), events were first discretized by thresholding (*Figure 5—figure supplement 1B*) and then convolved with basis kernels spanning –500 to 500 ms around the event. This window was consistent with the mean movement-aligned optical artifact observed in the tdt channel. The timing-dependent baseline offset was encoded as a constant offset spanning from lamp-off until first-lick, with amplitude taken as linearly proportional to the timed interval on the current trial. The timing-dependent stretch input was composed of 500 predictors, with each predictor containing 1's tiling 0.05% of the cue-to-lick interval, and 0's otherwise (*Figure 5—figure supplement 1D*). Importantly, the stretch was not constrained in any way to form ramps.

Basis sets were optimized to minimize Training Loss, as calculated by mean squared error of the unregularized model:

$$\operatorname{argmin}_{X^{(j)}}[\text{Training Loss}(\theta) = \tfrac{1}{n}(Y - \theta X^{(j)})^2]$$

Superfluous basis set elements that did not improve Training Loss compared to the Null Model were not included in the final model. Goodness of the training fit was assessed by Akaike Information Criterion (AIC), Bayesian Information Criterion (BIC), $R^2$, and Training Loss. The optimal, regularized model for each nest/session was selected by five-fold cross-validation in which the regularization parameter, $\lambda_j$, was optimized for minimal average Test Loss:

$$\operatorname{argmin}_{\lambda_j}[\text{Test Loss}(\theta, \lambda_j) = \tfrac{1}{n}(Y - \theta X^{(j)})^2 + \lambda_j|\theta|^2]$$

Test Loss for each optimal model was compared across nests to select the best model for each session. Models were refit with the optimal $\lambda_j$ to obtain the final fit.

Model error was simulated 1,000x by redrawing $\Theta$ coefficients consistent with the data following the method described by *Gelman and Hill, 2006*, and standard errors were propagated across sessions. The absolute value of each predictor was summed and divided by the total number of predictors for that input to show the contribution of the input to the model (*Figure 5—figure supplement 1G*). To simulate the modeled session's photometry signal for each nest *j*, Yfit was calculated as $\Theta X^{(j)}$ and binned by the time of first-lick relative to the cue. The error in the simulation was shown by calculating $\text{Yfit}_{sim} = \Theta_{sim} X^{(j)}$ for 300 simulated sets of $\Theta_{sim}$.

## Principal component analysis (PCA)

Unsmoothed ramping intervals for photometry timeseries were subjected to PCA and reconstructed with the first three principal components (PCs). To derive a PCA fit matrix with ramping intervals of the same number of samples, the length of each trial was scaled up by interpolation to the maximum ramping interval duration (5,700 samples):

$$7 \text{ s (trial duration)} - 0.7 \text{ s (cue buffer)} - 0.6 \text{ s (first lick buffer)} =$$

$$5.7 \text{ s (maximum ramping interval)}$$

Following PC-fitting, datasets were down-sampled to produce a fit of the correct time duration. Trials where the ramping interval was <0.1 s were excluded to exclude noise from down-sampling.

## First-lick time decoding model

A nested, generalized linear model was derived to predict the first-lick time on each trial in a session and to quantify the contribution of previous reward history and photometry signals to the prediction. The model was of the form:

$$\log(y) = bx$$

where *y* is the first-lick time, *b* is a vector of fit coefficients and *x* is a vector of predictors. The nested model was constructed such that predictors occurring further back in time (such as reward

history) and confounding variables (such as tdt photometry signals) were added first to determine the additional variance explained by predictors occurring closer to the time of first-lick, which might otherwise obscure the impact of these other variables. The predictors, in order of nesting, were:

> Nest 0: b0 (Null model, average log-first-lick time)
> Nest 1: b1 = b0 + first lick time on previous trial (trial 'n-1')
> Nest 2–5: b2 = b1 + previous trial outcome (1,0)*
> Nest 6: b3 = b2 + median photometry signal in 10 s window before lamp-off ('ITI')
> Nest 7: b4 = b3 + median photometry signal from lamp-off to cue ('lamp-off interval')
> Nest 8: b5 = b4 + tdt threshold crossing time**
> Nest 9: b6 = b5 + GCaMP6f threshold crossing time**

where all predictors were normalized to be in the interval (0,1).

\* Outcomes included (in order of nest): Reaction (first-lick before 0.5 s), Early (0.5–3.333 s), Reward (3.333–7 s), ITI (7–17 s). No-lick was implied when all four outcomes were encoded as zeros.

\*\* Details on threshold-crossing time and alternative models included in Materials and methods: 'Derivation of threshold and alternative decoding models'.

To exclude the sensory- and motor-related transients locked to the cue and the first-lick events in the threshold-crossing nests, the ramping interval was conservatively defined as 0.7 s post-cue up until 0.6 s before first-lick, and the minimum ramping interval for fitting was 0.1 s. Thus, for a trial to be included in the model, the first lick occurred between 1.4 s to 17 s (end of trial).

Initial model goodness of fit was assessed by $R^2$, mean-squared loss and BIC. Models were five-fold cross-validated with ridge regression at each nest to derive the final models, as described above. 95% confidence intervals on model coefficients were calculated by two-sided t-test with standard errors propagated across sessions.

## Derivation of threshold and alternative decoding models

### Single-threshold models

As a metric of the predictive power of ramping DAN signals on first-lick time, we derived a threshold-crossing model. A threshold-crossing event was defined as the first time after the cue when the photometry signal exceeded and remained above a threshold level up until the time of first-lick on each trial. Importantly, while the analysis approach is reminiscent of pacemaker-accumulator models for timing, we make no claims that the analysis is evidence for pacemaker-accumulator models. Rather threshold-crossing times provided a convenient metric to compare the rate of increase in signals between trials.

Photometry timeseries for GCaMP6f and tdt were de-noised by smoothing with a 100 ms Gaussian kernel (kernel was optimized by grid screen of kernels ranging between 0 and 200 ms to minimize noise without signal distortion). To completely exclude the sensory- and motor-related transients locked to the cue and the first-lick events, the ramping interval was conservatively defined as 0.7 s post-cue up until 0.6 s before the first-lick. To eliminate chance crossings due to noise, we imposed a stiff, debounced threshold condition: to be considered a threshold crossing event, the photometry signal had to cross the threshold from low-to-high and remain above this level until the end of the ramping interval.

To derive an unbiased threshold for each session, we tested 100 evenly-spaced candidate threshold levels spanning the minimum-to-maximum photometry signal during the ramping interval for each session. Depending on threshold level, some trials never crossed, that is, signal always remained below threshold or started and ended above threshold. Thus, the lowest candidate threshold for which there was a maximum number of trials crossing during the timing interval was selected as the 'mid-level' threshold-crossing point. This threshold was specific to each photometry signal tested on each session. Threshold-crossing time was included in the decoding model as the normalized time on the ramping interval (0,1). If a trial never crossed threshold, it was encoded as a zero. If no trials ever crossed threshold, the threshold predictor was encoded as a vector of ones, thus penalizing the model for an additional predictor but providing no new information.

### Multi-threshold model

An alternative model employed three unbiased thresholds: (1) the lowest threshold with ≥50 trials crossing ('min'); (2) the lowest threshold with the most crossings ('mid,' described above); and (3) the

highest threshold with ≥50 trials crossing ('max'). For tdt datasets, trials rarely met the monotonic threshold constraint (usually the signals oscillated above and below the threshold throughout the ramping interval, failing to meet the debouncing constraint). Thus, to include tdt signals as conservatively as possible, we relaxed the 50-trial minimum constraint, taking the threshold with the most trials crossing, which was usually around 10 or fewer. The addition of more thresholds did not substantially improve the cross-validated model compared to the single-threshold model (*Figure 6—figure supplement 1*).

## Principal component analysis (PCA) threshold-crossing models

In another version of the decoding model, the threshold-crossing procedures were applied to ramping intervals fit with the first three PCs (as described in Materials and methods: 'Principal Component Analysis (PCA)') to derive a PCA version of the single-threshold and multi-threshold models. PCA analysis on tdt datasets showed no consistent PCs, and thus these PCs were not included in the decoding model. Instead, the actual tdt data was employed in the threshold model as in the other models described.

## Hierarchical Bayesian modeling of single-trial dynamics

The probability of each single-trial SNc GCaMP6f signal belonging to a ramp *vs.* step model class was determined via Hierarchical Bayesian Model fitting with probabilistic programs written in the novel probabilistic programming language, Gen.jl, which is embedded in the Julia Programming Language (*Cusumano-Towner et al., 2019*). The top of the model hierarchy was the model class (linear ramp *vs.* step function) and the lower level was the respective parameterization of the two model classes (described below).

The probability of the step *vs.* ramp model class was inferred with data-driven inference. The best fit (step or ramp and parameterization) for each trial was calculated across 20 iterations (Gen *Traces*) of hierarchical modeling with 50 rounds of probabilistic refinement (computation *via* Gen Importance Resampling) per iteration (in model testing, models typically converged to their steady-state probability of model class within only 30 rounds of refinement, but 50 rounds were used conservatively to reduce the likelihood of suboptimal classifications).

### Data-driven inference procedure

Each iteration of model fitting began at the top level of the hierarchy with a coin toss: with 50% probability, the probabilistic program would initialize with a model of either the Ramp or Step class. For data-driven inference, a Gen *Proposal* for the parameterization for this model class was then probabilistically generated. Data-driven proposals were designed to improve fitting efficiency and reduce computation time, allowing for faster convergence and better model fits as determined by the fit log-likelihood. The proposal heuristics were as follows:

### Ramp model

A data-driven proposal was generated by dynamic noise random sample consensus (RANSAC; *Cusumano-Towner and Mansinghka, 2018*) with additional data-driven constraints (see function *ransac_assist_model_selection_proposal*; in the Julia Language Github files):

### Constraints:

1. SLOPE, *a*. The maximum data-supported slope was used to set the variance of slope sampling:

$$a \sim Gaussian(\text{RANSAC sampled slope}, \tfrac{maxslope}{2})$$

where *maxslope* was defined as the difference of the maximum and minimum signal within the trial dataset divided by the total duration of the trial (by definition, the largest slope supported by the data).

2. INTERCEPT, *b*. The initial search for the intercept ("$b_{max}$") was calculated as the intercept for the calculated *maxslope* parameter, and this was used to set the noise level on sampling of the intercept parameter:

$$b \sim Gaussian(\text{RANSAC sampled intercept}, \tfrac{b_{max}}{2})$$

3. NOISE, σ. Parametrized noise level was sampled as:

$$\sigma \sim Beta(\text{a}, \beta)$$

where α,β are the parameters of the beta distribution with mode = std(signal).

## Step model
The data-driven proposal included two constraints/heuristics:

1. STEPTIME. *Derivative constraint*: To avoid sampling all unlikely step-times, *steptimes* were sampled uniformly from the timepoints where the derivative of the signal was in the highest 5% of the signal's derivative across the trial dataset: $steptime \sim uniform$ (indices of 95th percentile of derivative of the signal)
2. LEFT and RIGHT SEGMENTS. Once a *steptime* was sampled, likely *left* and *right* segment amplitudes were sampled near the mean of the signal on either side of the step:

$$left \sim Gaussian(\text{mean(signal left of } steptime), \text{std(signal left of } steptime))$$

$$right \sim Gaussian(\text{mean(signal right of } steptime), \text{std(signal right of } steptime))$$

3. NOISE, σ. The noise level was sampled as in the ramp model, $\sigma \sim Beta(\text{α},\beta)$, except α,β were the parameterization of a Beta distribution with mode equal to the standard deviation of the signal left of *steptime*.

After model initialization for each Trace, 50 rounds of Importance Resampling of the hierarchical model were then conducted, each time randomly generating ramp or step hypotheses from the proposal heuristics. On each round, the best fitting hypothesis was retained, such that each of the 20 Trace iterations of model classification returned one optimized model from the 50 rounds of Importance Resampling.

The probability of the model class for each single-trial was then defined as the proportion of the 20 Trace iterations that found the optimal model to be derived from that model class (e.g., if the model returned 15 step-fits and five ramp-fits, the p(ramp) was 0.25). Examples of the 20 Trace iterations for two sample trials are shown in *Figure 6—figure supplement 2B*.

To determine whether the step model detected step-functions in the GCaMP6f dataset, the step model was inferred alone to find step-fits for every trial, and single-trial signals were realigned to the optimal *steptime* (GCaMP6f, tdTomato, EMG, *Figure 6—figure supplement 4A-B*).

## Single-trial dynamics analysis with geometric modeling ('Multiple threshold modeling')
The multi-threshold procedure described above was also employed to determine whether single-trial ramping dynamics were more consistent with a continuous ramp *vs.* discrete step dynamic on single-trials. The threshold-crossing time for each trial was regressed against its first-lick time, and the slope of this relationship was reported, as well as the variance explained.

## Single-trial variance analysis for discrete step dynamics
For discrete step single-trial dynamics to produce ramping on average, the time of the step across trials must be distributed throughout the trial interval (importantly, a peri-motor step occurring consistently just before first-lick cannot give rise to continuous ramping dynamics on average). As such, the variance in the GCaMP6f signals across trials for similar first-lick times should be minimal near the time of the cue (when few trials have stepped) *and* near the time of the first-lick (when all of the trials have stepped). This predicts an inverted-U shaped relationship of signal variance across trials *vs.* position in the timing interval.

To compare variance across trials equitably, trials were first aligned to the cue and pooled by first-lick time in pools of 1 s each (1–2 s, 2–3 s, etc.), truncated at the earliest first-lick time within the pool. The variance in GCaMP6f signals across trials within a pool was quantified in 10% percent increments of time from the cue up to the earliest first-lick time in the pool (i.e., 1–2 s pool truncated at 1 s,

divided into 100 ms increments). Measuring variance by percent of elapsed time within pool allowed pooling of trials across the entire session. The shape of the variance *vs.* percent of timed interval elapsed was compared to the inverted-U shape prediction to assess for discrete step dynamics.

## Optogenetics—determining the physiological range for activation experiments

To test whether optogenetic manipulations during the self-timing task were in the physiological range, we assessed the magnitude of the effect of activation on dopamine release in the DLS by simultaneous photometry recordings with optical activation (*Figure 7—figure supplement 2*). In two DAT-cre mice, we expressed ChrimsonR bilaterally in SNc DANs and the fluorescent dopamine indicator dLight1.1 bilaterally in DLS neurons. SNc cell bodies were illuminated bilaterally (ChrimsonR 550 nm lime or 660 nm crimson, 0.5–5 mW) on 30% of trials (10 Hz, 10 or 20 ms up-time starting at cue onset and terminating at first-lick). dLight1.1 was recorded with 35 μW 475 nm blue LED light at DLS. To avoid crosstalk between the stimulation LED and the photometry recording site, the brief stimulation up-times were omitted from the photometry signal and the missing points filled by interpolation between the adjacent timepoints.

In a few preliminary sessions, we also explored whether we could evoke short-latency licking (i.e., within a few hundred milliseconds of the stimulation) if light levels were increased above the physiological range for DAN signals. Rather than eliciting immediate licking, higher light levels produced bouts of rapid, nonpurposive limb and trunk movements throughout stimulation, and task execution was disrupted. The animals appeared to have difficulty coordinating the extension of the tongue to touch the lick spout. Simultaneous DLS dopamine detection showed large, sustained surges in dopamine release throughout the period of stimulation, with an average amplitude comparable to that of the reward transient (*Figure 7—figure supplement 2*, right). This extent of dopamine release was never observed during unstimulated trials. Consequently, to avoid overstimulation in activation experiments, we kept light levels well below those that generated limb and trunk movements.

## Optogenetics—naive/expert control sessions

To determine whether optogenetic stimulation directly elicited or prevented licking, licking behavior was first tested outside the context of the self-timed movement task on separate sessions in the same head-fixed arena but with no cues or behavioral task. Opsin-expressing mice were tested before any exposure to the self-timed movement task ('Naive') as well as after the last day of behavioral recording ('Expert'). In ChR2 control sessions, stimulation (5 mW 425 nm light, 3 s duration, 10 Hz, 20% duty cycle) was applied randomly at the same pace as in the self-timed movement task. stGtACR2 control sessions were conducted similarly (12 mW 425 mW light, 3 s duration, constant illumination); but to examine if inhibition could block ongoing licking, we increased the baseline lick-rate by delivering juice rewards randomly (5% probability checked once every 5 s).

## Optogenetics—self-timed movement task

SNc DANs were optogenetically manipulated in the context of the 3.3 s self-timed movement task. To avoid overstimulation, light levels were adjusted to be subthreshold for eliciting overt movements as described above, and mice were not stimulated on consecutive days.

### Activation

SNc cell bodies were illuminated bilaterally (ChR2: 0.5–5 mW 425 nm blue LED light; ChrimsonR 550 nm lime or 660 nm crimson) on 30% of trials (10 Hz, 10 or 20% duty cycle, starting at cue onset and terminating at first-lick). DAN terminals in DLS were stimulated bilaterally via tapered fiber optics on separate sessions.

### Inactivation

SNc cell bodies were illuminated bilaterally (stGtACR2: 12 mW 425 nm blue light) on 30% of trials (constant illumination starting at cue onset and terminating at first lick).

## Quantification of optogenetic effects

The difference in the distribution of trial outcomes between stimulated and unstimulated trials on *each session* was quantified in four ways.

1. 2-Sample Unsigned Kolmogorov-Smirnov Test.
2. Difference in empirical continuous probability distribution function (cdf). The difference in the integral of the stimulated and unstimulated cdf (dAUC) was calculated for each session from 0.7 to 7 s. Effect size was quantified by permutation test, wherein the identity of each trial (stimulated or unstimulated) was shuffled, and the distribution of dAUCs for the permuted cdfs was calculated 10,000x. Results were reported for all sessions.
3. Difference in mean movement time. Movement times on stimulated and unstimulated trials were pooled and the distribution of movement time differences was determined by non-parametric bootstrap, in which a random stimulated and unstimulated trial were drawn from their respective pools 1,000,000x, and the difference taken. The mean of each session's bootstrapped distribution was compared across sessions by the 1,000,000x bootstrapped difference of the mean between sessions of different categories.
4. Difference in median movement time. Same as above but with median.

## Single-trial probabilistic movement-state decoding model

The probability of transitioning to a movement state, $s_t = 1$, at time $= t$ was decoded with a logistic generalized linear model of the form:

$$p(s_t = 1) = \text{logit}(bX_t)$$

where $X_t$ is a vector of predictors for the timepoint, $t$, and $b$ is the vector of fit coefficients. The vector of predictors was comprised of the GCaMP6f signal at every timepoint (the current time, $t$) as well as the signal history, represented as 200 ms-wide signal averages moving back in time from $t$. Previous trial history (n-1[th] and n-2[th] first-lick times and reward/no-reward outcomes) did not contribute significantly to the model during model selection and were thus omitted (see Model Selection, below).

Movement state, $s_t$, was defined as a binary variable, where $s_t=0$ represented all timepoints between the cue up until 160 ms before the first-lick detection (to exclude any potential peri-movement responses), and $s_t=1$ represented the timepoint 150 ms before the first-lick. Because there were many more $s_t=0$ than $s_t=1$ samples in a session, $s_t=0$ points were randomly down-sampled such that states were represented equally in the fit. To avoid randomly sampling a particular model fit by chance, each dataset was fit on 100 randomly down-sampled (bootstrapped) sets, and the average fit across these 100 sets was taken as the model fit for the session.

GCaMP6f signals were smoothed with a 100 ms gaussian kernel and down-sampled to 100 Hz. The GCaMP6f predictors were then nested into the model starting with those furthest in time from the current timepoint, $t$:

Nest 0: b0 (Null model)
Nest 1: b1 = b0 + mean GCaMP6f 1.8:2.0 s before current time = $t$
Nest 2: b2 = b1 + mean GCaMP6f 1.6:1.79 s before current time = $t$
Nest 3: b3 = b2 + mean GCaMP6f 1.4:1.59 s before current time = $t$
Nest 4: b4 = b3 + mean GCaMP6f 1.2:1.39 s before current time = $t$
Nest 5: b5 = b4 + mean GCaMP6f 1.0:1.19 s before current time = $t$
Nest 6: b6 = b5 + mean GCaMP6f 0.8:0.99 s before current time = $t$
Nest 7: b7 = b6 + mean GCaMP6f 0.6:0.79 s before current time = $t$
Nest 8: b8 = b7 + mean GCaMP6f 0.4:0.59 s before current time = $t$
Nest 9: b9 = b8 + mean GCaMP6f 0.2:0.39 s before current time = $t$
Nest 10: b10 = b9 + GCaMP6 f signal at current time = $t$

Nesting the predictors from most distant in time to most recent permitted observation of the ability of more proximal signal levels to absorb the variance contributed by more distant signal history.

The fitted hazard function was then found as the average probability of being in the movement state across all trials in the session as calculated from the average model fit. Because $s_t = 0$ states were significantly downsampled during fitting, this rescaled the fit hazard. Thus, to return the fit hazard to the scale of the hazard function calculated from the behavioral distribution, both the fit hazard and true hazard function were normalized on the interval (0,1), and the goodness of fit was assessed by $R^2$ comparison of the fit and true hazard functions. This metric was similar between individual session fits as well as the grand-average fit across all animals and sessions.

To guard against overfitting, this procedure was repeated on the same datasets, except the datasets were shuffled before fitting to erase any non-chance correlations between the predictors and the predicted probability of being in the movement state.

## Model selection

To evaluate the contribution of task performance history to the probability of being in the movement state at time = $t$, we could not observe every timepoint in the GCaMP6f trial period timeseries as we did in the final model, because the trial history for a given timepoint was the same for all other points in the trial; hence this created bias, because the movement state $s_t = 1$ was represented for all trials, but the likelihood of the a trial's 0 state being represented after down-sampling was dependent on the duration of the trial (i.e., first-lick time). Consequently, model selection was executed on a modified version of the model that ensured that each trial would only be represented *one time at most* in the fit. Because this greatly reduced the power of the model, model selection was conducted on sessions from the two animals with the highest S:N ratio and most trials to ensure the best chance of detecting effects of each predictor (*Figure 8—figure supplement 1*).

The set of permutations of GCaMP6f signal and task history were fit separately, and the best model selected by BIC (though notably AIC and AICc were in agreement with the BIC selection). Each model was fit in 'time-slices'—windows of 500 ms from the time of the cue up until the first-lick. Only one point for each trial was fit within this window to ensure the movement state within the window was uniquely represented. For each time-slice model, the GCaMP6f signal for each trial was thus averaged within the time-slice window, and the movement state was taken as 1 only if the movement state occurred sometime within the window. The model fit for a session was taken as the average model fit across each of the time-slices. Notably, a time-slice required a sufficient number of trials to be present (either in the $s_t=0$ or terminating in the movement state $s_t=1$) for the fit to converge; once the first-lick occurred for a trial, it did not contribute data to later time-slices. The source data files for *Figure 8—figure supplement 1* contain plots of all time-slice coefficient fits, including for models with insufficient numbers of trials to converge.

## Code availability

All custom behavioral software is available at https://github.com/harvardschoolofmouse/HSOMbehaviorSuite (*Hamilos, 2021a*, copy archived at swh:1:rev:5a9b981afb658cfc-05277b1a257e1733f274c9a2). All custom analysis tools are available with sample datasets at https://github.com/harvardschoolofmouse/eLife2021 (*Hamilos, 2021b*, copy archived at swh:1:rev:fa4b2cdfd6d6a55b82124f86d2599faebccd4eee).

## Acknowledgements

We thank D Chicharro, JG Mikhael, SJ Gershman, J Drugowitsch, K Reinhold, S Panzeri, R Bliss and EN Brown for discussions on analytical methods; J Tenenbaum, C Wong, and A Lew for their instruction and advice pertaining to probabilistic programming in Gen; V Berezovskii, J LeBlanc, T LaFratta, O Mazor, P Gorelik, J Markowitz, A Lutas, KW Huang, L Hou, SJ Lee and NE Ochandarena for technical assistance; and B Sabatini, C Harvey, SR Datta, M Andermann, L Tian and W Regehr for reagents. The work was supported by NIH grants UF-NS108177 and U19-NS113201, and NIH core grant EY-12196. AEH was supported by a Harvard Lefler Predoctoral Fellowship and a Stuart HQ & Victoria Quan Predoctoral Fellowship at Harvard Medical School. The funders had no role in study design, data collection and interpretation, or the decision to submit the work for publication.

## Additional information

### Competing interests

John A Assad: co-founder of OptogeniX, which produces the tapered optical fibers used in some experiments. The other authors declare that no competing interests exist.

## Funding

| Funder | Grant reference number | Author |
|---|---|---|
| National Institutes of Health | UF-NS108177 | John Assad |
| National Institutes of Health | U19 NS113201 | John Assad |
| National Institutes of Health | EY-12196 | John Assad |
| Lefler Predoctoral Fellowship | | Allison E Hamilos |
| Stuart H.Q. and Victoria Quan Predoctoral Fellowship | | Allison E Hamilos |

The funders had no role in study design, data collection and interpretation, or the decision to submit the work for publication.

## Author contributions

Allison E Hamilos, Conceptualization, Data curation, Formal analysis, Funding acquisition, Investigation, Methodology, Project administration, Software, Supervision, Validation, Visualization, Writing – original draft, Writing – review and editing, performed all experiments, wrote the original software, trained, mentored and supervised G.S. and Y.H; Giulia Spedicato, Investigation, assisted with experiments using tapered fiber optics; Ye Hong, Investigation, assisted with optogenetic no-opsin control experiments; Fangmiao Sun, Resources, developed the dopamine sensor, DA2m, developed the dopamine sensor, DA2m; Yulong Li, Resources, developed the dopamine sensor, DA2m, developed the dopamine sensor, DA2m; John A Assad, Conceptualization, Funding acquisition, Methodology, Resources, Supervision, Writing – original draft, Writing – review and editing

## Author ORCIDs

Allison E Hamilos http://orcid.org/0000-0001-9486-0017
John A Assad http://orcid.org/0000-0002-7689-5336

## Ethics

All experiments and protocols were approved by the Harvard Institutional Animal Care and Use Committee (IACUC protocol #05098, Animal Welfare Assurance Number #A3431-01) and were conducted in accordance with the National Institutes of Health Guide for the Care and Use of Laboratory Animals. Surgeries were conducted under aseptic conditions with isoflurane anesthesia, and every effort was taken to minimize suffering.

## Decision letter and Author response

Decision letter https://doi.org/10.7554/eLife.62583.sa1
Author response https://doi.org/10.7554/eLife.62583.sa2

## Additional files

### Supplementary files

• Transparent reporting form

### Data availability

All datasets supporting the findings of this study are publicly available (DOI: 10.5281/zenodo.4062749). Source data files have been provided for all figures.

The following dataset was generated:

| Author(s) | Year | Dataset title | Dataset URL | Database and Identifier |
|---|---|---|---|---|
| Hamilos AE, Spedicato G, Hong Y, Assad JA | 2020 | Original single session datasets from "Slowly evolving dopaminergic activity modulates the moment-to-moment probability of reward-related self-timed movements." | https://doi.org/10.5281/zenodo.4062749 | Zenodo, 10.5281/zenodo.4062749 |

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
