## [Editor Report]

Dopamine loss in Parkinson's disease results in impaired movement initiation and execution, but the precise relationship between dopamine activity and the decision to move is poorly understood. Here, the authors imaged mesostriatal dopamine signals as head-fixed mice decided when, after a cue, to retrieve water from a spout. Surprisingly, ramps in dopamine activity predicted, even on single trials, the timing of licks. Fast ramps preceded early retrievals; slow ones preceded late ones. Optogenetic activation or suppression of dopamine activity accelerated or delayed lick initiation, respectively. Together, these findings reveal strong links between ramps in dopamine activity and the timing of self-initiated movement.

---

## [Decision Letter]

**Decision letter after peer review:**

[Editors’ note: the authors submitted for reconsideration following the decision after peer review. What follows is the decision letter after the first round of review.]

Thank you for submitting your work entitled "Slowly evolving dopaminergic activity controls the moment-to-moment decision of when to move" for consideration by *eLife*. Your article has been reviewed by 3 peer reviewers, including Jesse H Goldberg as the Reviewing Editor and Reviewer #1, and the evaluation has been overseen by a Senior Editor.

Our decision has been reached after consultation between the reviewers. Based on these discussions and the individual reviews below, we regret to inform you that your work will not be considered further for publication in *eLife*.

All three reviewers appreciated the potential novelty that even baseline dopamine levels could be predictive under the task conditions of when the animal will move. However, as detailed in the reviews below, there were critical concerns regarding the animal's performance in the task, the novelty of pre-movement dopamine ramps relative to prior work, and the single-trial analyses which were not sufficient to convincingly demonstrate that ramps did not result from artifact of averaging across trials with different event timing. Ultimately, it was determined that even if the authors addressed the relationship between ramps and movement on single trials, issues with the behavioral performance and task design make it difficult to infer how signals correspond to internal processes such as reward anticipation, RPE, behavioral inhibition, or – as the manuscript suggests – timing.

*Reviewer #1:*

Hamilos et al. image DA axonal and neuronal CA signals in mice engaged in a self-timed lick task. They observed a ramping signal prior to lick onset that could predict, on average when a mouse would lick. While ramps in DA Ca signals have been observed as animals locomote towards known rewards (e.g. Howe et al., Hamid et al., Mohebi et al., all cited) and while DA activity is known to exhibit correlations with movement onsets – what's new and potentially important in this paper is how well the DA ramps could predict movement onset time – even on single trials. Complementing this finding were causal experiments where photoactivation or inhibition on single trials could promote or delay movement initiation. Given the amount of effort invested into the relationship between DA activity and upcoming movements – it is surprising that the strong correlation between the DA ramp and movement initiation (or decision, see discussion of Guru et al., paper below) hasn't been so clearly observed before (to my knowledge). This paper links DA activity to the timing of a reward-related decision extremely well.

Figure 3 of Guru et al. (unpublished biorxiv paper from Warden lab, not cited) shows a similar result as this paper except the internally timed action is not to produce movement but rather to terminate movement (stop running on a wheel). The fact that the same ramp is observed in both of these conditions undermines the connection that the authors make about ramping DA and decision to move. Taken together, it seems the DA ramps are more about the timing of a self-paced decision (whether it be to start or to stop doing something) than about movement initiation. The authors may want to re-tweak the interpretations of this paper to allow for this more general perspective.

*Reviewer #2:*

In this study Hamilos and colleagues measured dopamine signals in mice performing a self-timing task. They took advantage of the variability in the first lick time to study how dopamine signaling preceding the first lick varied with the timing of the first lick. While a number of studies have reported short-latency (<500 ms) increases in dopamine activity immediately preceding movements, the authors make a number of novel findings. First, dopamine photometry signals ramped up over several seconds preceding the first lick. Second, the steepness of this ramping, even the amount of baseline signal predicted the first lick time. Third, optogenetic activation and inhibition reduced or increased the first lick time. The authors are commended for performing several different types of photometry experiments. The authors conclude that dopaminergic signals unfolding over seconds control the timing of movements, but not as much the ability to move itself. While the work is novel and adds an interesting perspective on the function of dopaminergic neurons, there were concerns about some of the evidence for the first two claims, as well as insufficient detail in some of the statistical analysis that make it difficult to fully judge the paper's merits. The main concerns are detailed below.

1. Lines 151-153: "we observed systematic differences in the steepness of ramping that were highly predictive of movement timing (Figure 1D-E)." The reviewer agrees this is quite evident from the ramping curves, but still the authors should formally show the fact that the steepness (i.e., slope) of ramping is predictive of timing. The decoding model in Figure 6 goes some way toward this goal, but it seems Figure 6 uses time to threshold rather than the slope. It would be worthwhile to check in Figure 1 or Figure 2 whether the time to first lick is negatively correlated with slope of ramping.

2. Figure 4: similar to the previous comment. The reviewer found it difficult to interpret Figure 4 and suggests including a more traditional analysis, such as a Pearson correlation between the time to first lick and the level of pre-cue baseline dopamine.

3. Figure 6: the reviewer was confused about the contribution of the pre-cue baseline signal (predictor # 7) to the nested model. From the results in Figure 6B and 6C it appears that the pre-cue signal has almost zero contribution to the model (it was the only predictor with a non-significant weight in 6B). But this seems to contradict the authors' claim that "higher pre-cue, baseline DAN signals are correlated with earlier self-timed movements." If baseline signals do indeed correlate with timing of movements, shouldn't the weight of predictor # 7 be higher? This reviewer may have misunderstood some important detail, so some clarification of this issue in the text would be helpful. The reviewer also requests clarification of the difference between the "offset" (Predictor # 0) and "pre-cue" (Predictor # 7) as they both seem to be referring to some sort of baseline signal.

4. In several figure panels (including 6B, 7C, 7D) stats are either missing or details are too sparse to fully evaluate the significance of the results. The authors are requested to include information about sample size, type of test used, and if possible the exact p value.

*Reviewer #3:*

Assad and colleagues examined nigrostriatal dopamine signals in head-fixed mice performing a licking task. Reward was available if mice first withheld licking for several seconds after a cue. They observed a ramping increase in dopamine before the time of first lick, that stretched in proportion with this hold interval. Optogenetic increases/decreases of dopamine caused licking to be earlier/later. They conclude that dopamine ramps are involved in controlling the timing of movements.

There are several interesting aspects of this work, including the comparison between different DA signals (Figure 2) and the ramps themselves. Based on prior work in the field I have no problem believing most of the title "…dopaminergic activity controls the moment-to-moment decision of when to move". The problem is that the aspects of this work that are novel are either not well supported by data, or rely on questionable analyses and interpretations. There are also serious problems with the behavioral task and the quality of some recordings.

Main points:

1) The mice are notably bad at the behavioral task. Based on Figure 1B it is rare for them to wait long enough to get the reward. They do wait longer in the 5s condition compared to 3.3s, just not long enough. This general inability to perform the task casts further doubt on what internal representations might be driving dopamine signals and behavior. I'm not sure it's reasonable to describe the behavior as "timing", if the timing is usually wrong. And given that they see the same ramps before spontaneous licks outside the task, the task seems largely irrelevant.

2) Optogenetic manipulation of dopamine made movements more or less likely, as expected from prior studies. But the authors claim more than this – that the optogenetic 'effects were expressed through a "higher-level" process related to the self-timing of movement'. I did not find this claim convincing, even beyond the issue that the mice are very poor at timing. If an action is prepotent in a given behavioral context as the result of training, then increasing dopamine may increase the likelihood of that action being emitted. Giving even more dopamine (higher laser power) may lower the threshold for *any* action being emitted. But this doesn't demonstrate anything much about timing per se.

3) A key claim is that the dopamine signals are "slowly-evolving". This makes it essential to define "slow". In the dopamine field "slow" or "tonic" has often been used to referred to microdialysis signals presumed to change over tens of minutes. Here the authors describe ramping processes that may complete over several seconds, or be done in less than a second. So the kinetics of the signal seem more linked to the (varying) speed of behavioral transitions than to any inherently "slow" process.

4) Furthermore, in the Discussion the authors describe the instantaneous state of the dopamine signal at trial onset as "tonic", which seems like a mistake: they demonstrate that this signal has been immediately "reset" and cares more about the upcoming behavior than the immediately preceding trial just a few seconds earlier. This is not how "tonic" is used in the dopamine literature (e.g. Niv/Daw/Dayan describing tonic dopamine as integrating reward rate over prior recent trials).

5) The ramp-like pattern is interesting and may indeed be comparable to dopamine ramps previously reported in other tasks. But ramps can arise as an artifact of averaging across trials in which events have different timing. To avoid this problem the authors examined single-trials (Figure 6). But the analysis employed does not actually measure ramping on single trials. Instead, they examine the timing of crossing an artificial threshold, and they see that this threshold crossing time predicts lick time well when the threshold is high. This seems equivalent to noting that dopamine increases shortly before licking, which we already knew from movement-aligned averages; it doesn't seem to demonstrate the point the authors wish to make.

6) The authors note correctly that the signals they observe "could reflect the superposition of dopaminergic responses to multiple task events, including the cue, lick, ongoing spurious body movements, and hidden cognitive processes like timing." But I wasn't convinced that the regression models provided much insight into those hidden processes. Adding a "stretch" parameter felt merely like a descriptive fit to the observed data than a process-based model.

7) It is not uncommon to see neural activity evolving between cues and movements in a manner that scales with the interval between these events (e.g. Renoult et al. 2006, Time is a rubberband: neuronal activity in monkey motor cortex in relation to time estimation). It is interesting that dopamine can do something similar, but that doesn't seem to support a special role for dopamine.

8) The figures are not well organized at all. Figs1 and 2 seem partly redundant, and are often referred to together. Figure 3 is mentioned only in passing to say that an accelerometer was present, without describing the Figure 3 results or why they are important (perhaps should be a supplemental figure after clarification).

9) line234: "(highest S:N sessions plotted for clarity, 4 mice, 4-5 sessions/mouse, 17 total)". This seems weird and concerning. If signals were good enough to use for other analyses, why not this one?

10) Fig6, what does it mean that the tdt (control) signal is also a significant predictor of first-lick time? This seems like a serious problem for a control signal.

[Editors’ note: further revisions were suggested prior to acceptance, as described below.]

Thank you for choosing to send your work entitled "Slowly evolving dopaminergic activity controls the moment-to-moment decision of when to move" for consideration at *eLife*. Your letter of appeal has been considered by a Senior Editor and a Reviewing editor, and we are prepared to consider a revised submission with no guarantees of acceptance.

This resubmission may go back to all of the same and/or new reviewers. To ensure your revision has the best chance possible, we encourage you to well address each of the prior concerns in both the manuscript and in the response to review. Below are a few key concerns that we think are especially important for you to address.

1. It will be key to nail down the legitimacy of the claim that dopamine ramps and amplitudes at movement onsets on single trials. The main concerns (points 1-4) of R2 and point 5 of R3 well-articulate this concern.

2. Well justify and better explain the task (as was partly done in the appeal). The idea of the dopamine signal as playing any role in 'timing' was problematic because there was not a stringent control that timing was necessary in the task.

We encourage you to clarify that the task was not designed to specifically rule in or out the many models of what dopamine encodes and rather clarify that the goal of the study is to identify dopamine relationship to upcoming movements or behavioral transitions. Fully addressing R3 comment 1 is necessary.

We also encourage you to address the statement that ramps existed before licks outside the task, which shows the irrelevance of the task for the finding. Showing examples of this phenomenon might help. And we encourage you to consider how this finding relates to the central claim of 'timing.'

Please be aware that the revision requirements outlined above, along with your response to these comments, will be published should your article be accepted for publication, subject to author approval. In this event you acknowledge and agree that the these will be published under the terms of the Creative Commons Attribution license.

[Editors’ note: further revisions were suggested prior to acceptance, as described below.]

Thank you for resubmitting your article "Slowly evolving dopaminergic activity modulates the moment-to-moment probability of movement initiation." for consideration by *eLife*. Your revised article has been reviewed by 3 peer reviewers, including Jesse H Goldberg as the Reviewing Editor and Reviewer #1, and the evaluation has been overseen by Kate Wassum as the Senior Editor.

Essential revisions:

1) Please address this concern regarding 'baseline" predictors: The authors have made significant improvements in describing the task, but there remained a lack of information about behavior in the lamp-off period before the cue was turned on (the baseline). The main concern is that this delay period is technically not a baseline because the lamp-off serves as an additional predictive cue. The authors tried to prevent this by randomizing the delay time (0.4-1.5 s), but this range looks comparable or even smaller than the variability in first-lick time shown in Figure 2B. They also did a control experiment without a lamp and showed qualitatively similar performance, but I didn't find the data very convincing (for some reason the data shown in figure supplement 1C does not have clear peaks at 3.3 s). Knowing whether this is a real baseline is important because the authors set cue-on as t=0, and do not consider behavior prior to that. This would not be a big concern if the authors can show that there is no statistically significant relationship between the lamp-off to cue-on delay and the lick onset on that trial. Otherwise, the concern is that their finding that higher baseline fluorescence predicts earlier licks may have a trivial explanation.

2) Address comments re: Figure 7, where there is still some confusion about the description of the sample size. For example, line 339 lists 12 mice but I think the correct variable is number of trials. The text should thus include the size of the actual variable used in the plots (listing the number of mice as well is ok, but this is not sufficient on its own). Furthermore, the results would more convincing if this figure included single-animal comparisons rather than just pooled data or across-session comparisons. Basically, could the authors please include a panel similar in style to Figure 7C but where individual points represent single animals, not sessions? This would be more consistent with other optogenetic-behavioral studies in rodents.

3) Please better address an issue with interpretation of results, specifically regarding movement timing versus reward expectation: The authors claim that their results mean a causal role for the level of dopamine ramps in modulating the probability of action initiation, but that interpretation seems strange given that these ramps are often observed when animals are already moving (Howe et al., 2013; Hamid et al., 2016; Engelhard et al., 2019; Kim et al., 2020; Guru et al., 2020). In common with all these papers, dopamine ramps seem to be related to the temporal prediction of when reward will be available. This was shown nicely by Kim et al., 2020, where they showed that regardless of the movements performed by the mice, ramps are elicited in relation to when the animal expects the rewards to arrive (see especially their moving bar experiments). In the present work, this interpretation is also consistent with the results (and more consistent with previous works): when animals decide to move early, the DA system receives this information and that results in a more steeply rising ramp, while when animals decide to move late, the same thing occurs, and the ramp rises more slowly. The issue at hand is that the initiation of movement (licking) is strongly correlated with reward delivery (or with the time of expected reward delivery in case of failed trials) and thus it is not possible to distinguish between these interpretations. So, in summary, the authors should address the concern that the interpretation of dopamine ramps modulating the moment-to-moment probability of action initiation is unwarranted. Results should be reframed to make it clear that this claim cannot be supported given the possible alternative explanations (such as the one suggesting that DA ramps reflect a prediction of the time of upcoming reward delivery, which is more consistent with the previous literature).

Alternatively, if the authors can find an experimental way to dissociate the expected time for reward delivery from the moment of action initiation, then that would be one way to make a decisive conclusion. For example, the water dispensation could occur at a fixed time relative to the cue – even as the mouse can initiate licking whenever it wants. If DA dynamics still predict initiation time, then the authors' perspective is supported. Yet if the DA dynamics no longer predict initiation time, and instead simply ramp to the reward time, then reviewer 4's perspective is supported. Either result would be interesting. Performing these experiments would be ideal, but not necessary for publication. If the authors do not wish to conduct new experiments along these lines, they should – throughout the manuscript – modify their text to include this alternative interpretation (reward expectation) – or offer in rebuttal a convincing argument as to why the logic of reviewer 4's (and the BRE's) thinking is not sound.

[Editors’ note: further revisions were suggested prior to acceptance, as described below.]

Thank you for submitting your article "Slowly evolving dopaminergic activity modulates the moment-to-moment probability of movement initiation." for consideration by *eLife*. Your article has been reviewed by 3 peer reviewers, including Jesse H Goldberg as the Reviewing Editor and Reviewer #1, and the evaluation has been overseen by Kate Wassum as the Senior Editor.

Essential revisions:

1) 2/3 reviewers were not persuaded by the author's rebuttal arguments that DA ramps reflect movement initiation and not reward expectation, resulting in an enduring concern that the main message of the paper, evident even in the title, is not watertight. All reviewers still think the paper's findings are timely and important – so a tempering / adjustment of the claims is a reasonable path forward to publication. The change in 'pitch' of this paper would need to include changes to title, abstract, and discussion. While some specific recommendations are included in the full reviews pasted below, the authors are in the best position to truly internalize reviewer's four's sound arguments to transform the message of the paper into one that allows for reward expectation to be the variable that DA ramps reflect, that optogenetic experiments manipulate, and that ultimately affects movement timing.

If you have not already done so, please ensure that your manuscript complies with *eLife*'s policies for statistical reporting: https://reviewer.elifesciences.org/author-guide/full "Report exact p-values wherever possible alongside the summary statistics and 95% confidence intervals. These should be reported for all key questions and not only when the p-value is less than 0.05."

*Reviewer #1:*

I am satisfied with the responses to points 1 and 2.

I am not completely persuaded by the author's arguments in revision point 3. First, their argument starts with the statement that "dopamine ramps have not, in fact, been observed in relation to passive temporal expectation of reward – except when some kind of external sensory indicator of proximity to reward was provided." They further argue that DA ramps have not been observed in Pavlovian conditioning – citing many classic Schults papers and more recent rodent work. But Figure 3b of Fiorillo, Tobler, Schultz, Science 2003 clearly shows ramps when rewards are uncertain even in a Pavlovian task. The first section of their argument was based on the absence of ramps in Pavlovian tasks, so this falls apart on scrutiny. There is more going on here than the authors are noting. Also, it's strange that in discussion the authors invoke an 'operant' nature of their task – based on the idea that the animal must move to earn reward. With this definition one wonders what constitutes a non-operant task – even the classic Schultz papers on DA signals during Pavlovial conditioning require tongue extension to retrieve water.

In revision, I recommend the following:

1) In abstract, the word 'controls' (line 20) seems potentially overstated. Please change to 'predicts' or 'is associated with'

2) In Discussion, line 456, I think the authors should include an expanded (~2-3 sentence) consideration that their signals that predict movement are expected value signals – essentially including Rev 4's interpretation as a plausible one.

Overall, this is an important paper suitable for *eLife*.

*Reviewer #2:*

The authors have addressed all my comments and I have no further concerns.

*Reviewer #4:*

I don't find the authors response to my comment satisfactory. Given that the paper title is "Slowly evolving dopaminergic activity modulates the moment-to-moment probability of movement initiation" and that if published as is, this would constitute peer-reviewed evidence that dopamine ramps modulate movement timing, the relationship between the ramps and movement initiation should be ironclad. Yet from the presented data as well as the authors comments, it still seems to me more likely that these ramps represent an internal measure of reward expectation/ expected time of reward delivery. Specifically, the authors state (in bold font) that "dopaminergic ramps have not, in fact, been observed in relation to passive temporal expectation of reward-except when some kind of external sensory indicator of proximity to reward was provided". However, that is not the case, see for example the science paper by Fiorillo, Tobler and Schultz, 2003, as well as their follow-up paper in 2005. Thus, we have evidence that these ramps occur A- when animals are moving (see all the references in my previous comment). B- when animals are not moving and don't intend to, where the time to reward is signaled to them externally (kim et al.). C- When animals are not moving and don't intend to, where the time to reward is not signaled to them externally (Fiorillo et al). D- when animals are not moving but do intend to (present manuscript). Given all these conditions where we see ramping, ascribing them to movement initiation does not seem warranted, especially when we have a parsimonious alternative explanation (Reward expectation).

It may seem at first glance that the optogenetic experiments performed by the authors would counter my point, because they had an effect in modulating the timing of movement initiation. However, these manipulations could just be changing the motivation or reward expectation of the animals, which would make them move earlier or later. The authors agree with this, and then gave the following strange (to me) response: "…reward expectation may be the very force that propels movement in our task. In this view, reward expectation is intrinsically intertwined with movement initiation. Our point is that whatever the dopaminergic signal is tracking, it can be harnessed to influence the probability of movement onset."

This point seems very strange to me because under this view, any type of brain signal related to any external variable that ends up modulating an animal's movement would be ascribed as a movement signal. For example, sensory signals in visual cortex in a perceptual task would be classified as movement initiation signals, because they end up making the animal move (e.g. if we inhibit visual cortex the animals can't perceive the stimulus and won't move correctly). The whole point of claiming that a particular neural signal is related to movement initiation is that it is NOT related to other signals: sensory, reward or others, even as those other variables typically have an end effect of modulating an animal's movement. To take this point further, behavior itself is ultimately about generating movements, and so any behaviorally relevant signal will ultimately affect movements. The authors' claim here is much more specific, and in my view not supported.

To conclude, I'm uncomfortable with the paper keeping its current title and conclusions. It seems to me much more correct to conclude that dopamine ramps reflect reward expectation, and that (of course!) reward expectation can end up influencing movements. This is much more in line with all the previous literature of dopamine and I don't think the authors made a compelling case of abandoning this view. To reiterate what I previously wrote, I do think this is a good paper that should be published, but not under the current framing. I would suggest the following title: "Slowly evolving dopaminergic activity reflects reward expectation and can modulate the moment-to-moment probability of movement initiation". I know it is not as catchy, but I think it's much more correct given what we know, and I don't want to give my stamp of approval to a conclusion which I think is unwarranted.

[Editors’ note: further revisions were suggested prior to acceptance, as described below.]

Thank you for resubmitting your work entitled "Slowly evolving dopaminergic activity modulates the moment-to-moment probability of reward-related self-timed movements" for further consideration by *eLife*. Your revised article has been evaluated by Kate Wassum (Senior Editor) and a Reviewing Editor.

The manuscript has been improved but there are some remaining issues that need to be addressed, as outlined below:

Your revised manuscript does not fully comply with *eLife*'s requirements for statistical reporting. Please ensure all statistics are included in full in the main manuscript. "Report exact p-values wherever possible alongside the summary statistics and 95% confidence intervals. These should be reported for all key questions and not only when the p-value is less than 0.05." more details can be found here:

https://reviewer.elifesciences.org/author-guide/full

---

## [Author Response]

[Editors’ note: The authors appealed the original decision. What follows is the authors’ response to the first round of review.]

We will look forward to receiving a revised article and a file describing the changes made in response to the decision and review comments for further consideration.This resubmission may go back to all of the same and/or new reviewers. To ensure your revision has the best chance possible, we encourage you to well address each of the prior concerns in both the manuscript and in the response to review. Below are a few key concerns that we think are especially important for you to address.Editor point 1. Well justify and better explain the task (as was partly done in the appeal). The idea of the dopamine signal as playing any role in 'timing' was problematic because there was not a stringent control that timing was necessary in the task.

We addressed this point in our initial appeal, which we reproduce here in case the reviewers didn’t see the appeal. We have added a paragraph to the Results sections of the revised manuscript summarizing these arguments, as well as a new main-text figure (New Manuscript Figure 1) summarizing these results.

Reviewer 3 was critical of the animals’ performance on the self-timed movement task. On reading the reviewer’s comments, we immediately realized that we should have included more discussion of the rich empirical history of timing tasks to contextualize and justify the behavior that we observed in our self-timed movement task. We did not think we had room for this discussion in the manuscript, but this was clearly a mistake! In fact, the behavior of the animals in our specific task was much “better” than the reviewer realized, and our observations were absolutely in line with previous findings in timing tasks. Below, we offer what we believe is overwhelming evidence on this point.

However, before we outline this evidence, we stress that we did *not* intend to study precise timing behavior. Rather, we set out to determine whether dopaminergic activity could explain *when a movement would occur.* We focused on s*elf-timed* movements, i.e., movements that are *not* abrupt, temporally-stereotyped reactions to external cues (in contrast to standard stimulus-response paradigms used in the vast majority of studies of the motor system; see Lee and Assad (2003) for discussion of this key distinction)*.* We used self-timed movements as a strategy to induce animals to produce movements with *seconds* of variability (relative to a reference cue) from trial-to-trial – but we were not overly concerned whether the animal timed “accurately.” In fact, the high degree of variability in movement time from trial-to-trial (from ~1 s to >6 s post-cue) was precisely the handle we needed to address our question: what happens differently in the dopaminergic system when animals move *early* versus *late* relative to an initial “reset” cue? If we had omitted the initial cue and allowed the animals to self-initiate movements whenever they pleased, we would not have had a reference to align neural signals, and thus could not have categorized movements as “early” or “late”. Our ability to categorize the movement time relative to the cue allowed us to discover that dopaminergic signals were highly predictive of the movement time. (That is, movement *time* implies time *relative to something*.) In this light, early (unrewarded) trials were just as informative as late (rewarded) trials. Thus, the percentage of rewarded trials was not at stake in this manuscript – although we clearly need to explain this logic better.

Nevertheless, there is strong evidence that our animals did, in fact, time well, and their behavior was consistent with the well-established body of findings on timing in animals and humans. We didn’t present this evidence in the original manuscript because we thought it was tangential to our main points and would require too much space. However, we clearly confused the reviewers, so we have addressed these issues in detail in the revised manuscript.

We presume that Reviewer 3 was critical of the task performance because (1) the animals’ movement times were highly variable, and/or (2) the peak of the movement-time distributions preceded the criterion time. We’ll address both issues in turn.

1) Variability in movement time. An observation across timing tasks and species is that the distribution of estimated time intervals has substantial variability. Moreover, that variability (measured by standard deviation) *scales proportionally to the length of the timed interval*, such that rescaling one interval by the peak time of another will result in overlapping distributions (Author response image 1; human self-timing data). This property is formalized by Weber’s Law (σ = *k*T*), where the standard deviation of the distribution for a given subject is a product of the subject’s Weber fraction (*k*, an empirically measured quantity) and the criterion time, *T* (Gallistel and Gibbon, 2000). The 3.3 s and 5 s task distributions in our task exhibit this scalar property of timing, following Weber’s Law (New Manuscript Figure 1B). This observation alone provides evidence that the animals were timing.

2) Animals’ and humans’ distributions of timed intervals invariably tend to *anticipate* the criterion interval, even at the expense of reward. The degree to which the distributions anticipate the criterion interval depends on the specific timing task. Most common, time-interval reproduction tasks in animals can be divided into two categories: “peak-procedure” or “differential reinforcement of low response rates” (DRL). Some years ago, our lab introduced a third category into animal studies, “self-timed movement tasks,” which have key features in common with DRL tasks. We will examine these in turn:

**Author response image 1. sa2fig1:** Performance in timing tasks follows the scalar property. (**A**) Human response-time distributions for three different criterion times, 8, 12 and 21 s (Figure adapted from Allman et al., 2014 (Figures 2 and 3)*.;* data from (Rakitin et al., 1998)). (**B**) When the distributions in A are scaled by the time of the peaks, they closely overlap, demonstrating the scalar property canonical to timing tasks across species.

Peak procedure (Author response image 2): In peak-procedure timing tasks, subjects are allowed to respond *as often* as they want, but are only rewarded for the first movement after the criterion time has elapsed (relative to a start-timing cue). Like an impatient passenger pressing a door-close button in the elevator, subjects in these tasks begin responding substantially earlier than the criterion time, and their response frequency increases monotonically as the target time approaches. To define the timing performance of the subject, occasional probe trials are included in which reward is *not* given at the criterion time. On probe trials, subjects continue to respond beyond the criterion time, but their rate of responding gradually wanes. It is well-known that the peak response-frequency typically occurs at or near the criterion time in these tasks (see Author response image 2 for an example from mice and Author response image 1 for an example from humans). Importantly, given the broad timing distributions, there are still many premature responses in peak procedures, observed in all species, from rodents to birds to primates (Gallistel and Gibbon, 2000).

Although peak procedures generally reveal timing peaks at or near the criterion time, we chose not to use a peak procedure in our study for a crucial reason: timing in peak procedures is explicitly manifested as an accelerating sequence of motor responses; thus, the *responses themselves* could drive motor-related neuronal activity that could mimic or obscure a neural timing signal, such as a ramping timing signal.

Differential Reinforcement of Low Response Rates (DRL): Historically, the most common alternative to peak procedures for studying operant timing in rodents has been DRL tasks. In typical DRL tasks, rats or mice are rewarded if they wait to respond for at least a criterion time after their last response. If the animal responds prematurely, it is *not* rewarded, and the response/reward-clock resets (unlike peak procedures).

Self-timed movement tasks: We used a self-timed movement task in our study. Like DRL tasks, reward was given only if the *first movement* occurred after a criterion time. Unlike DRL tasks however, our selftimed movement was referenced to a start-timing cue rather than the animal’s last response, and thus incorporated an explicit inter-trial period. To keep the animal performing briskly, we also enforced a maximal response window (7 s following the cue) to receive reward. Our lab pioneered self-timed movement tasks in monkeys (Lee and Assad, 2003; Lee et al., 2006), and we believe our mouse study is the first use of a self-timed movement task in rodents.

**Author response image 2. sa2fig2:** Animals across species frequently move before the criterion time during timing tasks, even at the expense of reward.

As in the peak procedure, subjects executing DRL or self-timed movement tasks tend to anticipate the criterion time, even though they receive less overall reward. For example, in Lee and Assad (2003), monkeys executing the self-timed movement task had first-movement-time distributions that peaked near the criterion time (2 s), but ~1/3 of trials were premature, and thus unrewarded (Author response image 2). Rats show a similar anticipatory pattern on DRL tasks (Author response image 2), although rats tend to move even earlier than monkeys, and on some trials, they are also unable to withhold short-latency reactions to the initial start-timing cue, producing an initial “spike” in the movement-time distributions (Kirshenbaum et al., 2008; Schuster and Zimmerman, 1961).

Mice executing DRL tasks behave very similarly to mice performing our self-timed movement task (Author response image 2, (Eckard and Kyonka, 2018)). Like rats, mice sometimes react quickly to the cue. They are also “less patient” than the rats: the peak of their timing distribution tends to anticipate the criterion time even more. Our mice showed similar behavior in our self-timed movement task (Author response image 2). Importantly, in the Eckard and Kyonka study, the *same* mice were also trained on a peak procedure task with the same criterion interval (18 s), in which the peak frequency of probe-trial responses occurred close to the criterion time (Author response image 2). This demonstrated that these same mice were capable of accurately estimating the criterion interval and suggested that the difference in peak position between the timing tasks was not the result of poor timing behavior. Rather, it has been emphasized since the 1950s that the “premature” response-peaks found in the DRL tasks belie a more accurate *latent* timing process, as follows:

Anger (1956) pointed out that if the animal responds early on a trial, it obviously eliminates the opportunity to respond *later* on that trial (Anger, 1956). The raw frequency of a particular response time is thus “distorted” by how often the animal has the *chance* to respond at that time. In other words, timing distributions from DRL or self-timed movement tasks – in which only *one* movement is allowed per trial – should take into account the number of response *opportunities* the animal had at each timepoint. This is accomplished by computing the hazard function, which is defined as the conditional probability of moving at a time, T, given that the movement has not yet occurred*.* (The hazard function was referred to as “IRT/Op” analysis in the old DRL literature*.*) In practice, the hazard function (HF) is computed by dividing the number of first-movements in each bin of the histogram by the total number of first-movements occurring at that bin-time or later *–* the total “opportunities.” The HF effectively captures the *instantaneous probability* of moving at a given time. In the peak procedure, that instantaneous probability can be read out directly from the rate of response itself, but in DRL/selftimed movement tasks, the instantaneous probability is a latent variable that must be derived by computing the HF.

To ground discussion of the HF, first consider a “null hypothesis” case: that the animal does not time its response, but rather has a *uniform* instantaneous probability (over time) of moving after the cue. This equates to a flat hazard function and manifests as an exponential movement-time distribution (modeled in New Manuscript Figure 1C). Indeed, for first training sessions, when the animals were not yet aware of the timing contingency of the task, we found a flat HF on average (following the typical “spike” of short-latency reactions to the cue, New Manuscript Figure 1D).

Next, we computed the hazard functions from our data *after* the animals had been trained for at least a week in the self-timed movement task. As Reviewer 3 pointed out, the raw first-movement-time distributions in our manuscript showed anticipatory peaks for both the 3.3 s and 5 s criterion times (New Manuscript Figure 1E top). However, HFs computed from those distributions reveal peaks close to the criterion times (New Manuscript Figure 1E bottom). This indicates that the instantaneous probability of moving in our task was maximal near the criterion time, demonstrating that the animals’ behavior was accurately tuned to the target-timing interval. We stress that this HF property is implicit in the *shape* of the first-movement-time histogram, but is not obvious if one considers that histogram’s peak alone. New Manuscript Figure 1E bottom-right shows average HFs for all 12 GCaMP6f photometry animals, revealing that the peak instantaneous probability of movement was close to the 3.3 s criterion.

In summary, these combined data indicate that after training, the animals “understood” the timing contingencies of the task, in that their instantaneous probability of moving peaked close to the criterion time. In the revised Results section, we have summarized these arguments and also present the hazard functions as evidence that the animals indeed were able to well-estimate the timing contingencies (New Manuscript Figure 1E bottom-right).

Notwithstanding the hazard-function analysis, it might seem surprising that our animals did not adopt a more patient strategy in an effort to receive more rewards. One possibility is that the animals were under-trained. This is unlikely, however, because the animals’ movement-time distributions evolved over the first 4-7 days of training but then were stable over *months*. Moreover, the animals were indeed *capable* of being more patient: the mice tested with the 5 s criterion time were first trained for weeks with the 3.3 s criterion, but within only *one* session of switching to the 5 s task, the mode of the movement-times distribution shifted later. If the mice had adopted that later mode for the 3.3 s task, they would have received rewards at a much higher rate (e.g., New Manuscript Figure 1E top). Clearly, during training, the animals “sought” to maximize the instantaneous movement probability near the criterion time (as revealed by the hazard function), rather than to optimize average reward rate. We can only speculate as to the cognitive pressures driving this strategy, but this behavioral pattern may reflect particularly strong temporal discounting of reward value, driving the animals to acquire rewards as quickly as possible when they become available.

Although previous rodent DRL studies invariably found frequent movements before the criterion time, there are two additional reasons why our mice might have been even “jumpier”:

First, we imposed a maximum movement-time window (unlike DRL studies), which may have added additional temporal urgency that drove earlier responses. We have anecdotal evidence from early training without the maximum movement-time window that is consistent with that possibility.

Second, our animals were water deprived, and their thirst-urgency may have driven them to move at earlier times, even at the expense of fewer rewards. Our mice presumably sated over the course of a session; for example, as sessions progressed, animals also licked less frequently during the inter-trial interval and made shorter lick bouts in response to reward. We also noticed that (unsurprisingly) animals tended to move progressively *later* over the course of a daily session, presumably as their thirst-urgency eased with accumulated rewards (Author response image 3). We were initially concerned that this non-stationarity could artifactually produce differences in dopaminergic signals on early- vs. late movement trials, based on the different proportion of trial outcomes over the course of a session. In our original manuscript, we addressed this concern by dividing sessions into quartiles and comparing the dopaminergic signals (as a function of first-movement-time) within and across each quartile. (We discussed this analysis in our “dF/F Validation Methods” section of the original and revised manuscripts.) We detected no differences in the relative pattern of ramping or baseline offset over the course of sessions (Author response image 3). For example, the relative difference in average ramping slope for trials with a first-lick-time at 2 s vs. 4 s was similar regardless of whether those trials occurred at the beginning or end of the session.

This inter-quartile comparison also helps to address Reviewer 3’s concerns about interpretability. First, the basic pattern of dopaminergic signals was similar whether the animals had fewer “correct” trials (beginning of the session) or more correct trials (toward the end of the session). Second, because many variables are presumably changing over the course of a session (e.g., thirst, motivation, vigor/fatigue, etc.), the consistency in the dopaminergic signal across the session suggests these signals are related instead to what is the same across the session – that the animal has to time its movement relative to the cue on a given trial.

In their summary, the editors stated, *“Issues with the behavioral performance and task design make it difficult to infer how signals correspond to internal processes such as reward anticipation, RPE, behavioral inhibition, or—as the manuscript suggests—timing.”* As we have shown here, the behavior we observed is consistent and interpretable in the context of behavioral timing. However, we reiterate this really wasn’t the point we were trying to make! Given the vigorous debate in the literature about “what dopamine signals represent,” it is likely that many different motivating factors (anticipation, RPE, value, *etc*.) are multiplexed into creating the dopaminergic signals that we observed during selftimed movements. (We have posted a BiorXiv manuscript showing how RPE signals could explain premovement ramping in our task, based on a theory developed by our colleague Sam Gershman; see (Hamilos and Assad, 2020).) But we are not asking about the (undoubtedly complex) *origin* of the dopaminergic signals, but rather how dopaminergic signals influence behavior in real-time.

Specifically, we desired to relate trial-by-trial differences in dopaminergic signaling to the timing of the animal’s movement, and to relate causal manipulations of dopaminergic activity to movement timing. As Reviewer 1 noted, we have “link[ed] DA activity to the timing of a reward-related decision extremely well.” That was exactly our goal.

**Author response image 3. sa2fig3:** The peak of the timing distribution becomes later over the course of a behavioral session, but the pattern of dopaminergic responses as a function of movement time does not change. (**A**) Median first-lick time divided into quartiles across sessions (only first-licks between 0.7-7 s considered to exclude rapid reactions and licks occurring after end-of-trial). Boxplot shown for individual photometry sessions with at least 100 first-licks (102 sessions total, red line: median, box: 25-75^th^ percentiles, whiskers: 1.5 IQR). Blue lines: individual animal averages across sessions; n=12 mice. (**B**) Histograms of median first-lick time for each of the 102 sessions shown in A. (**C**) Corresponding SNc GCaMP6f signals from DAN cell bodies averaged separately for each quartile, aligned both on cue onset (left) and first-lick-time (right). Traces plotted to 150 ms before first-lick to exclude movement-related response. Trials pooled into bins of 1 s each, e.g., blue: 1-2 s, green: 2-3 s… etc.

Editor point 2. It will be key to nail down the legitimacy of the claim that dopamine ramps and amplitudes at movement onsets on single trials. The main concerns (points 1-4) of R2 and point 5 of R3 well-articulate this concern.

This point was extremely useful, because it led us to consider in much greater depth questions of (i) single-trial dopaminergic dynamics, and (ii) how this signal may be influencing movement timing. We have added several major new analyses to the revised manuscript to address these points, culminating in a critical new analysis that revealed a more nuanced, *probabilistic* interpretation of the role of dopaminergic signals in movement timing.

(i) Are single-trial dopaminergic dynamics slow ramps, or are they better explained by discrete steps? In the original manuscript, our single-trial analyses consisted of a movement-time decoding model that incorporated GCaMP6f-signal threshold-crossing time as a predictor. This model was based on well-established diffusion-to-threshold models that have been extensively used by Shadlen and colleagues (and many others) to characterize ramping neural signals in the context of perceptual decision-making tasks (*e.g*., (Roitman and Shadlen, 2002)). Nevertheless, Reviewer 3 (R3) correctly pointed out that slow ramping signals *on average* could be due to slow ramping on *single trials*, or could be due to *discrete steps* on single trials that are distributed throughout the timed interval from trial-to-trial. Either underlying single-trial dynamic could result in slow ramping when averaged across trials.

R3 is probably aware that this question of “ramping vs. stepping” has been vigorously debated in the perceptual decision-making field. Alex Huk and Jonathan Pillow first suggested that average ramping dynamics (in single-unit recordings from monkey parietal cortex) could instead be explained by discrete step-functions occurring at different times from trial-to-trial (Latimer et al., 2015). Shadlen *et al.* (2016) immediately challenged this interpretation on analytical grounds – and the debate has continued to simmer (Latimer et al., 2016; Shadlen et al., 2016; Zoltowski et al., 2019; Zylberberg and Shadlen, 2016). However, Sahani and colleagues have argued that the step *vs*. ramp question may be fundamentally non-resolvable (Chandrasekaran et al., 2018). They showed that classification models are extremely sensitive to model parameterization, to the extent that different data-supported model parameterizations can produce *opposite* classification results when applied to the *same* simulated, ground-truth datasets. This confusion may be due to contamination of signals by ongoing neural activity unrelated to the decision process, as well as detection noise (Chandrasekaran et al., 2018).

Nonetheless, prompted by R3, we decided to take a stab at the “step *vs*. ramp” question in our data set. At the outset, we note that for single-trial steps to produce average ramping dynamics, the steptimes must be randomly distributed across the timed interval from trial-to-trial. A consistent, perimovement step (as R3 seemed to suggest) would not produce slow ramping on average, but rather a sharp jump at the time of movement – and that was not what we observed. However, we agree that a simple threshold-crossing analysis might not be able to distinguish a single-trial ramp *vs*. step process, because relatively late threshold-crossing times are only possible on trials with relatively late movements (e.g., a 3 s step-time cannot occur on a 2 s trial, by definition). To wit, our original singletrial threshold-crossing analysis made no assumptions about the underlying trial dynamics (step *vs*. ramp). However, we see R3’s point that the title “slowly-evolving dopaminergic activity determines the timing of self-timed movement” would not necessarily be uniquely supported by a simple thresholdcrossing analysis.

Caveats aside, with advice from Josh Tenenbaum at MIT, we implemented a single-trial hierarchical Bayesian model to analyze our single-trial GCaMP6f signals, using probabilistic programs in the novel probabilistic programming language, Gen.jl (Cusumano-Towner et al., 2019). These analyses are summarized in New Manuscript Figure 6—figure supplement 2. These programs infer underlying signal dynamics and return a probabilistic classification of single trials as either linear ramps or discrete steps. Both the step and ramp models were individually optimized to fit single-trial data, and both models were capable of capturing intuitive step and ramp fits for ambiguous signals that, by eye, could have “gone either way.” However, like previous efforts, our model returned inconclusive results, with about half of trials being better classified by a ramp and half by a step function. Given the findings of Chandrasekaran et al. (2018), this ambiguity was perhaps not surprising.

We thus took a step back and examined three additional, complementary approaches to tease the two models apart:

*Multiple threshold levels*. Although a simple threshold-crossing analysis does not definitively distinguish between underlying step *vs*. ramp dynamics, the two types of dynamics will have a distinct relationship with respect to *different* threshold levels, as our lab previously showed (Maimon and Assad, 2006). Consider the idealized simple ramp and step dynamics shown in New Manuscript Figure 6—figure supplement 3A. If we draw *three* arbitrary thresholds across each dataset, the thresholdcrossing time will always be the same for the discrete step model, regardless of threshold level (New Manuscript Figure 6—figure supplement 3B, blue lines). But the ramp model will have a thresholdcrossing time progressively *closer to the first-lick time as the threshold is increased* (New Manuscript Figure 6—figure supplement 3B, red lines). That is, for single-trial *ramping*, the *slope* of the thresholdcrossing vs. movement time relationship will increase as the threshold is increased (New Manuscript Figure 6—figure supplement 3B). In the original manuscript, we showed that the slope of the relationship between threshold-crossing time and first-lick time increased with increasing threshold level for a single behavioral session, consistent with the ramp model, but inconsistent with the step model. In the revised manuscript, we performed the same multiple-threshold analysis for all SNc GCaMP6f sessions in our 12 mice. Across animals and sessions, we found the slope of the relationship increased markedly from low to high as threshold level is increased, supporting the slow-ramp model on single trials (New Manuscript Figure 6—figure supplement 3C).

*Aligning trials on step*. If single trials involve a step change occurring at different times from trial-totrial, then *aligning trials on that step* should produce a clear step on average, rather than a ramp (Latimer et al., 2015). We thus used the Bayesian step model to find the optimal step position for each trial, and then aligned single-trial signals to that optimal step position. However, GCaMP6f signals aligned to the fitted step position and averaged did not yield a step function, but rather detected an apparent transient superimposed on a “background” ramping signal (New Manuscript Figure 6—figure supplement 4A). Step-aligned tdTomato and EMG averages showed a small inflection starting at the time of the step, but neither signal showed a background of ramping, unlike the GCaMP6f signal. This suggests that the detected “steps” were likely transient movement artifacts superimposed on the slower ramping dynamic, rather than *bona fide* steps.

*Variance analysis*. The ideal step model holds that steps occur at different times from trial-to-trial, producing a ramping signal when averaged together. In this view, the trial-by-trial *variance* of the signal across trials should be *maximal* at the time at which 50% of the steps have occurred among all trials, and *minimal* at the beginning and end of the interval (when no steps and all steps have occurred, respectively). To examine this, we derived the optimal step time for each trial using the Bayesian step model, and then calculated variance as a function of time within pools of trials with similar movement times (pooled at 1 s intervals). Rather than exhibiting an inverted “U” shape with respect to cumulative probability of when the steps occurred, the signal variance showed a monotonic downward trend during the timed interval, with minimal variance at the time of the movement, rather than at the point at which 50% of steps had occurred among trials (New Manuscript Figure 6—figure supplement 4B). This is inconsistent with a step model but consistent with a ramp-to-bound model, in which signals trend toward a similar level just before movement onset (Roitman and Shadlen, 2002).

Overall, we did not find evidence for a step dynamic on single trials; on the contrary, our combined observations concord with slow ramping dynamics on single trials. This indeed suggests that “slow dopaminergic dynamics” are informative of movement time. However, we again stress that our decoding models (New Manuscript Figure 6 and New Manuscript Figure 8) make no assumptions about the underlying single-trial dynamics.

(ii) Do dopaminergic signals encode the moment-to-moment probability of movement initiation? Reviewer 2 (R2) was concerned that our original movement time decoding model used only one point in the signal – a threshold crossing. We chose the threshold-crossing model based on common drift-diffusion-to-threshold models used to effectively describe the timing of perceptual decisions (see section (i) above).

However, R2’s comment really got us thinking. As described above, our evidence suggests that a slowly-changing dynamic characterizes single-trial dopaminergic signals—but what is the actual link to movement initiation? Our optogenetic stimulation and inhibition results suggested that dopaminergic signaling influences the moment-to-moment *probability* of movement onset: stimulation/inhibition shifted the probability distributions of movement times to earlier/later times, rather than immediately triggering or inhibiting movement (New Manuscript Figure 7B). Thus, in addition to trying to predict the *exact* movement time from single trial dynamics (which we already knew we could do to some extent from the threshold-crossing GLM), we asked whether we could predict the moment-to-moment *probability* of movement from the dopaminergic signals. In this view, we would take into account the *full time-course* of the dopaminergic signals, not just the single threshold crossing, which also addresses R2’s original suggestion.

As described in our response to Editor point #1, we showed that the hazard function – the probability of movement initiation *given* that movement has not yet occurred -- reveals the animals’ (accurate) latent timing process. The hazard function, by definition, defines the instantaneous probability of movement initiation; thus, if dopaminergic signals are predictive of the moment-to-moment probability of movement initiation, we should be able to reproduce the behavioral hazard function from the dopaminergic signals.

To test this hypothesis, we derived a nested probabilistic movement-state decoding model (New Manuscript Figure 8A). We applied a GLM based on logistic regression, in which each moment of time is classified as either a non-movement (0) or movement (1) state (New Manuscript Figure 8A-B). The model allows us to examine whether various measured parameters are good predictors for the probability of transitioning from the non-movement state to the movement state. In our initial model selection, we found that the instantaneous dopaminergic signal itself was a robustly significant predictor of movement state, whereas previous trial outcomes were insignificant contributors to the model (New Manuscript Figure 8—figure supplement 1).

The continuous dopaminergic signal was indeed predictive of current movement state at any time *t* – and up to 2 seconds in the past (New Manuscript Figure 8C). However, remarkably, the signals became progressively more predictive of the current movement state *at t* as time *approached t*. That is, the dopaminergic signal levels closer to time *t* tended to absorb the *behavioral* variance explained by more distant, previous signal levels (New Manuscript Figure 8C). (Note also that this observation is consistent with a diffusion-like *ramping* process on single trials – in which the most recent measurement gives the best estimate of whether there will be a transition to the movement state – but this finding is difficult to relate to a step process on single trials.)

We applied the fitted probabilities of being in the movement state to derive the fitted hazard function for each behavioral session. The dopaminergic signals were remarkably predictive of the hazard function, both for individual sessions and on average, explaining 65% of the variance on average (New Manuscript Figure 8D). Conversely, when the model was fit on the same data in which the timepoint identifiers were shuffled, this predictive power was essentially abolished, with only 5% variance explained on average (New Manuscript Figure 8E). These results of these analyses are included in (entirely new) New Manuscript Figure 8 in the revised manuscript.

Together, these results demonstrate that slowly-evolving dopaminergic signals are predictive of the moment-to-moment probability of movement initiation. When combined with the optogenetics results, they demonstrate that dopaminergic signals causally set this moment-to-moment probability of movement. To our knowledge, this is a completely novel view of how dopaminergic signals can influence movement initiation. Moreover, because this analysis takes into account the *entire* timecourse of dopaminergic signals, the probabilistic decoding model also addresses R2’s suggestion. (Thanks R2 – that was an incredibly useful suggestion!)

Editor point 3. We also encourage you to address the statement that ramps existed before licks outside the task, which shows the irrelevance of the task for the finding. Showing examples of this phenomenon might help. And we encourage you to consider how this finding relates to the central claim of 'timing.'

We set out to determine how dopaminergic signaling influences the timing of “self-initiated” movements, movements that, by definition, depend on some internal cognitive process to determine when they are initiated (rather than occurring in response to abrupt external sensory input; (Lee and Assad, 2003; Lee et al., 2006)). However, to detect whether that internal process unfolds relatively quickly or slowly, we required some *reference event* to align both behavior and neural signals. Our self-timed movement task does exactly that: it provides a visual/auditory cue-event that presumably “starts the clock”. Having an unambiguous time-reference point was the key element that allowed us to relate variations in dopaminergic signaling to variations in the timing of movement initiation.

Indeed, we found striking evidence that dopaminergic signals were informative of the timing process, both on average and from trial-to-trial (New Manuscript Figures 1,6, and 8). However, it is not clear that the specific processes we observed in the context of the self-timed movement task can be generalized to *all* “self-initiated” movements. Animals obviously also make “spontaneous” movements outside the context of contrived behavioral tasks in a lab. For example, our animals made exploratory licks *between trials*, when there was no explicit timing requirement governing these movements. We asked whether similar ramping might be present before these spontaneous licks precisely because we were curious whether the slow dopaminergic dynamics may be generalizable to *all* spontaneous movements –or rather are specific to our behavioral task, with its explicit timing requirement.

For the spontaneous licks, one approach could have been to just globally average the dopaminergic activity preceding *every* spontaneous lick during the inter-trial period, as done in previous studies that allowed animals to initiate movements *ad libitum* (da Silva et al., 2018; Howe and Dombeck, 2016). But at best, this global averaging could only provide evidence that activity ramps up before spontaneous licks. Obviously, global averaging throws away potential *variation* in dopaminergic dynamics that might be related to *variation* in movement time (and remember, such variation was the crux of our self-timed movement task!). So instead of globally averaging signals before spontaneous licks, we decided to pool spontaneous lick events with respect to the *only possible* reference we had – the time elapsed since the *previous* spontaneous lick (example shown in New Manuscript Figure 8—figure supplement 2). We didn’t expect this to be a great reference point, but it was the only reference available. Although the patterns were noisy and quite variable between animals, we did find variation in ramping in some animals that was indeed related to the elapsed time since the previous lick. To be clear, it is not surprising that these patterns were less stereotyped and well-formed—there is no strong reason *a priori* to believe that the previous lick would serve as a “good” reference event for the next spontaneous lick. What *was* surprising was that we found any differences in the ramping slope at all. Previous studies have reported short-latency increases in dopaminergic signaling within 500 ms of spontaneous movement initiation (Coddington and Dudman, 2018; da Silva et al., 2018; Dodson et al., 2016; Howe and Dombeck, 2016; Wang and Tsien, 2011), but never tried to relate that activity to the elapsed time since the last spontaneous movement.

Our findings for spontaneous licks *suggest* that slow dopaminergic dynamics may occur before *any* self-initiated movement, whether or not there is an explicit timing requirement. (An important caveat is that it is possible that our highly trained animals may have been “practicing” their timed movements between trials.) Thus, we are not concluding that the signals we observed are specific to explicit timing tasks – but nor was that the goal of our paper! But it is critical to emphasize that our findings for the spontaneous licks are far less conclusive than for our self-timed movement task. The presence of an unambiguous timing reference in the self-timed movement task (the visual/auditory start-timing cue) was the critical design feature that allowed us to detect the fine-grained relationship between the dynamics of dopaminergic signal and the dynamics of movement. We have made this point more clearly in the revised manuscript.

Editor point 4. We encourage you to clarify that the task was not designed to specifically rule in or out the many models of what dopamine encodes and rather clarify that the goal of the study is to identify dopamine relationship to upcoming movements or behavioral transitions. Fully addressing R3 comment 1 is necessary.

We believe we have fully addressed R3 Comment 1 in Editor points #1 and #3**.** We will simply add here that the key finding of our study is that dopaminergic signals predict movement timing remarkably well and are also *causal* to behavioral output—both of which are novel findings. Put differently, our study is concerned with the *downstream* (behavioral) impact of dopaminergic signals, and we remain agnostic to the *origin(s)* of the dopaminergic signal itself. Whether dopaminergic signals encode value, ongoing RPE, vigor or otherwise, our results demonstrate that these signals can causally influence behavior, potentially by modulating the instantaneous probability of initiating a movement. Moreover, the specific *temporal variation* of the dopaminergic signals was the key observation: regardless of their origin, the signals were excellent predictors of the (highly variable) timing of movement. To our knowledge, this is an entirely novel view, although it accords with the longstanding view (mainly from the clinical perspective – Parkinson’s disease, etc.) that dopamine somehow influences movement initiation.

[Editors’ note: what follows is the authors’ response to the second round of review.]

Reviewer #1:Hamilos et al. image DA axonal and neuronal CA signals in mice engaged in a self-timed lick task. They observed a ramping signal prior to lick onset that could predict, on average when a mouse would lick. While ramps in DA Ca signals have been observed as animals locomote towards known rewards (e.g. Howe et al., Hamid et al., Mohebi et al., all cited) and while DA activity is known to exhibit correlations with movement onsets – what's new and potentially important in this paper is how well the DA ramps could predict movement onset time – even on single trials. Complementing this finding were causal experiments where photoactivation or inhibition on single trials could promote or delay movement initiation. Given the amount of effort invested into the relationship between DA activity and upcoming movements – it is surprising that the strong correlation between the DA ramp and movement initiation (or decision, see discussion of Guru et al., paper below) hasn't been so clearly observed before (to my knowledge). This paper links DA activity to the timing of a reward-related decision extremely well.Figure 3 of Guru et al. (unpublished biorxiv paper from Warden lab, not cited) shows a similar result as this paper except the internally timed action is not to produce movement but rather to terminate movement (stop running on a wheel). The fact that the same ramp is observed in both of these conditions undermines the connection that the authors make about ramping DA and decision to move. Taken together, it seems the DA ramps are more about the timing of a self-paced decision (whether it be to start or to stop doing something) than about movement initiation. The authors may want to re-tweak the interpretations of this paper to allow for this more general perspective.

This is a great catch! Guru et al.’s work was pre-printed about a week after our own, and we took the opportunity to read it carefully. We were very comfortable re-tweaking our interpretation to allow for this more general perspective. We note an exciting connection to our initial motivation by movement disorders like Parkinson’s—patients not only have difficulty initiating movement, but also with *changing* movement (e.g., perseveration), implying a more general need for dopamine to flexibly transition between behavioral and possibly cognitive states. We’ve always liked this idea, but with our movement task alone, we were not in a position to make that argument. In the revised Discussion, we now suggest that our results may apply more generally to behavioral transitions, which would encompass both movement initiation and changing to a different movement, like stopping. We note that we are not entirely convinced that stopping isn’t in of itself a kind of movement initiation— counterbalancing musculature must be invoked to stop the momentum of the body and the running wheel that is different from the ongoing movement of running. But “behavioral transition” seems a good way to describe both the self-initiated lick and the stopping behavior.

Reviewer #2:In this study Hamilos and colleagues measured dopamine signals in mice performing a self-timing task. They took advantage of the variability in the first lick time to study how dopamine signaling preceding the first lick varied with the timing of the first lick. While a number of studies have reported short-latency (<500 ms) increases in dopamine activity immediately preceding movements, the authors make a number of novel findings. First, dopamine photometry signals ramped up over several seconds preceding the first lick. Second, the steepness of this ramping, even the amount of baseline signal predicted the first lick time. Third, optogenetic activation and inhibition reduced or increased the first lick time. The authors are commended for performing several different types of photometry experiments. The authors conclude that dopaminergic signals unfolding over seconds control the timing of movements, but not as much the ability to move itself. While the work is novel and adds an interesting perspective on the function of dopaminergic neurons, there were concerns about some of the evidence for the first two claims, as well as insufficient detail in some of the statistical analysis that make it difficult to fully judge the paper's merits. The main concerns are detailed below.1. Lines 151-153: "we observed systematic differences in the steepness of ramping that were highly predictive of movement timing (Figure 1D-E)." The reviewer agrees this is quite evident from the ramping curves, but still the authors should formally show the fact that the steepness (i.e., slope) of ramping is predictive of timing. The decoding model in Figure 6 goes some way toward this goal, but it seems Figure 6 uses time to threshold rather than the slope. It would be worthwhile to check in Figure 1 or Figure 2 whether the time to first lick is negatively correlated with slope of ramping.

We did not quantify the average slopes in New Manuscript Figure 2 for two reasons: First, Stimulated by R2’s feedback, we derived a new logistic regression analysis to characterize single-trial dynamics that takes into account the *entire time-course* of dopaminergic signaling on single trials (See Editor point #2), not just a single threshold crossing (New Manuscript Figure 8). This model suggests a more nuanced view of dopaminergic dynamics in which dopaminergic signaling sets a moment-to-moment probability of unleashing a prepotent movement. We also quantitatively analyzed the dynamics on *single* trials, but we could not definitively characterize those dynamics as linear ramps vs. discrete steps, as others have found (see Editor point #2). Because we did not find definitive evidence for sloping on single trials, we focused on movement-time decoding models that make no assumptions about the specific shape of single-trial dynamics (New Manuscript Figures 6 and 8). In this light, we felt that quantifying the slope of the average ramping responses would distract from the larger point.

We think these new analyses and interpretations will satisfy R2. However, if R2 feels strongly that we should quantify the average slopes in New Manuscript Figure 2, we can do so; it’s easy enough! But we would have to immediately add a caveat about the complexity of the underlying *single*-trial dynamics. In the revised manuscript, we have added a line at the end of the section describing the qualitative shapes of the average signals to let readers know we will be quantitatively addressing single-trial dynamics in a subsequent section of the paper.

2. Figure 4: similar to the previous comment. The reviewer found it difficult to interpret Figure 4 and suggests including a more traditional analysis, such as a Pearson correlation between the time to first lick and the level of pre-cue baseline dopamine.

We have eliminated this figure and focused on simpler analyses in the text, as per your suggestions. We also added the Pearson correlation coefficient for the relationship between baseline-signal amplitude and movement times in the Results text when referencing New Manuscript Figure 2 (r = -0.89).

3. Figure 6: the reviewer was confused about the contribution of the pre-cue baseline signal (predictor # 7) to the nested model. From the results in Figure 6B and 6C it appears that the pre-cue signal has almost zero contribution to the model (it was the only predictor with a non-significant weight in 6B). But this seems to contradict the authors' claim that "higher pre-cue, baseline DAN signals are correlated with earlier self-timed movements." If baseline signals do indeed correlate with timing of movements, shouldn't the weight of predictor # 7 be higher? This reviewer may have misunderstood some important detail, so some clarification of this issue in the text would be helpful. The reviewer also requests clarification of the difference between the "offset" (Predictor # 0) and "pre-cue" (Predictor # 7) as they both seem to be referring to some sort of baseline signal.

We see how that might be confusing—but there is indeed a simple explanation. First, this is a *nested* and cross-validated model, and we are only showing the final model with all predictors. We nested in the predictors one at a time during initial fitting, starting with those furthest from the lick. If you nest in the baseline predictors without the subsequent predictors (threshold crossing time), both baseline predictors are indeed significant. However, the GCaMP6f threshold crossing time was such a strong predictor that it absorbed much of the variance contributed by the baseline predictors; thus, after cross-validation, the threshold-crossing time survived as a significant predictor, but the lamp-off-to-cue (pre-cue) interval did not.

Second, there are actually *two* baseline predictors in the decoding model – the inter-trial interval signal (ITI; predictor #6), and the pre-cue signal (predictor #7) – and the way we named the second one caused the confusion. To clarify, we have renamed this as “lamp-off interval” in the revision. Predictor #6 spans the beginning of the ITI to the Lamp-Off event (10 s) and Predictor #7 spans the lamp-off to cue-on interval (drawn randomly from 400-1500ms). When we say “baseline signals” we are referring to *both* the ITI and pre-cue predictors in this model. We split up the baseline into two regions because of preliminary analyses suggesting that the baseline activity abruptly becomes better explained by the upcoming trial outcome after the lamp-off event. This suggested our best baseline prediction should be found during the lamp-off interval (which is indeed the case in the averages). However, this analysis is done on noisy single trials; because the ITI predictor (#6) covers 10 s vs. only 400-1500 ms for the lamp-off-to-cue interval (#7), the estimate of the median amplitude on single trials is much “cleaner” for the ITI. The end result is that, because the ITI predictor (#6) is nested first, it tends to absorbs almost all the contribution of the baseline in the decoding model, and is consequently a statistically significant predictor (in the expected effect direction – inversely proportional to the lick time). To clarify this point, we renamed predictors #6 and #7 to be more consistent with the text interpretation, as follows: 6=baseline:ITI, 7=baseline:lamp-off interval.

The reviewer also requests clarification of the difference between the "offset" (Predictor # 0) and "pre-cue" (Predictor # 7) as they both seem to be referring to some sort of baseline signal.

Predictor #0 (offset) actually has nothing to do with baseline. Here, “offset” is simply the boring constant term that is included in any linear regression (*e.g*, the “b” term in y = ax +b). We caused the confusion by using the term “offset” as a predictor of the modeled dopaminergic signal. To clarify, we renamed this constant term “b0”, the typical label for the constant term in multiple linear regression.

4. In several figure panels (including 6B, 7C, 7D) stats are either missing or details are too sparse to fully evaluate the significance of the results. The authors are requested to include information about sample size, type of test used, and if possible the exact p value.

We provided the sample size and test type in figure legends, but we have also added these details inline to the main text in the revision. We also added the sample size to the figure-panel image itself wherever possible to make this information easier to extract. We note that we have provided exact p-values and confidence interval bounds in the embedded source data that we provided for each figure.

For the specific panels mentioned:

New Manuscript Figure 6B shows regression coefficients, the 95% confidence intervals of which are determined by 2-sided t-test by the MATLAB glmfit package, and whose error is propagated across datasets. The figure legend indicates that the error bars indicate the 95% confidence intervals of the coefficients, which provides both the significance information of the p-value as well as the effect size. We added “2-sided t-test” to the figure legend for this panel.

New Manuscript Figure 7C shows the 1,000,000x bootstrapped mean difference in first-lick time between the stimulated and unstimulated conditions. The procedure used to calculate these values is detailed in the Methods section under *Quantification of Optogenetic Effects.* The statistical comparison of the data in this panel is explicitly shown as the plotted data of New Manuscript Figure 7D, in which we compare the category distributions. We added additional descriptors to this figure legend for these 2 panels.

Reviewer #3:Assad and colleagues examined nigrostriatal dopamine signals in head-fixed mice performing a licking task. Reward was available if mice first withheld licking for several seconds after a cue. They observed a ramping increase in dopamine before the time of first lick, that stretched in proportion with this hold interval. Optogenetic increases/decreases of dopamine caused licking to be earlier/later. They conclude that dopamine ramps are involved in controlling the timing of movements.There are several interesting aspects of this work, including the comparison between different DA signals (Figure 2) and the ramps themselves. Based on prior work in the field I have no problem believing most of the title "…dopaminergic activity controls the moment-to-moment decision of when to move". The problem is that the aspects of this work that are novel are either not well supported by data, or rely on questionable analyses and interpretations. There are also serious problems with the behavioral task and the quality of some recordings.Main points:1) The mice are notably bad at the behavioral task. Based on Figure 1B it is rare for them to wait long enough to get the reward. They do wait longer in the 5s condition compared to 3.3s, just not long enough. This general inability to perform the task casts further doubt on what internal representations might be driving dopamine signals and behavior.

We addressed this in detail in Editor point #1

I'm not sure it's reasonable to describe the behavior as "timing", if the timing is usually wrong. And given that they see the same ramps before spontaneous licks outside the task, the task seems largely irrelevant.

We addressed this in detail in Editor point #3

2) Optogenetic manipulation of dopamine made movements more or less likely, as expected from prior studies.

If we understand correctly, R3 is suggesting that optogenetic manipulation made movements more or less likely to occur *at all* on a trial. This is not what we found. Rather, we found that optogenetic manipulation *changed the timing of movement on single trials*. Specifically, optogenetic *stimulation* shifted the *distribution* of movement times to earlier times, while *inhibition* shifted the distribution to later times. However, a movement still occurred on every trial. Others have shown that *nonphysiological* stimulation can evoke nonspecific movements with short latency (*e.g*., activation can cause running bouts and inhibition can cause freezing; (Coddington and Dudman, 2018)), but modulation of a planned movement’s *timing* is more subtle – and is completely novel (to our knowledge).

But the authors claim more than this – that the optogenetic 'effects were expressed through a "higher-level" process related to the self-timing of movement'. I did not find this claim convincing, even beyond the issue that the mice are very poor at timing. If an action is prepotent in a given behavioral context as the result of training, then increasing dopamine may increase the likelihood of that action being emitted. Giving even more dopamine (higher laser power) may lower the threshold for *any* action being emitted. But this doesn't demonstrate anything much about timing per se.

Actually, we could not have said this better ourselves! The idea that “increasing dopamine may increase the likelihood of [a prepotent] movement” is *exactly* our current view. But this is nonetheless a completely novel view of the role of dopamine in movement initiation. Moreover, R3’s comment (and also a different comment from R2) spurred us to think more deeply about what the optogenetic manipulations were telling us mechanistically about the role of *endogenous* dopaminergic signaling in movement timing. In fact, we realized that the “probabilistic” view provides a beautiful mechanistic framework for how dopamine could affect *timing* (thanks R3!) Let us explain:

As described in our response to the editors (Editor point #2), opto-manipulations appeared to modulate the *probability* of a movement being emitted, such that the distribution of movement times was shifted earlier (opto-stimulation) or later (opto-inhibition). Moreover, as R3 notes, low-level (physiological) opto-stimulation only increases the probability of the “prepotent” movement – the desired lick. But this was for artificial stimulation of dopamine neurons; what does this mean with respect to the *endogenous* dopamine signals we had recorded during the self-timed movement task? Recall that we had found monotonically increasing dopaminergic signals during the timed interval (whether by stepping or ramping is immaterial for this point). If elevating dopamine increases the probability of movement, then this *endogenous* elevation in dopaminergic signaling should likewise lead to an increased probability of moving over time. It thus follows that the *specific dynamics* of the increasing dopamine signal should lead to a *specific time-course* of increasing probability of movement. A “specific time-course of increasing probability of movement” is *exactly* what we mean by “timing” – and now emerges a mechanistic hypothesis for how dopamine plays a role in that timing.

To test this hypothesis, we developed a new logistic regression framework to try to predict the instantaneous probability of movement (the hazard function) for individual behavioral sessions from the dopaminergic signals recorded during those sessions. In fact, the dopaminergic signals were remarkably accurate predictors of instantaneous movement probability (see New Manuscript Figure 8 and detailed response under Editor point #2). These results argue that the *specific dynamics* of dopaminergic signals regulate the timing of movement precisely by modulating the probability of movement initiation in a time-dependent fashion. That’s a fancy way of saying that dopamine plays a role in timing!

Honestly, we probably never would have taken the time to think this through in such detail if not for R3 and R2’s feedback—thank you! This really improved the science, not only the manuscript.

3) A key claim is that the dopamine signals are "slowly-evolving". This makes it essential to define "slow". In the dopamine field "slow" or "tonic" has often been used to referred to microdialysis signals presumed to change over tens of minutes. Here the authors describe ramping processes that may complete over several seconds, or be done in less than a second. So the kinetics of the signal seem more linked to the (varying) speed of behavioral transitions than to any inherently "slow" process.

That’s a good point. We were referring to signals that evolve over timescales of seconds, but for dopamine, the term “slow” might imply a minutes-long timescale. We modified the title for clarity.

4) Furthermore, in the Discussion the authors describe the instantaneous state of the dopamine signal at trial onset as "tonic", which seems like a mistake: they demonstrate that this signal has been immediately "reset" and cares more about the upcoming behavior than the immediately preceding trial just a few seconds earlier. This is not how "tonic" is used in the dopamine literature (e.g. Niv/Daw/Dayan describing tonic dopamine as integrating reward rate over prior recent trials).

Another good point. We changed this to reflect the timescale of seconds.

5) The ramp-like pattern is interesting and may indeed be comparable to dopamine ramps previously reported in other tasks. But ramps can arise as an artifact of averaging across trials in which events have different timing.

We addressed this in detail in Editor point #2.

To avoid this problem the authors examined single-trials (Figure 6). But the analysis employed does not actually measure ramping on single trials. Instead, they examine the timing of crossing an artificial threshold, and they see that this threshold crossing time predicts lick time well when the threshold is high. This seems equivalent to noting that dopamine increases shortly before licking, which we already knew from movement-aligned averages; it doesn't seem to demonstrate the point the authors wish to make.

We addressed this in detail in Editor point #2. Here we just add that in all our quantitative analyses, we truncated the dopaminergic signals at least 150 ms before movement onset to exactly avoid the problem that “dopamine increases shortly before licking”. We also reiterate that the threshold analysis was informative for *any* threshold level, not only a high level, as expected for a ramping model but inconsistent with a discrete step model (New Manuscript Figure 6—figure supplement 3, also see Maimon and Assad, 2006).

6) The authors note correctly that the signals they observe "could reflect the superposition of dopaminergic responses to multiple task events, including the cue, lick, ongoing spurious body movements, and hidden cognitive processes like timing." But I wasn't convinced that the regression models provided much insight into those hidden processes. Adding a "stretch" parameter felt merely like a descriptive fit to the observed data than a process-based model.

The point of the model was to test whether ongoing “nuisance” movements or optical artifacts could give rise to the ramping signals we observed. They could not, which indicated that the residual ramping and baseline offset signals were more likely related to hidden cognitive processes related to movement timing in the task. The stretch feature provides the insight that these hidden processes are unfolding at different rates on trials with different lick times. We go on to explore this idea in our process-based decoding models, particularly the new model that uses the dopaminergic signal to decode the instantaneous probability of movement initiation (see Editor point #2).

We note that the stretch feature is *model-free*. It makes no assumptions about the underlying “shape” of the dopaminergic signal; it just allows for temporal “expansion” or “contraction” to agnostically fit whatever shapes were present in the data. This quantitative, model-free description of the dynamics was necessary before developing and testing more constrained, process-based models. In particular, the stretch feature only encodes *percentages of timing intervals* and thus *cannot produce ramping* – or any other shape for that matter – unless that shape is present in the signal and stretches/contracts with the length of the interval. The approach is similar to encoding each timepoint in the dataset, a technique often used in model-free signal fitting with encoding GLMs (Park et al., 2014; Runyan et al., 2017). In such timepoint models, each timepoint relative to an event is represented with a feature (basis sets are sometimes used to reduce the number of parameters without changing the underlying structure of these models). This allows whatever shape is present in the data to be extracted *without any model assumptions* other than that timepoints relative to the alignment are similar across trials*.*

The only difference between our stretch feature and explicit time-encoding is that *the stretch feature allows trials of different lengths to be fit with the same parameterization*, and thus assumes only that *percentages* of an elapsed interval have similar dynamics across trials. If we hadn’t done this and instead explicitly encoded time (which we did try during model selection), some trials would not have been fittable by a subset of the predictors (e.g., a 6.5 s timepoint predictor is not encodable in a trial shorter than 6.5 s, by definition), which makes the model overly susceptible to fitting noise present on long trials. Employing the stretch feature allowed our model-free parameterization to properly fit *all* of the data present in our datasets with *every* predictor. The implicit stretching of this feature resulted in scaling of whatever shape the feature found to the length of the trial-interval. On the other hand, the stretch feature *cannot* fit if the underlying dynamics (shape) on long vs. short trials are *not* stretched versions of one another. Thus, the stretch parameter is extremely useful in assessing whether the dopaminergic dynamics evolve at different rates on trials with different movement times. Because the GLM empirically found a ramp (though was *not* constrained to do so), the stretch aspect of this feature translated to changing the steepness of this ramp, quantitatively revealing that an underlying ramping process is taking place at different rates (at least on average) during self-timed movement. We then explore this idea with process-based, single-trial decoding models. We have explained this rationale more clearly in the revised Results section.

7) It is not uncommon to see neural activity evolving between cues and movements in a manner that scales with the interval between these events (e.g. Renoult et al. 2006, Time is a rubberband: neuronal activity in monkey motor cortex in relation to time estimation). It is interesting that dopamine can do something similar, but that doesn't seem to support a special role for dopamine.

Indeed – in fact, our lab was among the first to show compelling examples of “stretching” in timing tasks in monkeys (Lee and Assad, 2003; Maimon and Assad, 2006). But this is precisely why we’ve been obsessed for years with examining these signals in dopamine neurons! As laid out in the Introduction of our manuscript, decades of pharmacological and lesions studies pointed to a central role for dopamine in self-timed movements. Our experiments not only show that the dynamics of dopamine signals are predictive of movement initiation**,** they also indicate a causal role. The ability to record and manipulate genetically-defined populations of dopamine neurons is a prime reason that our lab switched from using monkeys to using mice.

8) The figures are not well organized at all. Figs1 and 2 seem partly redundant, and are often referred to together. Figure 3 is mentioned only in passing to say that an accelerometer was present, without describing the Figure 3 results or why they are important (perhaps should be a supplemental figure after clarification).

Thank you for pointing this out, we totally agreed after thinking on this. We re-organized figures substantially to make the progression through the body of the text linear. We no longer jump around! You will find we moved Original Manuscript Figure 2 to the supplement, and we also no longer refer to any figures in passing.

9) Line234: "(highest S:N sessions plotted for clarity, 4 mice, 4-5 sessions/mouse, 17 total)". This seems weird and concerning. If signals were good enough to use for other analyses, why not this one?

We only intended to illustrate the effect with the sessions with highest S:N (because we had injected 2x virus in those animals). But we confused everyone with that figure, so we dropped it. We only report the results of the ANOVA now, which included all sessions from all animals.

10) Fig6, what does it mean that the tdt (control) signal is also a significant predictor of first-lick time? This seems like a serious problem for a control signal.

We can see why this might have been confusing—but tdt is working exactly as intended:

We know *a priori* there are movement artifacts expressed in both red and green channels: this is why we use tdt in the first place -- to “correct” for optical artifacts in the GcaMP6f signal. As we see in New Manuscript Figure 4B, we can detect optical artifact with tdt within ~150 ms before lick-detection, presumably when the mouse begins the action of extending the tongue, which requires some advance postural adjustments, jaw opening, etc. It’s not at all surprising to see these artifacts in tdt—in fact, it’s reassuring, because it tells us our movement control channel is detecting these artifacts.

That said, if the threshold-crossing model were trivially detecting upswings in GcaMP6f related to these optical artifacts, that would indeed be very concerning. To guard against optical artifacts causing trivial threshold-crossings, we first aggressively truncated GcaMP6f signals well before the first-lick (0.6 s), which, on average, appears to abolish the pre-movement tdt upswing. However, on a small subset of trials, the preparatory movements might start a little earlier than on average, and we were concerned that this could artifactually give rise to GcaMP6f threshold-crossing predictions of first-lick time. Thus, we regressed away optical artifacts by fitting the decoding model as a *nested* model. By adding predictors one at a time to these models, we regress out variance explained by the control signals in predicting the first-lick time. Specifically, we added the tdt predictor to the nested regression model *before* adding GcaMP6f to the model, such that any threshold crossing artifacts present in *both* green and red signals will have already been accounted for by the model iteration that included tdt but not GcaMP6f. Even so, the GcaMP6f threshold-crossing time was the most dominant predictor to the model and independently explained more variance in the first-lick time than any other predictor *despite having regressed away optical artifacts detected in the tdt channel.* We are thus assured that the predictive power of the GcaMP6f threshold crossing time was not the result of optical artifacts present in tdt. To summarize, it is not particularly surprising that tdt is a significant predictor to the model, given that on some trials the animals may have begun preparatory movements earlier than usual. The point is that we have not incorrectly ascribed such artifacts to genuine neural signal.

As a final note, the way we have treated tdt in the model is analogous to regressing away the tdt signal from the GcaMP6f signal, as has been the trend in the field, where papers often describe regressing out some “inert” red-flourophore signal (tdt, mCherry, etc.) from the GcaMP6f signal before showing any data. We have found this approach unsatisfying because the reader (a) does not know how much artifact/predictive power was present in tdt; and (b) must trust the authors that they have done the regressing away properly. By including tdt as a predictor to our decoding model, we *are* regressing away the optical artifact, but in a more specific way that takes into account how that artifact is expected to create confounds in *this particular* model (here, how the pre-movement preparatory motions of the animals presumably give rise to trivial threshold crossings). Furthermore, we report the tdt signal throughout the paper (including in this model), thereby allowing readers to draw their own conclusions about the reliability of the GcaMP6f signal.

[Editors’ note: what follows is the authors’ response to the third round of review.]

Essential revisions:1) Please address this concern regarding 'baseline" predictors: The authors have made significant improvements in describing the task, but there remained a lack of information about behavior in the lamp-off period before the cue was turned on (the baseline). The main concern is that this delay period is technically not a baseline because the lamp-off serves as an additional predictive cue. The authors tried to prevent this by randomizing the delay time (0.4-1.5 s), but this range looks comparable or even smaller than the variability in first-lick time shown in Figure 2B. They also did a control experiment without a lamp and showed qualitatively similar performance, but I didn't find the data very convincing (for some reason the data shown in figure supplement 1C does not have clear peaks at 3.3 s). Knowing whether this is a real baseline is important because the authors set cue-on as t=0, and do not consider behavior prior to that. This would not be a big concern if the authors can show that there is no statistically significant relationship between the lamp-off to cue-on delay and the lick onset on that trial. Otherwise, the concern is that their finding that higher baseline fluorescence predicts earlier licks may have a trivial explanation.

This is a good point. We have addressed this issue in four ways:

i) Provide a clear distinction between “baseline” and “lamp-off” intervals. Because the lamp-off could act as a partial predictive cue, we agree that for us to call something a “baseline effect”, it must be present *before* the lamp-off cue. Thus, to avoid confusion, we’ve clearly defined the pre-cue intervals in the revised manuscript as follows:

“Baseline”: now only refers to timepoints *before* the lamp-off cue.

“Lamp-off Interval (LOI)”: now refers to the interval between the lamp-off and cue onset

ii) Relationship between the lamp-off interval (LOI) and first-lick time. Per your suggestion, we implemented a regression model to characterize the relationship between LOI duration and lick-time:

First-lick(s) = b*lamp-off(s) + b0

Only 14/98 sessions recorded at SNc showed any significant relationship between LOI duration and first-lick time, and the relationship was very weak (R^2^ < 0.04 for 13/14 significant sessions). Moreover, the majority of those 14 sessions were performed by 2 animals (out of 12), and for one of those two animals, the direction of the relationship between LOI duration and first-lick time was not even consistent.

To make the same point more intuitively, we have additionally added new Figure 2—figure supplement 1 showing averaged GcaMP6f responses aligned to the Lamp-Off Event rather than the cue onset (as in Figure 2C). When average responses were aligned to the Lamp-off, we found lower-amplitude transients more spread out in time**,** compared to aligning responses to the cue onset. This also argues that the cue onset was the more reliable “start timing” cue.

iii) Examination of the pre-Lamp-Off Baseline interval. Notwithstanding point #2, if the weak correlation between LOI duration and first-lick time explained the baseline effect we observed, then the correlation of dopaminergic signals with first-lick time should only be present *after* the lamp-off event. That is, if we restrict the analysis to the *preceding* (aka *pre*-Lamp-Off) Baseline period (in any and all sessions), we should no longer see the relationship.

In the revised manuscript, we now examine the Pearson Correlation between the Baseline (pre-Lamp-Off) dopaminergic signal and the first-lick time. We averaged the dopaminergic signals from 2 s before the Lamp-off Event *until* the Lamp-Off Event. The “true” Baseline dopaminergic signal remained inversely correlated with the first-lick time (r = -0.63). This is also clearly evident by eye in the Lamp-Off aligned average responses in new Figure 2— figure supplement 1A. The correlation was weaker than observed after the Lamp-Off Event (r = -0.89)—but this is not unexpected: the lamp-off event comes later than the Baseline period, and in our movement timing/probability decoding models, we found that dopaminergic signals become progressively more informative of the first-lick time as the movement approaches. We are thus quantitatively assured that baseline correlations are not trivially explained by dopaminergic responses to the Lamp-Off Event.

iv) Omitting sessions with significant LOI/first-lick time relationship. We also repeated the analysis in Point (iii) while omitting the 14/98 sessions for which we found a significant (albeit weak) relationship between the duration of the LOI and the first-lick time (Point #2). Omitting those sessions did not eliminate the inverse correlation between the average dopaminergic signals and the first-lick time (r = -0.83) despite being nosier from reduced averaging, and the averaged responses were qualitatively unchanged compared to the averages with all 98 sessions included (Figure 2—figure supplement 1).

In summary, the baseline dopaminergic signal was correlated with the first-lick time even before the Lamp-Off Event, independent of whether the animal’s timing was influenced by the duration of the lamp-off interval.

2) Address comments re: Figure 7, where there is still some confusion about the description of the sample size. For example, line 339 lists 12 mice but I think the correct variable is number of trials. The text should thus include the size of the actual variable used in the plots (listing the number of mice as well is ok, but this is not sufficient on its own). Furthermore, the results would more convincing if this figure included single-animal comparisons rather than just pooled data or across-session comparisons. Basically, could the authors please include a panel similar in style to Figure 7C but where individual points represent single animals, not sessions? This would be more consistent with other optogenetic-behavioral studies in rodents.

Sure, thanks for pointing that out. We have updated the main figure panel with this animal-by-animal analysis and moved the by-session plot to the supplement.

**Author response image 4. sa2fig4:** Original by-session figure compared to by-animal figure (figure 7—figure supplement 3C).

3) Please better address an issue with interpretation of results, specifically regarding movement timing versus reward expectation: The authors claim that their results mean a causal role for the level of dopamine ramps in modulating the probability of action initiation, but that interpretation seems strange given that these ramps are often observed when animals are already moving (Howe et al., 2013; Hamid et al., 2016; Engelhard et al., 2019; Kim et al., 2020; Guru et al., 2020). In common with all these papers, dopamine ramps seem to be related to the temporal prediction of when reward will be available. This was shown nicely by Kim et al., 2020, where they showed that regardless of the movements performed by the mice, ramps are elicited in relation to when the animal expects the rewards to arrive (see especially their moving bar experiments). In the present work, this interpretation is also consistent with the results (and more consistent with previous works): when animals decide to move early, the DA system receives this information and that results in a more steeply rising ramp, while when animals decide to move late, the same thing occurs, and the ramp rises more slowly. The issue at hand is that the initiation of movement (licking) is strongly correlated with reward delivery (or with the time of expected reward delivery in case of failed trials) and thus it is not possible to distinguish between these interpretations. So, in summary, the authors should address the concern that the interpretation of dopamine ramps modulating the moment-to-moment probability of action initiation is unwarranted. Results should be reframed to make it clear that this claim cannot be supported given the possible alternative explanations (such as the one suggesting that DA ramps reflect a prediction of the time of upcoming reward delivery, which is more consistent with the previous literature).Alternatively, if the authors can find an experimental way to dissociate the expected time for reward delivery from the moment of action initiation, then that would be one way to make a decisive conclusion. For example, the water dispensation could occur at a fixed time relative to the cue – even as the mouse can initiate licking whenever it wants. If DA dynamics still predict initiation time, then the authors' perspective is supported. Yet if the DA dynamics no longer predict initiation time, and instead simply ramp to the reward time, then reviewer 4's perspective is supported. Either result would be interesting. Performing these experiments would be ideal, but not necessary for publication. If the authors do not wish to conduct new experiments along these lines, they should – throughout the manuscript – modify their text to include this alternative interpretation (reward expectation) – or offer in rebuttal a convincing argument as to why the logic of reviewer 4's (and the BRE's) thinking is not sound.

We’ve thought a lot about this issue since early in the project and discussed it extensively with colleagues. In a nutshell, we fundamentally agree with the editor/reviewer’s point: because movement occurs just before reward, we cannot strictly rule out that the dopaminergic ramping in the self-timed movement task represents reward anticipation. We tried to make this clear in the previous iterations when discussing the *origin* of the ramping signal (as opposed to the behavioral *readout* of the ramping signal, which is the focus of this manuscript), but we have now modified the manuscript to more explicitly reflect this alternative interpretation. Nonetheless, logic still points toward a relationship between the ramping signals and movement initiation in our task. There is a lot to unpack in this regard, and we appreciate the opportunity to share our thoughts here, as follows:

First, it is important to note that dopaminergic ramps have *not*, in fact, been observed in relation to *passive* temporal expectation of reward*—*except when some kind of external sensory indicator of proximity to reward was provided. For example, in Kim et al.’s moving bar task, the mouse was provided a *visual update* of the timing of reward delivery, and dopaminergic ramping was observed. Likewise, in Howe et al.’s navigation tasks (and Kim et al.’s VR navigation task), animals encountered spatial cues indicating the proximity to reward. In contrast, in Pavlovian conditioning, where animals are not provided external cues as to the (fixed) timing of a reward following a sensory cue, dopaminergic ramps are *not* observed—for example in the classic work of Wolfram Schultz (Schultz et al., 1997) and in numerous Pavlovian studies in rodents (e.g., Menegas et al., 2015; Schultz et al., 1997; Starkweather et al., 2017; Tian et al., 2016, etc.). Because Pavlovian conditioning does not require a movement to obtain reward, it is arguably the cleanest means of dissociating timed reward anticipation from movement initiation. The *lack of* dopaminergic ramping in Pavlovian conditioning thus argues that temporal tracking of proximity to reward *alone* is insufficient to generate dopaminergic ramps.

Given the observations from Pavlovian conditioning, we were initially surprised that ramping was present in our self-timed movement task, which also lacked explicit cues indicating proximity to reward. That is, if the dopaminergic signaling were *solely* concerned with temporal proximity to reward rather than movement, we would not expect ramping in *either* Pavlovian conditioning or our self-timed movement task—yet we did find ramping.

To reconcile the lack of dopaminergic ramping in Pavlovian conditioning with the presence of ramping in other paradigms (e.g., Howe et al., 2013), our colleagues Sam Gershman and Nao Uchida have proposed a model in which ramping arises from “resolving uncertainty” of one’s position in the value landscape, for example, as informed by the presence of external visuospatial cues. (We are not aware of any other unifying model that has addressed why ramping occurs in certain paradigms but not during Pavlovian conditioning.) In the Gershman/Uchida model, temporal uncertainty in Pavlovian tasks leads to a “smearing” among internal timing states, distorting the value function in a way that eliminates ramping under an assumption of an “ongoing” RPE calculation by dopamine neurons (Kim et al., 2019; Mikhael and Gershman, 2019; Mikhael et al., 2019). But how do we explain ramping in our self-timed movement task, where the only indicator of proximity to reward was the animals’ own internal determination of elapsed time?

When we discussed our findings with the Gershman and Uchida groups, we were all inevitably drawn to the idea that there must be something crucial about the *operant* nature of our task that leads to ramping—that is, the dependence on the animal actively moving to pursue reward. In light of Gershman et al.’s formal model, we infer that the animal must resolve its own uncertainty about its current state in time in order to determine when to move. The animal’s subjective timing may not accurately reflect objective time from trial-to-trial, but its traversal of internal “timing states” still represents a reduction in uncertainty of its current timing state. In this view, the difference between our operant timing task and Pavlovian timing tasks is that the animal itself exactly determines its movement time in the self-timed movement task (i.e., it “knows” its own timeline/state, inaccurate as it may), but the animal does not accurately predict its movement/reward timing in Pavlovian tasks, where reward is delivered passively. We developed this model in detail in a companion theory paper (Hamilos and Assad, 2020), which we wrote mainly because we did not have room in the *eLife* Discussion section to fully articulate the idea. However, we think this idea is important, and have summarized it in the revised Discussion.

Given the foregoing argument, the most parsimonious explanation for ramping in the self-timed movement task is that ramping depends on the process of movement generation, which is the singular element that is missing from the Pavlovian paradigm. (Note this does not mean that *all* dopaminergic ramping requires movement—e.g., Kim et al., 2019—but rather that ramping in the absence of external sensory cues can occur if the animal controls when it moves.) However, one could still argue that even if self-timed movements lead to ramping, ramping nonetheless reflects anticipation of reward without itself influencing movement. That is, one might suppose that the dopaminergic signal could be a “passive” monitor of one’s trajectory through the value landscape, as indicated either by the passage of time or by explicit sensory/spatial cues. Testing this idea was our main motivation for doing the optogenetic stimulation/inhibition experiments. Essentially, our question was not whether ramping better predicts reward or movement onset, but rather whether the dopaminergic signal can also be leveraged to causally influence movement initiation. If dopamine were simply a passive observer of reward, we might expect changes in behavior on subsequent trials, but not necessarily on the same trial*.* Rather, we found that DAN manipulation causally altered movement time on the same trial as the stimulation, arguing strongly that the value-related signals carried by the dopaminergic system could be harnessed moment-by-moment to influence when the movement occurs.

Note our emphasis on “could be” in the previous sentence. We stress that we have no reason or evidence to think that the presence of dopamine ramping always leads to movement. As we pointed out in the first revision of the manuscript (and in our previous reply to Reviewer #3), we expect dopamine-influenced movement under the right circumstances, that is, when a pre-potent movement is “loaded and ready to go”—exactly as in our self-timed movement task.

Importantly, we also note that our arguments here and in the revised manuscript do not depend on the Gershman/Uchida model and are fully compatible with value and other models of the *origin* of the dopaminergic signal. We simply note Gershman/Uchida’s model because it is the only model we are aware of that has mathematically and experimentally attempted to reconcile the lack of ramping observed in Pavlovian tasks with the ramping observed in these notable “special cases.”

Finally, one might argue that modulation of dopaminergic signaling during the optogenetic manipulations changed the animal’s belief of when reward might occur, thereby leading to the change in the timing of movement initiation. We do not disagree with this interpretation—in fact, we suspect this is precisely what happens, as we lay out in Hamilos and Assad, 2020. That is, reward expectation may be the very force that propels movement in our task. In this view, reward expectation is intrinsically intertwined with movement initiation. Our point is that whatever the dopaminergic signal is tracking, it can be harnessed to influence the probability of movement onset.

We have updated the revised manuscript to better reflect these arguments, both in the logical lead-in to the optogenetics experiment in Results and in the revised Discussion.

A final point: The editors/reviewers suggested that one way to disentangle movement and reward anticipation would be to delay reward. We also considered that idea early on, but abandoned it. The reason is that even if we found dopaminergic ramping up until movement with a delayed reward, that signal could *still* be reward-related (even if ramping stops right at the movement). In the Gershman/Uchida model, temporal uncertainty would be resolved whether or not the animal is immediately rewarded: the value function still increases as reward is approached (due to reduction in temporal discounting), and the reduction in temporal uncertainty (and thus ramping) is afforded by the operant nature of the task, as described above. Thus, *in principle* we don’t expect that experiment could rule out a reward-related explanation for ramping.

Just to show we’re not fast-talking here (or too lazy/chicken to do that experiment for real, LOL), we actually have done that temporal dissociation in a different experiment in the lab. We have another movement task in which the mouse makes a self-timed forelimb reach to a touch sensitive bar. We also found (amazing) seconds-scale ramping (either up or down) in spike rates from pallidal projection neurons in the SNR during the timing interval before the limb movement. However, we wanted to disentangle whether those ramping signals were in relation to the forelimb movement or to the licking that followed almost immediately after the reach. For that reason, we used a servo-motor to withdraw the juice tube before each trial, and then on successful trials, we swung the tube back toward the animal’s mouth 833 ms after the lever touch. For most SNR neurons, we found that the ramping still occurred up until the time of the reach (and did not continue during the delay). However, in some of those experiments we also likely encountered dopamine neurons (DANs), either from stray microwires in the SNC or putative DANs known to be sprinkled in the SNR (based on broad spike width and characteristic 2-10 Hz background firing rate). For those putative DANs, we found exactly what you were suggesting: the neurons showed ramping, ramping still peaked at the time of the forelimb movement, and ramping did not continue during the delay between the lever press and the reward delivery (see Author response image 5 for example). It would be tempting to claim that this supports the movement-initiation argument for ramping in (putative) DANs. But as we explained above, this observation cannot in principle rule out a reward-based signal. We maintain that a reward-based signal can still influence movement.

**Author response image 5. sa2fig5:** Ramping of putative DAN spiking rates peaks at the time of self-timed movement initiation, not the time of reward delivery.

References

Hamilos, A.E., and Assad, J.A. (2020). Application of a unifying reward-prediction error (RPE)-based framework to explain underlying dynamic dopaminergic activity in timing tasks. bioRxiv, 2020.2006.2003.128272.

Howe, M.W., Tierney, P.L., Sandberg, S.G., Phillips, P.E., and Graybiel, A.M. (2013). Prolonged dopamine signalling in striatum signals proximity and value of distant rewards. Nature *500*, 575-579.

Kim, H.R., Malik, A.N., Mikhael, J.G., Bech, P., Tsutsui-Kimura, I., Sun, F., Zhang, Y., Li, Y., Watabe-Uchida, M., Gershman, S.J.*, et al.* (2019). A unified framework for dopamine signals across timescales. bioRxiv, 803437.

Menegas, W., Bergan, J.F., Ogawa, S.K., Isogai, Y., Umadevi Venkataraju, K., Osten, P., Uchida, N., and Watabe-Uchida, M. (2015). Dopamine neurons projecting to the posterior striatum form an anatomically distinct subclass. *ELife 4*, e10032.

Mikhael, J.G., and Gershman, S.J. (2019). Adapting the flow of time with dopamine. J Neurophysiol *121*, 1748-1760.

Mikhael, J.G., Kim, H.R., Uchida, N., and Gershman, S.J. (2019). Ramping and State Uncertainty in the Dopamine Signal. bioRxiv, 805366.

Schultz, W., Dayan, P., and Montague, P.R. (1997). A neural substrate of prediction and reward. Science *275*, 1593-1599.

Starkweather, C.K., Babayan, B.M., Uchida, N., and Gershman, S.J. (2017). Dopamine reward prediction errors reflect hidden-state inference across time. Nat Neurosci *20*, 581-589.

Tian, J., Huang, R., Cohen, J.Y., Osakada, F., Kobak, D., Machens, C.K., Callaway, E.M., Uchida, N., and Watabe-Uchida, M. (2016). Distributed and Mixed Information in Monosynaptic Inputs to Dopamine Neurons. Neuron *91*, 1374-1389.

[Editors’ note: what follows is the authors’ response to the fourth round of review.]

Essential revisions:1) 2/3 reviewers were not persuaded by the author's rebuttal arguments that DA ramps reflect movement initiation and not reward expectation, resulting in an enduring concern that the main message of the paper, evident even in the title, is not watertight. All reviewers still think the paper's findings are timely and important – so a tempering / adjustment of the claims is a reasonable path forward to publication. The change in 'pitch' of this paper would need to include changes to title, abstract, and discussion. While some specific recommendations are included in the full reviews pasted below, the authors are in the best position to truly internalize reviewer's four's sound arguments to transform the message of the paper into one that allows for reward expectation to be the variable that DA ramps reflect, that optogenetic experiments manipulate, and that ultimately affects movement timing.If you have not already done so, please ensure that your manuscript complies with eLife's policies for statistical reporting: https://reviewer.elifesciences.org/author-guide/full "Report exact p-values wherever possible alongside the summary statistics and 95% confidence intervals. These should be reported for all key questions and not only when the p-value is less than 0.05."

Thank you again for your continued careful consideration of our manuscript. You guys have certainly inspired us to think more deeply about our findings. We think that the overarching problem has been that we inadvertently blurred the questions of (1) the *origin* of dopaminergic ramping with (2) the connection of dopaminergic ramping to movement. Your last set of comments really crystallized this issue for us. In the revised manuscript, we’ve separated the questions, and we think the whole thing works better. In this reply, we’ll deal with both of these questions in turn, and then summarize how we have integrated them in our revised manuscript. (Please forgive the rather lengthy reply, but we want to lay out our logic for you as clearly as possible.)

*Origin of dopaminergic ramping*. We agree fundamentally with your main point, that the ramping dopaminergic signals that we observe before self-timed movements are likely related to “reward expectation.” Because ramping is observed in other *non-movement* scenarios that also end up with the animal receiving reward, reward expectation has always struck us as the most parsimonious explanation for the origin of dopaminergic ramping. In both our Discussion section and a separate BiorXiv manuscript (Hamilos and Assad, 2020), we presented a model for how reward expectation could lead to ramping in dopaminergic neurons (DANs) during our self-timed movement task. The model was adapted from that of Sam Gershman, who treated DAN ramping as an “ongoing” reward-predictionerror signal (Gershman, 2014), but our results are also compatible with a more expansive view of reward expectation (e.g., increasing value over time, etc.). We also agree that our previous title and abstract did not make the reward connection clear. We’ve revised the title and abstract accordingly (see below).

There are, however, some points to clarify on the subject of “reward expectation,” informed by the extensive literature on the topic. The points may seem niggling, but we believe they impose important constraints on the interpretion of our findings, and we have treated them carefully in the revised Discussion.

In our previous reply, we argued that “passive” reward expectation, defined as automatically receiving a reward a fixed time after a cue (without any external indication of progress to reward, e.g., Kim et al., 2019), was not sufficient on its own to cause ramping activity in DANs. For example, in the classic work of Wolfram Schultz (subsequently reproduced by many others), simple associative learning does not lead to ramping increases in DAN activity. The reviewers pointed out one exception, the paper of Fiorillo et al. (2003), in which apparent ramping was observed in monkey DANs following conditioned stimuli that indicated reward probability was <1 (Figure 3B of that paper). We know that paper very well (our lab worked with monkeys for >20 years before switching to mice), but we skirted it for a reason: the “gradual increase” in DAN activity reported by Fiorillo et al. was not reproduced in subsequent papers, either from the authors’ own lab(s) or from other labs. In particular, Tobler, Fiorillo and Schultz (2005) reprised the Pavlovian associative learning paradigm, but did not observe ramping in any condition for which reward probability was 0.5 (see authors’ Figures 1B and 4B). Chris Fiorillo also did a variant of the same experiment when he was a postdoc in Bill Newsome’s lab (Fiorillo, 2011). Though Fiorillo focused on the DAN response to the cue, there was no hint of ramping-up in the p=0.5 data from either of the two animals in the study; rather, the DAN activity appeared to *decrease* monotonically to a low plateau following cue presentation (see red p=0.5 traces in Figures 5A and 7 of that paper).

We are aware of three additional studies that used Pavlovian paradigms with reward uncertainty. Matsumoto and Hikosaka (2009) recorded from DANs in monkeys using a Pavlovian task, and they found no ramping following visual conditioned stimuli predicting reward with p=0.5 (authors’ Figures 2B and 3B). Tian and Uchida (2015) likewise did not observe ramping in opto-tagged mouse DANs following conditioned odor cues predicting reward with p=0.5 (authors’ Figure 2G). Finally, Hart et al. used fast-scan cyclic voltammetry to record DA release in the nucleus accumbens of rats undergoing Pavlovian conditioning with variable reward probability. With training across multiple sessions, a sustained DA signal exceeding the baseline level emerged for the p=0.5 cohort of animals—but that DA signal still appeared to *decrease* monotonically following the conditioned stimulus (authors’ Figure 3E).

It’s not clear why Fiorillo et al.’s 2003 findings bucked the “no ramping” trend, but one possibility is that Fiorillo et al. did not require the monkeys to fixate, which may have allowed the animals to make an accelerating pattern of saccades during the delay in the p=0.5 case, potentially driving increasing DAN activity. Regardless, the preponderance of evidence suggests that reward expectation *alone* is insufficient to cause ramping activity in DANs. We have stated that explicitly in our revised Discussion. Our intention is not to throw shade on Fiorillo et al., but rather to better interpret our own findings.

In our revised Discussion, we have made clear the reward-expectation interpretation for the origin of dopaminergic ramping, but the available evidence nonetheless suggests that “pure” reward expectation is insufficient to drive ramping activity in DANs. We also added a paragraph about the Gershman model, to explain how our findings could fit into the broader “reward-expectation” context. To our knowledge, the Gershman “ongoing RPE” model is unique in addressing the discrepancy in ramping between operant and Pavlovian tasks, but we also emphasize in the Discussion that our results are compatible with broader views of reward expectation (increasing value over time, etc.).

In addition, we have changed the title and abstract of the paper to better reflect the reward-related component of our experiment. However, we are hesitant to state “slowly evolving dopaminergic activity reflects reward expectation” in the title, because we did not test that directly in our experiment, nor do we have direct evidence for that view; rather, “reward expectation” provides a *parsimonious* explanation for our findings given the parallel experimental and computational literature on dopaminergic signaling. Our new title is “Slowly evolving dopaminergic activity modulates the momentto-moment probability of reward-related self-timed movements.” However, we could also go with a more “neutral” (although less informative) title: “The relationship between slowly evolving dopaminergic activity and reward-related self-timed movements.” The final line of our updated abstract also makes clear the reward-expectation connection: “We propose that ramping dopaminergic signals, likely encoding dynamic reward expectation, can modulate the decision of when to move.”

*Connection of dopaminergic ramping to movement.* In our previous manuscript, we pointed out that dopaminergic ramping might serve as a “passive” monitor of reward expectation, or ramping might have real-time effects on behavior, such as influencing reward-related movements. To examine this question, we optogenetically activated or inhibited DANs from the time of the start-timing cue until the initiation of the self-timed movement. We found that manipulating DAN activity indeed affected movement time *on the same trial*, as if increasing/decreasing the probability of movement. However, Reviewer #4 pointed out that DAN activation/inhibition could have a *secondary* effect on movement, by modulating motivation or by heightening/dampening reward expectation. R4’s interpretation implies that the evoked movements would be reward-related, such as licking or reward approach. This is a reasonable proposal, and we consider it explicitly in our revised Discussion. However, the literature again suggests a more nuanced view, as follows:

Optogenetic stimulation (or inhibition) of DANs has been shown to modulate movements in unconstrained mice (Barter et al., 2015; Howe and Dombeck, 2016; da Silva et al., 2018); however, those studies did not have rewards available during the stimulated trials, and thus could not address the question of whether increased motivation leads immediately to reward-related movement. More relevant are studies that manipulated DANs in the context of a reward-related behavior. There are numerous studies that have reported increased reward-related movements following DAN activation (or other manipulations that raised DA levels)—*but not on the same trial*, unlike what we found. Rather, most of these studies reported reward-related movements that *emerged in subsequent trials or sessions*, interpreted as a learning effect over time/training (e.g., Steinberg, et al., 2014; Ilango et al., 2014). We thus restrict the following discussion to studies that examined whether modulating DAN activity could affect reward-related movements in “real time”—*on the same trial*.

We are aware of two studies in which activating DANs may have caused reward-related movements on the same trial. First, Phillips et al. (2003) studied reward-related approach behavior in rats trained to press a lever to obtain intravenous cocaine. *Electrical* stimulation of the VTA caused increased leverapproach behavior in the period 5-15 s after stimulation. However, electrical stimulation could have excited afferent fibers in the VTA (anti- or ortho-dromically), potentially engaging non-DAN-dependent pathways. Second, Hamid et al. (2016) observed that optogenetic stimulation of DANs could shorten the latency to engage in a port-choice task in freely behaving rats—but only for trials in which the rat was not already engaged in the task. If the rat was “already engaged in task performance,” the latencies actually became slightly *longer*.

In contrast to these two equivocal findings, numerous reports suggest that optogenetic manipulation of DANs does *not* modulate reward-related movements on the same trial. First, we ourselves could not evoke licking (nor inhibit spontaneous licking) outside the context of our self-timed movement task (Figure 7—figure supplement 4). Our mice were thirsty and perched near their usual juice tube, but “offline” DAN stimulation/inhibition did not alter licking behavior, even though we applied the same optical power that robustly altered movement probability during the self-timed movement task. Coddington and Dudman (2018) used a Pavlovian task in which they detected reward-anticipatory body movements of mice perched in a harness. Optogenetic stimulation of DANs did not affect anticipatory movements on the concurrent trial (except with stimulation well above the calibrated physiological range), and never evoked anticipatory licking at any strength. Pan et al. (2021) examined freely behaving mice executing conditioned approach behavior toward a reward port in response to an auditory cue. Optogenetic stimulation of DANs did not substitute for the conditioned stimulus in evoking approach. In a follow-up study, Coddington et al. (2021) paired DAN stimulation with an auditory cue in mice, but the stimulation had no effect on anticipatory movements or licking on the *concurrent* trial. Lee et al. (2020) found that optogenetic inhibition of mouse DANs at the same time as an olfactory conditioned stimulus had no effect on anticipatory licking on the same trial, even though inhibition at the time of the reward delivery was potent in reducing the probability and rate of anticipatory licking on *subsequent* trials. Saunders et al. (2018) examined conditioned approach behavior in TH-cre transgenic rats, and found that optogenetic DAN stimulation delivered at the same time as cue presentation enhanced cueapproach behavior in *subsequent* trials and sessions, but did not induce approach behavior on its own, in the absence of the cue. Morrens et al. (2020) used an olfactory conditioning task in mice and stimulated DANs simultaneously with presentation of a familiar, non-rewarded stimulus. Anticipatory licking was unaffected, indicating that DAN stimulation was insufficient to induce licking. Finally, Maes et al. (2020) examined aspects of classical conditioning in rats while optogenetically inhibiting DANs. DAN inhibition in isolation did not block conditioned food-approach behavior in several control experiments; it only affected behavior on *subsequent* trials.

In the aforementioned studies, the rodents were thirsty or hungry and regularly made conditioned movements in anticipation of rewards, yet the vast majority of the studies found that optogenetic manipulation of DANs had no effect on conditioned movements *on the same trial*. However, in most of those studies, optogenetic modulation affected conditioned movements on *subsequent* trials (a learning effect rather than a real-time, *same-trial* effect), providing a positive control for the effectiveness of DAN stimulation. Thus the *preponderance* of evidence argues against a simple scheme whereby (1) modulating DAN activity leads to a change in motivation/reward-expectation that (2) automatically evokes or suppresses reward-related movements on the same trial. The fact that we *were* able to affect the timing of reward-related movements in our experiment thus suggests there are other factors at play for self-timed movements.

How we have addressed these issues in the revised Discussion.

– We open the revised Discussion with the question of why ramping is seen in some contexts (locomoting toward a reward goal; being propelled through a VR-environment toward reward; self-timing movement) and not others (classical conditioning). In this light, we make the following arguments:

– We point out that reward expectation is presumably a common factor in these ramp-producing tasks, and thus provides a parsimonious explanation for all DAN ramping—but we also contend that reward expectation *alone* is insufficient to produce ramping, as we argued above.

– We then refer to the Gershman model that posits that reducing uncertainty about the value trajectory— which can happen exogenously (e.g., VR-environment) or endogenously (e.g., movement through environment toward reward, self-timed movement, etc.)—can lead to ramping, as opposed to tasks with temporal uncertainty (e.g., simple classical conditioning—Fiorillo et al. notwithstanding).

– We then address our optogenetic stimulation/inhibition results, pointing out that the modulation of movement timing that we observed suggests that the DAN signals are not “passive” monitors of reward expectation, but can influence behavior. The question then is *how* does optogenetic modulation of DANs influence behavior.

– We first consider the exact hypothesis offered by Reviewer #4—that modulating DANs could affect motivation/reward-expectation, leading to real-time changes in reward-obtaining movements—but we also point out the preponderance of evidence that suggests DAN activation *alone* is typically insufficient to produce reward-related movements, even in motivated (thirsty/hungry) animals.

– We then discuss hypothetical possibilities as to why optogenetic modulation of DANs was able to affect the timing of self-timed movements, including (1) that endogenous ramping signals might sum with exogenous (optogenetic) activation of DANs to produce supra-heightened motivation, leading to reward-obtaining movement; (2) that the explicit timing requirement in our task may somehow allow exogenous modulation of DANs to affect movement timing; and (3) that the pre-potency or preparation for movement during the delay before self-timed movements might allow DAN modulation to affect movement timing.

Overall, we conclude that ramping DAN signals likely represent aspects of reward expectation, and that, at least during self-timed movements, these ramping signals could causally influence the probability (and thus the timing) of movement. We believe our logic addresses the reviewers’ lingering concerns about the interpretation of our findings, while also acknowledging the constraints posed in the literature concerning (1) reward expectation and DAN ramping; and (2) real-time behavioral effects of DAN optogenetic manipulation during reward-related behavior.